

# Exploring superconformal Yang-Mills theories through matrix Bessel kernels

**Zoltan Bajnok[1], Bercel Boldis[1,2] and Gregory P. Korchemsky[3]**

**1** HUN-REN Wigner Research Centre for Physics, Konkoly-Thege Miklos ut 29-33,
1121 Budapest, Hungary
**2** Budapest University of Technology and Economics Műegyetem rkp. 3.,
1111 Budapest, Hungary
**3** Institut de Physique Théorique,[†] Université Paris Saclay,
CNRS, 91191 Gif-sur-Yvette, France

## Abstract

A broad class of observables in four-dimensional $\mathcal{N} = 2$ and $\mathcal{N} = 4$ superconformal Yang-Mills theories can be exactly computed in the planar limit for arbitrary 't Hooft coupling as Fredholm determinants of integrable Bessel operators. These observables admit a unifying description through a one-parameter generating function, which possesses a determinant representation involving a matrix generalization of the Bessel operator. We analyze this generating function over a wide range of parameter values and finite 't Hooft coupling. We demonstrate that it has a well-behaved weak-coupling expansion with a finite radius of convergence. In contrast, the strong-coupling expansion exhibits factorially growing coefficients, necessitating the inclusion of non-perturbative corrections that are exponentially suppressed at strong coupling. We compute these non-perturbative corrections and observe a striking resemblance between the resulting trans-series expansion of the generating function and the partition function of a strongly coupled theory expanded in powers of a mass gap.



## Contents

---

[†]Unité Mixte de Recherche 3681 du CNRS.

# 1  Introduction and summary

Computing observables in four-dimensional superconformal Yang-Mills theories in the planar limit for finite 't Hooft coupling $\lambda = g_{\text{YM}}^2 N$ remains a challenging task. While perturbative methods are well-suited at weak coupling, and techniques like localization, integrability, and AdS/CFT correspondence have provided valuable insights at strong coupling [1–3], a comprehensive understanding of dynamics of these theories across the entire coupling range remains an open problem.

In recent years, a significant progress has been achieved in computing various observables specified below. A common feature of these observables is that they exhibit well-behaved weak coupling expansions with finite radii of convergence, but their strong coupling expansions involve factorially growing coefficients, necessitating the inclusion of non-perturbative correc-

tions that are exponentially suppressed at strong coupling, $\lambda \gg 1$. The resulting expressions at strong coupling take a form of a transseries [4]

$$\mathcal{F}(g) = \mathcal{F}_0(g) + g^{a_1} e^{-8\pi g x_1} \mathcal{F}_1(g) + g^{a_2} e^{-8\pi g x_2} \mathcal{F}_2(g) + \dots , \tag{1}$$

where $g = \sqrt{\lambda}/(4\pi)$ and the coefficient functions $\mathcal{F}_n(g)$ are given by series in $1/g$ with factorially growing coefficients. In spite of the fact that the nonperturbative corrections are exponentially small at strong coupling, they are essential for understanding the dynamics of gauge theories at finite coupling. They provide a crucial link between physics at weak and strong coupling and shed a new light into the AdS/CFT correspondence.

In the context of the AdS/CFT correspondence, non-perturbative effects in (1) are connected to specific non-perturbative configurations in the dual string sigma-model on AdS space, such as string worldsheet instantons. In superconformal Yang-Mills theories, these configurations are associated with various emergent phenomena at strong coupling, including flux tube formation and the appearance of a mass gap. For example, the cusp anomalous dimension in $\mathcal{N} = 4$ super Yang-Mills theory, a fundamental quantity characterizing the ultraviolet behavior of Wilson loops with a cusp [5,6], has a leading non-perturbative correction at strong coupling related to the mass gap of the two-dimensional $O(6)$ non-linear sigma model

$$\Lambda^2 = \mu^2 \bar{g}^{-2\beta_1/\beta_0^2} e^{-\frac{1}{\beta_0 \bar{g}^2}} , \tag{2}$$

where $\bar{g}^2(\mu) = 1/(2g)$ is the coupling constant of this model defined at the scale $\mu$ and $\beta_n$ (with $n = 0, 1$) are the first two coefficients of the beta-function. The $O(6)$ model emerges as a low-energy effective theory describing excitations of a folded string spinning in AdS in the dual description of the cusp anomalous dimension [7–10]. The nonperturbative scales in (1) have the form similar to (2) but their interpretation differs and depends on the specific observable under consideration.

In this paper, we study a special class of observables in four-dimensional superconformal Yang-Mills theories which, in the planar limit and for an arbitrary 't Hooft coupling constant, admit a representation as a Fredholm determinant of a certain semi-infinite matrix $\mathbb{K}$ depending on some parameters denoted as $\alpha$

$$Z_\ell = e^{\mathcal{F}_\ell(\alpha)} = \det\left(\delta_{nm} + \mathbb{K}_{nm}(\alpha)\right)\Big|_{1+\ell \leq n, m < \infty} . \tag{3}$$

The explicit expression of the matrix and the value of nonnegative $\ell$ depend on the choice of the observable and the gauge theory. The determinant representation (3) has previously appeared in the study of various observables in maximally supersymmetric $\mathcal{N} = 4$ and superconformal $\mathcal{N} = 2$ Yang-Mills theories, see e.g. [11–23].

Recent example comes from the study of multi-gluon scattering amplitudes [15, 24] and form factors of half-BPS operators [25–27] in planar $\mathcal{N} = 4$ SYM theory. In this case, the semi-infinite matrix in (3) has the following general form

$$\mathbb{K}(\alpha) = 2\cos\alpha \begin{bmatrix} \cos\alpha\, K_{oo} & \sin\alpha\, K_{oe} \\ -\sin\alpha\, K_{eo} & \cos\alpha\, K_{ee} \end{bmatrix} . \tag{4}$$

It depends on the angle $0 \leq \alpha \leq \pi$ and consists of four semi-infinite blocks. The entries in each block are expressed in terms of the matrix elements $K_{nm}$ with indices of the parity indicated by the subscript, $(K_{oo})_{nm} = K_{2n-1,2m-1}$, $(K_{oe})_{nm} = K_{2n-1,2m}$ etc. The matrix elements $K_{nm}$ admit an integral representation

$$K_{nm} = 2m \int_0^\infty \frac{dt}{t} \frac{J_n(2gt) J_m(2gt)}{e^t - 1} , \tag{5}$$

where $J_n(x)$ is the Bessel function. The dependence on the coupling constant enters through the argument of the Bessel functions. In this paper, we apply a powerful technique developed in [28, 29] to compute (3) for arbitrary 't Hooft coupling.

For $\alpha = 0$, the off-diagonal blocks in (4) vanish, and the determinant (3) factorizes into a product of Fredholm determinants of the diagonal blocks. These determinants govern the correlation functions of infinitely heavy half-BPS operators in planar $\mathcal{N} = 4$ SYM, known as octagons [12–14, 30, 31]. Moreover, with a slightly modified function multiplying the Bessel functions in (5), the determinant (3) yields the leading non-planar correction to the partition function of $\mathcal{N} = 2$ SYM theory [18–20, 22, 23].

For $\alpha = \pi/4$, the matrix (4) is known as the BES kernel [11]. It underlies the integral equation that governs the dependence of the cusp anomalous dimension on the coupling constant in planar $\mathcal{N} = 4$ SYM. The solution to this equation is expressed as a ratio of the determinants (3) evaluated at $\alpha = \pi/4$ (see (11) and (12) below). For general angles $\alpha = \pi r$ with rational $r$, the determinant (3) emerges in the analysis of multi-gluon scattering amplitudes and form factors of half-BPS operators in planar $\mathcal{N} = 4$ SYM theory [15, 24–27].

Another motivation for studying the determinants (3) stems from the observation that, up to a similarity transformation, the diagonal blocks of the semi-infinite matrix (4) coincide with the integrable Bessel kernels [32,33]. The Fredholm determinants of these kernels possess numerous remarkable properties and are intimately connected to the Tracy-Widom distribution in random matrix theory [34]. This distribution characterizes the statistics of eigenvalue spacings in the Laguerre ensemble near the hard edge, where the eigenvalue density vanishes abruptly [35]. The off-diagonal blocks in the matrix (4) introduce the interaction between the two ensembles described by the diagonal blocks of (4), leading to non-trivial modifications in the eigenvalue distributions. The situation here closely resembles that of the two-dimensional Ising model. In this model, correlation functions can be computed as Toeplitz determinants with scalar symbols [36]. When line defects or boundaries are introduced, these correlation functions are expressed as block Toeplitz determinants with matrix symbols [37, 38]. Similar block Toeplitz determinants also arise in the computation of entanglement entropy for specific two-dimensional fermionic systems [39].[1]

We compute the Fredholm determinant (3) for arbitrary angle $0 \leq \alpha \leq \pi/2$ and show that the function $\mathcal{F}_\ell$ is given at strong coupling by

$$\mathcal{F}_\ell(\alpha) = \pi(1 - 4a^2)g - \left(\tfrac{1}{4} + \ell + a^2\right)\log(8\pi g) + B_\ell(a) + \mathcal{F}^{(0)}(g) + \sum_{n,m} \Lambda_-^{2n}\Lambda_+^{2m}\,\mathcal{F}^{(n,m)}(g), \quad (6)$$

where $a = \alpha/\pi$. At large $g$, each successive term on the right-hand side is smaller than the preceding one. The first three terms constitute a generalization of Szegő-Akhiezer-Kac formula [40–43] for the matrix generalization of the Bessel kernel (4). For the special case of $\ell = 0$, the first two terms in (6) were derived in [15]. The constant term $B_\ell$ is conventionally called the Widom-Dyson constant [44], its explicit expression is given by (112) below.

The last two terms on the right-hand side of (6) vanish at strong coupling. They form a transseries dependent on two exponentially small parameters

$$\Lambda_-^2 = g^{2a}e^{-4\pi g(1-2a)}, \qquad \Lambda_+^2 = g^{-2a}e^{-4\pi g(1+2a)}, \quad (7)$$

and involve the coefficient functions $\mathcal{F}^{(0)}(g)$ and $\mathcal{F}^{(n,m)}(g)$ given by series in $1/g$. The expansion coefficients of these series exhibit factorial growth at large orders, rendering the expansion (6) formal. While regularizing the Borel singularities of these series makes them finite, it also introduces an intrinsic ambiguity. For the strong coupling expansion (6) to be well-defined,

---

[1]We would like to thank Yizhuang Liu for bringing this to our attention.

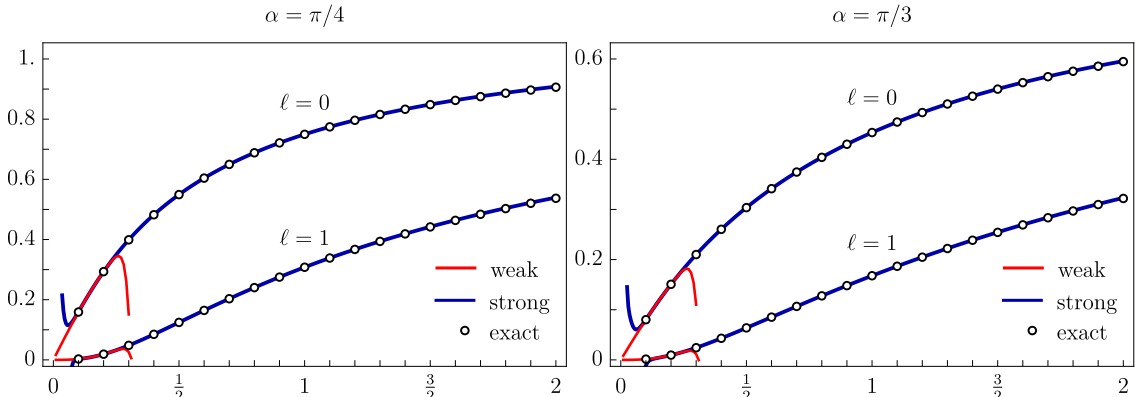

Figure 1: Comparison of the numerical values of the Fredholm determinant (3) with the weak and the strong coupling expansion for different values of $\alpha$ and $\ell$. The horizontal and vertical axes represent the coupling constant $g = \sqrt{\lambda}/(4\pi)$ and the function $\mathcal{F}_\ell/(2g)$, respectively.

this ambiguity has to cancel in the sum of all terms in (6). This leads to the set of nontrivial resurgence relations [45–47] connecting the coefficient functions in (6). We compute the functions $\mathcal{F}^{(0)}(g)$ and $\mathcal{F}^{(n,m)}(g)$ and verify that they indeed satisfy these relations.

In the dual holographic correspondence, the asymptotic behavior (6) arises naturally from the semiclassical expansion of the logarithm of a string partition function. From this viewpoint, it is more appropriate to examine the expansion of $Z = e^{\mathcal{F}_\ell(\alpha)}$ rather than the function itself,

$$Z_\ell = Z^{(0)}\left( 1 + \Lambda_-^2 Z^{(1,0)} + \Lambda_+^2 Z^{(0,1)} + \Lambda_-^2 \Lambda_+^2 Z^{(1,1)} + \Lambda_-^4 \Lambda_+^2 Z^{(2,1)} + \Lambda_-^2 \Lambda_+^4 Z^{(1,2)} + \dots \right). \quad (8)$$

The leading term, $Z^{(0)}$, can be interpreted as the result of a semiclassical calculation around the dominant saddle point, while the exponentially suppressed terms stem from contributions of the subdominant saddle points. Uncovering a dual holographic interpretation of (8) remains an interesting open problem.

All terms on the right-hand side of (8) can be expressed through the coefficient functions in (6). The advantage of the expansion (8) as compared with (6) is that, as we show below, infinitely many coefficient functions $Z^{(n,m)}$ vanish. Consequently, the relation (8) takes on a significantly simpler form (see (148) below). We calculate the first few nonvanishing coefficient functions in (8) and show that, when combined with the weak coupling expansion, the expansion (8) enables the computation of the observable (3) for arbitrary value of the coupling constant (see figure 1).

The relation (8) exhibits a striking resemblance to the expansion of the partition function of a strongly coupled theory in powers of a mass gap (2) (see [48]). However, it involves two different nonperturbative scales (7) which are polynomially independent for arbitrary angle $\alpha = a\pi$. For the special rational values of $a = p/(2(p+2))$ these scales are related to each other as $\Lambda_+^2 = g^{-p}\Lambda_-^{2p+2}$. In this case, the expansion (8) runs in powers of $\Lambda_-^2$ which assumes the expected form (2) upon the identification $\bar{g}^2 = 1/(kg)$, with the corresponding beta-function coefficients given by

$$\beta_0 = \frac{k}{8\pi}(p+2), \qquad \beta_1 = \left(\frac{k}{8\pi}\right)^2 p(p+2), \quad (9)$$

where $p$ is a non-negative integer and $k$ is an arbitrary normalization factor. Notably, for $p = 4/(n-4)$ and $k = n-4$, the expressions in (9) coincide with the well-known two-loop beta-

function of the two-dimensional $O(n)$ nonlinear sigma model [49]. In particular, for $n = 6$, corresponding to $a = 1/4$, this aligns with the previously mentioned connection to the $O(6)$ model [7–10]. The interpretation of the beta-function coefficients (9) and the relationship of (8) to two-dimensional models remains an intriguing open question, warranting further exploration.

The paper is organised as follows. In section 2 we establish the connection between the semi-infinite matrix (4) and the matrix generalization of the Bessel kernel and discuss the properties of the Fredholm determinant (3) at weak and strong coupling. In section 3 we derive the system of integro-differential equations for the function $\mathcal{F}_\ell$ defined in (3), valid for any coupling constant. In section 4 we construct the asymptotic solution of these equations at strong coupling and use it to compute the perturbative function $\mathcal{F}^{(0)}(g)$ in (6). In section 5 we compute the Fredholm determinant (3) in the limit $\alpha \to \pi/2$ and use it to derive the expression for the Widom-Dyson constant $B_\ell$. In section 6 we employ the method of differential equations to compute the non-perturbative coefficient functions $\mathcal{F}^{(n,m)}(g)$. In section 7 we study the large order behaviour of the series defining the coefficient functions $\mathcal{F}^{(0)}(g)$ and $\mathcal{F}^{(n,m)}(g)$ and show that they satisfy the resurgence relations. Technical details of the calculations are summarised in appendices.

## 2 Matrix generalization of the Bessel kernel

The function $\mathcal{F}_\ell$ defined in (3) depends on 't Hooft coupling constant

$$g = \frac{\sqrt{\lambda}}{4\pi}, \tag{10}$$

the angle $0 \le \alpha < \pi$ and nonnegative integer $\ell$. For $\ell \ge 1$ the determinant on the right-hand side of (3) involves the semi-infinite matrix cut down from the matrix $\mathbb{K}_{nm}(\alpha)$ (with $n, m \ge 1$) by removing the first $\ell$ rows and columns.

In application to $\mathcal{N} = 4$ SYM, we are interested in computing (3) for $\ell = 0$ and rational values of $a = \alpha/\pi$. For the sake of simplicity, we will consider this ratio to be arbitrary. Another interesting quantity related to the semi-infinite matrix (4) is the so-called tilted cusp anomalous dimension [15]. It is defined as the top left component of the resolvent

$$\Gamma_\alpha = 4g^2 \left[ \frac{1}{1 + \mathbb{K}(\alpha)} \right]_{11}. \tag{11}$$

For $\alpha = 0$ and $\alpha = \pi/4$ it coincides with the octagon and cusp anomalous dimensions, respectively. The rationale for introducing the parameter $\ell$ in (3) is that, according to the Cramer's rule, the matrix element in (11) is given by the ratio of the determinants (3) evaluated at $\ell = 1$ and $\ell = 0$, or equivalently

$$\left[ \frac{1}{1 + \mathbb{K}(\alpha)} \right]_{11} = \frac{Z_{\ell=1}}{Z_{\ell=0}} = e^{\mathcal{F}_{\ell=1} - \mathcal{F}_{\ell=0}}. \tag{12}$$

Substituting this relation into (11) we can obtain the representation of $\Gamma_\alpha$ in terms of the function (3).

The diagonal blocks of (4) contain the matrices $K_{oo}$ and $K_{ee}$, whose entries are given by (5) for odd and even indices, respectively. The matrix (5) is not symmetric under the exchange of indices but this symmetry can be restored by a similarity transformation

$$K_{nm} \to \sqrt{n/m} K_{nm}. \tag{13}$$

This transformation does not affect the Fredholm determinant (3) but it allows us to bring the matrix (5) to the form of the so-called truncated Bessel kernel [50]

$$K_{nm} = \int_0^\infty dx\, \psi_n(x) \chi\left(\frac{\sqrt{x}}{2g}\right) \psi_m(x). \tag{14}$$

Here $\psi_n(x)$ (with indices $n \geq 1$) are given by the normalized Bessel functions

$$\psi_n(x) = \sqrt{n}\frac{J_n(\sqrt{x})}{\sqrt{x}}. \tag{15}$$

For indices of the same parity, these functions satisfy the orthogonality condition

$$\langle \psi_{2n-1} \psi_{2m-1} \rangle = \langle \psi_{2n} \psi_{2m} \rangle = \delta_{nm}, \tag{16}$$

where $\langle \psi_n \psi_m \rangle = \int_0^\infty dx\, \psi_n(x) \psi_m(x)$. For indices of different parity we have $\langle \psi_{2n} \psi_{2m-1} \rangle \neq 0$.

The function $\chi(x)$ is conventionally called the symbol of the Bessel matrix (14). In general, this function can be arbitrary but in order to match (5) we choose it to be

$$\chi(x) = \frac{2}{e^x - 1}. \tag{17}$$

This function suppresses the contribution to (14) from large $x \gg (2g)^2$ and plays the role of a cut-off.

The diagonal blocks $K_{oo}$ and $K_{ee}$ are given by the truncated Bessel kernel (14) with odd and even indices, respectively. An advantage of this identification is that their Fredholm determinants can be computed exactly [22].

## 2.1 Properties of the tilted Bessel kernel

As follows from (4), the matrix $\mathbb{K}(\alpha)$ is invariant under $\alpha \to \alpha + \pi$. Moreover, the off-diagonal elements in (4) change their sign under the transformation $\alpha \to -\alpha$ leading to $\mathbb{K}(-\alpha) = \sigma_3 \mathbb{K}(\alpha)\sigma_3$, where $\sigma_3$ is a Pauli matrix. Using this identity we obtain from (3)

$$\mathcal{F}_\ell(\alpha) = \mathcal{F}_\ell(-\alpha) = \mathcal{F}_\ell(\pi - \alpha). \tag{18}$$

This relation allows us to limit the possible values of the angle as

$$0 \leq \alpha \leq \frac{\pi}{2}. \tag{19}$$

At the end points $\alpha = 0$ and $\alpha = \pi/2$, the function $\mathcal{F}_\ell(\alpha)$ can be found in a closed form.

For $\alpha = \pi/2$ and the coupling constant $g$ held fixed, the matrix elements (4) vanish. Then, it follows from (3) that $\mathcal{F}_\ell(\pi/2) = 0$. For $\alpha = 0$, the off-diagonal elements of the matrix (4) vanish and, therefore, its Fredholm determinant (3) factorizes into the product of the determinants corresponding to the two diagonal blocks. This leads to

$$\mathcal{F}_\ell(0) = F_\ell + F_{\ell+1}, \tag{20}$$

where the function $F_\ell$ is given by either $\log \det(1 + K_{oo})$ or $\log \det(1 + K_{ee})$, depending on the parity of $\ell$. This function can be written in a concise form as[2]

$$F_\ell = \log \det \left(\delta_{nm} + K_{2n+\ell-1,2m+\ell-1}\right)\Big|_{1 \leq n,m < \infty}. \tag{21}$$

---

[2]The function (21) has previously appeared in the study of the octagon form factor. In notations of [22], $F_\ell(g) = \mathcal{F}_\ell^{\text{BES}}(g)$.

For lowest values $\ell = 0, 1, 2$, it can be expressed in terms of hyperbolic functions for any coupling constant (see (36) below). Starting from $\ell = 3$ its expression involves special functions [51]. For our purposes we will not need their explicit expressions.

Using the results of [22] we obtain from (20)

$$\mathcal{F}_{\ell=0}(0) = \frac{1}{4} \log\left( \frac{\sinh(4\pi g)}{4\pi g} \right),$$

$$\mathcal{F}_{\ell=1}(0) = \frac{1}{4} \log\left( \frac{\sinh(4\pi g)}{4\pi g} \right) + \log\left( \frac{\log \cosh(2\pi g)}{2\pi^2 g^2} \right). \tag{22}$$

We would like to emphasize that these relations are exact and they hold for any coupling constant (10).

## 2.2 Weak coupling expansion

At weak coupling, the matrix elements (5) and (14) can be expanded in powers of $g$

$$K_{nm} = g^{n+m} \frac{2\sqrt{nm}\,\Gamma(n+m)}{\Gamma(n+1)\Gamma(m+1)} \zeta_{n+m} + O(g^{n+m+2}), \tag{23}$$

where Riemann zeta values $\zeta_n \equiv \zeta(n)$ arise from integration of the symbol function (17) against $x^{n-1}$. Taking into account this relation we can expand the determinant (3) in powers of the matrix (4)

$$\mathcal{F}_\ell = \mathrm{tr}\,\mathbb{K}_\ell - \frac{1}{2}\,\mathrm{tr}\,(\mathbb{K}_\ell^2) + \frac{1}{3}\,\mathrm{tr}\,(\mathbb{K}_\ell^3) + \dots, \tag{24}$$

where the semi-infinite matrix $\mathbb{K}_\ell$ is obtained from the matrix (4) by removing the first $\ell$ rows and columns.

It follows from (4) and (23) that $\mathrm{tr}\,\mathbb{K}_\ell = \sum_{n \geq \ell+1} \mathbb{K}_{nn} = O(g^{2(\ell+1)})$. Each subsequent term in (24) is suppressed by the factor of $g^{2(\ell+1)}$. Going through the calculation, we find the contribution of the first two terms in (24)

$$\mathcal{F}_\ell(\alpha) = 2c^2 g^{2(\ell+1)} \frac{\Gamma(2\ell+3)}{\Gamma^2(\ell+2)} \left[ \zeta_{2\ell+2} - 2g^2 \zeta_{2\ell+4} \frac{(2\ell+1)(2\ell+3)}{\ell+2} + \dots \right] \tag{25}$$

$$- 2c^2 g^{4(\ell+1)} \frac{\Gamma^2(2\ell+3)}{\Gamma^4(\ell+2)} \left[ c^2 \zeta_{2\ell+2}^2 - 8g^2 \left( c^2 \zeta_{2\ell+2} \zeta_{2\ell+4} \frac{(\ell+1)(2\ell+3)}{\ell+2} + s^2 \zeta_{2\ell+3}^2 \frac{\ell+1}{\ell+2} \right) + \dots \right]$$

$$+ O(g^{6(\ell+1)}),$$

where the notation was introduced for $c = \cos\alpha$ and $s = \sin\alpha$. The first and second lines in (25) come from the weak coupling expansion of $\mathrm{tr}\,\mathbb{K}_\ell$ and $\mathrm{tr}\,(\mathbb{K}_\ell^2)$, respectively.

It is straightforward to compute subleading terms in (25). The expansion coefficients in (25) are homogenous polynomials in zeta values. Note that the first line in (25) contains only even zeta values $\zeta_{2n} \sim \pi^{2n}$. The odd zeta values $\zeta_{2n+1}$ first appear at order $O(g^{4\ell+6})$ and they accompanied by powers of $\sin\alpha$.

These properties follow from a peculiar form of the matrix (4). It is convenient to interpret the matrix (4) as defining a Hamiltonian. The diagonal blocks in (4) describe two independent quantum mechanical systems, and the off-diagonal blocks describe their interaction. For $\alpha = 0$ the interaction is switched off and a logarithm of the spectral determinant $\mathcal{F}_\ell(0)$ splits into the sum of two functions (20) corresponding to the two systems.

For $\alpha = 0$, the relation (25) accurately reproduces the weak coupling expansion of (22) and involves powers of $\pi^2$. The odd zeta values are present in (25) for $\alpha \neq 0$ due to the interaction described by the off-diagonal elements of (4). Their leading contribution to (25) is $4c^2 s^2 \, \mathrm{tr}\,(K_{\mathrm{oe}} K_{\mathrm{eo}})$ and produces the term on the second line of (25) proportional to $\zeta_{2\ell+3}^2$.

We can show following [29] that the weak coupling expansion (25) has a finite radius of convergence. For $\alpha = 0$ we can apply (22) to verify that $\mathcal{F}_{\ell=0,1}(0)$ has a logarithmic cut at $g^2 = -1/16$. The same is true for arbitrary $\ell$ and $\alpha$. To see this, it suffices to examine convergency properties of the integral in (5) for large $t$. For negative $g^2$, or equivalently $g = ih$ with $h$ real positive, the Bessel functions in (5) grow exponentially fast $J_n(2iht) \sim e^{2ht}/\sqrt{t}$ as $t \to \infty$, whereas the symbol function (17) decreases as $\chi(t) \sim e^{-t}$. As a result, the integral in (5) scales at large $t$ as $K_{nm} \sim \int^\infty dt\, t^{-2} e^{-t(1-4h)}$ and generates a singularity at $h = 1/4$. A close examination shows (see [29] for details) that for $h \to 1/4$ each term in (24) develops a logarithmic singularity $\log(h^2 - 1/16)$.

## 2.3 Strong coupling expansion

The expansion (24) is not applicable at strong coupling because all terms become equally important and have to be taken into account. As we demonstrate below, the strong coupling expansion of $\mathcal{F}_\ell(\alpha)$ can be worked out using the technique developed in [28, 29].

The strong coupling expansion of the function $F_\ell$ defined in (21) was derived in [22]. Its substitution into (20) yields the strong coupling expansion of $\mathcal{F}_\ell(\alpha)$ at $\alpha = 0$

$$\mathcal{F}_\ell(0) = \pi g - \left(\tfrac{1}{4} + \ell\right) \log(8\pi g) + B_\ell(0) + O(1/g) + O(e^{-4\pi g}). \qquad (26)$$

Here the second term, proportional to $\log(8\pi g)$, is a manifestation of the Fisher-Hartwig singularity [52] of the symbol function (17). The Widom-Dyson constant $B_\ell(0)$ is given by (110) below and the $O(1/g)$ term denotes a 'perturbative' series in $1/g$. The last term in (26) represents 'nonperturbative', exponentially small correction. It is given by series in $e^{-4\pi g}$ with the coefficients which are themselves polynomial in $1/g$. For $\ell = 0, 1$ the last two terms in (26) can be determined by matching (26) to the exact expressions (22).

We show below that for arbitrary $0 \le \alpha < \pi/2$, the function $\mathcal{F}_\ell(\alpha)$ has the form (6). For $\alpha = 0$ it coincides with (26). In virtue of (18) each term in (6) has to be invariant under $a \to -a$. For the first two terms in (6) this property is manifest. The remaining terms have to be even functions of the angle

$$B_\ell(-a) = B_\ell(a), \qquad \Delta\mathcal{F}_\ell(-\alpha, g) = \Delta\mathcal{F}_\ell(\alpha, g). \qquad (27)$$

Here the function $\Delta\mathcal{F}_\ell(\alpha, g)$ represents the sum of subleading perturbative and exponentially small, nonperturbative corrections to (6). It takes the form of a transseries

$$\begin{aligned}
\Delta\mathcal{F}_\ell(\alpha, g) = {} & \mathcal{F}_\ell^{(0)}(\alpha, g) + \Lambda_-^2 \mathcal{F}_\ell^{(1,0)}(\alpha, g) + \Lambda_+^2 \mathcal{F}_\ell^{(0,1)}(\alpha, g) \\
& + \Lambda_-^4 \mathcal{F}_\ell^{(2,0)}(\alpha, g) + \Lambda_-^2 \Lambda_+^2 \mathcal{F}_\ell^{(1,1)}(\alpha, g) + \Lambda_+^4 \mathcal{F}_\ell^{(0,2)}(\alpha, g) + \dots,
\end{aligned} \qquad (28)$$

where the superscript of the coefficient function $\mathcal{F}^{(n,m)}$ refers to the factor of $\Lambda_-^{2n} \Lambda_+^{2m}$ that it accompanies. The expansion on the right-hand side of (28) runs in powers of two angle-dependent exponentially small parameters (7), which are related to each other through the substitution $a \to -a$. For $a = 0$ these parameters are identical and the relations (6) and (28) reduce to the simpler form (26).

## 2.4 From semi-infinite matrices to integral operators

It is advantageous to interpret the semi-infinite matrix (4) as representing a particular integral operator and reformulate (3) as its Fredholm determinant. In the next section, we utilise the properties of this operator to compute the function $\mathcal{F}_\ell$ using the method of differential equations.

To begin with, we define an operator $\boldsymbol{\chi}$ which acts on a test function $f(x)$ as[3]

$$\boldsymbol{\chi} f(x) = \chi\left(\frac{\sqrt{x}}{2g}\right) f(x).$$ (29)

Using this operator, we can express (14) as matrix elements of $\boldsymbol{\chi}$ with respect to the normalized Bessel functions (15)

$$K_{nm} = \langle\psi_n|\boldsymbol{\chi}|\psi_m\rangle.$$ (30)

For indices $n$ and $m$ of the same parity, the Fredholm determinant of the corresponding matrices $K_{\mathrm{oo}}$ and $K_{\mathrm{ee}}$ can be evaluated as

$$\det(1 + K_{\mathrm{oo}}) = \det(\delta_{nm} + \langle\psi_{2n-1}|\boldsymbol{\chi}|\psi_{2m-1}\rangle) = \det(1 + \boldsymbol{K}_{\mathrm{oo}}\boldsymbol{\chi}),$$
$$\det(1 + K_{\mathrm{ee}}) = \det(\delta_{nm} + \langle\psi_{2n}|\boldsymbol{\chi}|\psi_{2m}\rangle) = \det(1 + \boldsymbol{K}_{\mathrm{ee}}\boldsymbol{\chi}).$$ (31)

Here on the right-hand side we used the cyclic property of the determinant and introduced the operators

$$\boldsymbol{K}_{\mathrm{oo}} = \sum_{n\geq 1}|\psi_{2n-1}\rangle\langle\psi_{2n-1}|, \qquad \boldsymbol{K}_{\mathrm{ee}} = \sum_{n\geq 1}|\psi_{2n}\rangle\langle\psi_{2n}|.$$ (32)

They are given by an infinite sum of the normalized Bessel functions (15) with indices of definite parity. Due to the orthogonality condition (16), the operators (32) have the properties of the projectors on spaces spanned by functions $\psi_n(x)$ with odd and even indices, respectively.

The operators (32) are special cases of the integral Bessel operator defined as

$$\boldsymbol{K}_\ell f(x) = \int_0^\infty dy\, K_\ell(x,y) f(y),$$
$$K_\ell(x,y) = \sum_{n\geq 1}\psi_{2n+\ell-1}(x)\psi_{2n+\ell-1}(y),$$ (33)

where $f(x)$ is a test function. Using the properties of the Bessel functions, the kernel of this operator can be found in a closed form [34]

$$K_\ell(x,y) = \frac{1}{2}\int_0^1 dt\, t J_\ell(t\sqrt{x}) J_\ell(t\sqrt{y}).$$ (34)

The operators (32) coincide with the Bessel operator for $\ell = 0$ and $\ell = 1$

$$\boldsymbol{K}_{\mathrm{oo}} = \boldsymbol{K}_{\ell=0}, \qquad \boldsymbol{K}_{\mathrm{ee}} = \boldsymbol{K}_{\ell=1}.$$ (35)

By applying the relations (31), we can express the functions $F_{\ell=0}$ and $F_{\ell=1}$ defined in (21) as Fredholm determinants of the operators $\boldsymbol{K}_{\mathrm{oo}}\boldsymbol{\chi}$ and $\boldsymbol{K}_{\mathrm{ee}}\boldsymbol{\chi}$, which are known as truncated Bessel operators. These integral operators possess remarkable properties that enable us to compute their Fredholm determinants, leading to the following expressions [22]

$$F_{\ell=0} = \log\det(1 + \boldsymbol{K}_{\mathrm{oo}}\boldsymbol{\chi}) = \frac{3}{8}\log\cosh(2\pi g) - \frac{1}{8}\log\frac{\sinh(2\pi g)}{2\pi g},$$
$$F_{\ell=1} = \log\det(1 + \boldsymbol{K}_{\mathrm{ee}}\boldsymbol{\chi}) = -\frac{1}{8}\log\cosh(2\pi g) + \frac{3}{8}\log\frac{\sinh(2\pi g)}{2\pi g}.$$ (36)

---

[3]In the following, we adopt boldface notation for the operators to distinguish them from their integral kernels.

These relations holds for arbitrary coupling. In agreement with (20), the sum of the two functions (36) yields $\mathcal{F}_{\ell=0}(0)$ in (22).

Let us now consider the matrix (4) and take into account its off-diagonal blocks. In a close analogy with (30), we can write down an operator representation of the matrix (4) in a compact form by introducing a matrix generalization of the operator (29)

$$\mathcal{X}f(x) = \chi\left(\frac{\sqrt{x}}{2g}\right)Uf(x),$$

$$U = \cos\alpha\, e^{i\sigma_2\alpha} = \cos\alpha \begin{pmatrix} \cos\alpha & \sin\alpha \\ -\sin\alpha & \cos\alpha \end{pmatrix}, \tag{37}$$

where $\sigma_2$ is the Pauli matrix.

In addition, we define two-component states

$$|\Psi_{2n-1}\rangle = \begin{pmatrix} |\psi_{2n-1}\rangle \\ 0 \end{pmatrix}, \qquad |\Psi_{2n}\rangle = \begin{pmatrix} 0 \\ |\psi_{2n}\rangle \end{pmatrix}. \tag{38}$$

These states are built out of normalized Bessel functions (15) and satisfy the orthogonality condition $\langle\Psi_n|\Psi_k\rangle = \delta_{kn}$. It is straightforward to verify that the entries of the matrix (4) are given by matrix elements of the operator (37) with respect to the states (38)

$$\mathbb{K}_{nm} = \langle\Psi_n|\mathcal{X}|\Psi_m\rangle, \tag{39}$$

where $n, m \geq 1$.

Taking into account the relation (39), we can rewrite (3) as a Fredholm determinant of an integral operator

$$e^{\mathcal{F}_\ell} = \det\left(\delta_{nm} + \langle\Psi_n|\mathcal{X}|\Psi_m\rangle\right)\Big|_{1+\ell \leq n,m < \infty} = \det\left(1 + \mathcal{H}_\ell\mathcal{X}\right). \tag{40}$$

Here in the second relation we introduced the notation for the operator

$$\mathcal{H}_\ell = \sum_{n \geq 1+\ell} |\Psi_n\rangle\langle\Psi_n|, \qquad \mathcal{H}_\ell\mathcal{H}_\ell = \mathcal{H}_\ell. \tag{41}$$

The kernel of the operator (41) is given by a diagonal matrix

$$\mathcal{H}_\ell(x,y) = \begin{bmatrix} K_\ell(x,y) & 0 \\ 0 & K_{\ell+1}(x,y) \end{bmatrix}, \tag{42}$$

whose nonzero entries are the Bessel kernels (34).

We conclude that the function $\mathcal{F}_\ell(\alpha)$ is given by a logarithm of the Fredholm determinant of the product of the operators defined in (37) and (41)

$$\mathcal{F}_\ell(\alpha) = \log\det(1 + \mathcal{H}_\ell\mathcal{X}). \tag{43}$$

This representation can be thought of as a matrix generalization of the analogous relation (36) for the Fredholm determinants of the truncated Bessel operators.

## 3 Method of differential equations

The representation (43) is the starting point for applying the method of differential equation [32, 34, 53]. This method can be readily applied to compute the Fredholm determinants (31) of the Bessel kernels (33) and (34).

In this section, we extend the method of differential equations to compute the Fredholm determinant (43) involving a matrix generalization of the Bessel kernel. We derive a system of integro-differential equations for the function $\mathcal{F}_\ell(\alpha)$ and show that these equations are highly efficient for obtaining the strong coupling expansion (6) and exploring its resurgence properties.

### 3.1 Differential equations

For the functions $F_{\ell=0}(g)$ and $F_{\ell=1}(g)$ defined in (36) the differential equations were derived in [14,16]. The derivation of the differential equations for the function (43) follows the same steps but it is slightly more complicated due to a matrix form of the symbol (37) (see appendix A for details).

We begin by defining a diagonal $2 \times 2$ matrix $\Phi_\ell(x)$ whose nonzero entries are given by the Bessel functions $\phi_\ell(x)$ and $\phi_{\ell+1}(x)$

$$|\Phi_\ell\rangle\rangle = \begin{pmatrix} |\phi_\ell\rangle & 0 \\ 0 & |\phi_{\ell+1}\rangle \end{pmatrix}, \qquad \phi_\ell(x) = J_\ell(\sqrt{x}). \tag{44}$$

Here we use the double bracket notation to distinguish this matrix from the two-component states (38).

We apply (44) to define an auxiliary matrix

$$|Q\rangle\rangle = \frac{1}{1 + \mathcal{H}_\ell \mathcal{X}} |\Phi_\ell\rangle\rangle. \tag{45}$$

Denoting by $R(x,y)$ the kernel of the operator $1/(1 + \mathcal{H}_\ell \mathcal{X})$, we can rewrite this relation as

$$Q(x) = \int_0^\infty dy \, R(x,y) \Phi_\ell(y). \tag{46}$$

All functions in this relation are given by $2 \times 2$ matrices.

Finally, we define a $2 \times 2$ matrix conventionally called the potential

$$u = -\langle\langle \Phi_\ell | \mathcal{X} \frac{1}{1 + \mathcal{H}_\ell \mathcal{X}} |\Phi_\ell\rangle\rangle = -\langle\langle \Phi_\ell | \frac{1}{1 + \mathcal{X} \mathcal{H}_\ell} \mathcal{X} |\Phi_\ell\rangle\rangle. \tag{47}$$

It can be written explicitly as a double integral of the product of the symbol function (17) and the matrices defined above

$$u = -\int_0^\infty dx \int_0^\infty dy \, \chi\left(\frac{\sqrt{x}}{2g}\right) \Phi_\ell(x) U R(x,y) \Phi_\ell(y). \tag{48}$$

We show in appendix A that $u$ satisfies the matrix equation $u^t = \sigma_3 u \sigma_3$, where $\sigma_3$ is the Pauli matrix. As a consequence, $u$ has three independent components and its off-diagonal elements satisfy the relation $u_{12} = -u_{21}$.

According to their definition (46) and (48), the entries of $2 \times 2$ matrices $Q(x)$ and $u$ are real-valued functions depending on the coupling constant $g$, the angle $\alpha$ and the parameter $\ell$. The reason for introducing these auxiliary matrices is that, together with $\mathcal{F}_\ell(\alpha)$, they satisfy a closed system of equations. Its derivation can be found in appendix A.

- The first equation relates the derivative $g \partial_g \mathcal{F}_\ell(\alpha)$ and trace of the matrix (47)

$$g \partial_g \mathcal{F}_\ell(\alpha) = -\frac{1}{2} \operatorname{tr} u. \tag{49}$$

- The second equation expresses the derivative of the potential $g \partial_g u$ in terms of the matrix $Q(x)$

$$g \partial_g u = 2 \int_0^\infty dx \, (\sigma_3 Q^t(x) \sigma_3 U Q(x)) x \partial_x \chi\left(\frac{\sqrt{x}}{2g}\right), \tag{50}$$

where $\sigma_3$ is the Pauli matrix and $Q^t(x)$ denotes transposed matrix $Q(x)$.

- Finally, the last equation is a partial differential equation for the matrix $Q(x)$

$$\left(g\partial_g + 2x\partial_x\right)^2 Q(x) = Q(x)\left(L^2 - x + (g\partial_g - 1)u\right), \tag{51}$$

where $L = \mathrm{diag}(\ell, \ell+1)$ is a diagonal matrix.

The relations (50) and (51) define a (complicated) system of integro-differential equations for the matrices $u$ and $Q(x)$. By solving this system, we can determine $u$ and subsequently use (49) to compute $\mathcal{F}_\ell(\alpha)$ up to a few integration constants. The equations (49) – (51) represent a matrix generalization of the analogous relations for the functions (36) (see (C.13) in [23]). Importantly, these equations are exact and hold for any coupling constant.

It is easy to solve (50) and (51) at weak coupling and reproduce the weak coupling expansion (25). In this case, the matrices $Q(x)$ and $u$ can be obtained from (46) and (48) by replacing the resolvent $R(x,y) = \langle x|1/(1+\mathcal{H}_\ell\mathcal{X})|y\rangle$ by its expansion in powers of the matrix (42)

$$R(x,y) = \delta(x-y) - \mathcal{H}_\ell(x,y)\,U\chi\left(\frac{\sqrt{y}}{2g}\right) + \dots, \tag{52}$$

where the first term on the right-hand side is proportional to the identity matrix. In this way, we obtain from (48) and (46)

$$u = -\int_0^\infty dx\,\chi\left(\frac{\sqrt{x}}{2g}\right)\Phi_\ell(x)U\Phi_\ell(x) + \dots,$$

$$Q(x) = \Phi_\ell(x) - \int_0^\infty dy\,\chi\left(\frac{\sqrt{y}}{2g}\right)\mathcal{H}_\ell(x,y)U\Phi_\ell(y) + \dots, \tag{53}$$

where dots denote subleading terms suppressed by the factor of $g^{2(\ell+1)}$.

Replacing the matrices $U$ and $\Phi_\ell(x)$ with their expressions (37) and (44), we can expand the integrals in (53) in powers of the coupling constant to get

$$\begin{pmatrix} u_{11} \\ u_{12} \\ u_{22} \end{pmatrix} = 2cg^{2\ell+2}\frac{\Gamma(2\ell+3)}{\Gamma^2(\ell+2)}\begin{pmatrix} -4c(\ell+1)\zeta_{2\ell+2} + 16cg^2(\ell+1)(2\ell+3)\zeta_{2\ell+4} + \dots \\ -8sg(\ell+1)\zeta_{2\ell+3} + 16sg^3(2\ell+3)^2\zeta_{2\ell+5} + \dots \\ -8cg^2(2\ell+3)\zeta_{2\ell+4} + 32cg^4(2\ell+3)(2\ell+5)\zeta_{2\ell+6} + \dots \end{pmatrix}, \tag{54}$$

where $c = \cos\alpha$, $s = \sin\alpha$ and $u_{21} = -u_{12}$. Note that the matrix elements have different behaviour in $g$ and involve zeta values $\zeta_n$ with $n$ of different parity.

We checked that the weak coupling expansion of $\mathrm{tr}\,u = u_{11} + u_{22}$ is in agreement with (49) and (25). We also verified that the resulting expression for $Q(x)$ satisfies the relations (50) and (51).

## 3.2 Master formulae

At strong coupling, it is advantageous to change the integration variable in (50) to $z = \sqrt{x}/(2g)$. This transformation ensures that the argument of the cut-off function in (50) is independent of $g$ and eliminates square-root singularities in $Q(x)$.

Instead of dealing with the $2 \times 2$ matrix $Q_{ij}(x)$ we introduce two scalar functions

$$q_\pm(z) = e^{-i\alpha/2}(1,i)Q(x)\begin{pmatrix} 1 \\ \pm 1 \end{pmatrix} = e^{-i\alpha/2}\left[Q_{11}(x) \pm Q_{12}(x) + iQ_{21}(x) \pm iQ_{22}(x)\right], \tag{55}$$

where $x = (2gz)^2$. In contrast to $Q_{ij}(x)$, the functions $q_\pm(z)$ take complex values for real $z$ (see (D.9)). We can invert (55) by supplementing it with the analogous relation for complex conjugated functions

$$Q(x) = \frac{1}{4} e^{i\alpha/2} \left[ \begin{pmatrix} 1 & 1 \\ -i & -i \end{pmatrix} q_+(z) + \begin{pmatrix} 1 & -1 \\ -i & i \end{pmatrix} q_-(z) \right] + \text{c.c.}, \tag{56}$$

where the last term denotes a complex conjugated terms involving the functions $\bar{q}_\pm(z)$.

Combining together (51) and (56), we find that the functions $q_\pm(z)$ satisfy a system of differential equations

$$\left[ (g\partial_g)^2 + (2gz)^2 + W_0(g) \right] q_+(z) = W_-(g) q_-(z),$$
$$\left[ (g\partial_g)^2 + (2gz)^2 + W_0(g) \right] q_-(z) = W_+(g) q_+(z). \tag{57}$$

Here the dependence of $q_\pm(z)$ on the coupling constant is tacitly assumed and the notation was introduced for

$$W_0(g) = \frac{1}{2}(w_{11} + w_{22} - (\ell+1)^2 - \ell^2),$$
$$W_-(g) = \frac{1}{2}(w_{11} - w_{22} + (\ell+1)^2 - \ell^2) + w_{12}, \tag{58}$$
$$W_+(g) = \frac{1}{2}(w_{11} - w_{22} + (\ell+1)^2 - \ell^2) - w_{12},$$

and $w_{ij}$ are entries of the matrix $w$ defined as

$$w = u - g\partial_g u = -g^2 \partial_g (u/g). \tag{59}$$

As functions of the angle, $W_0(g|a)$ and $W_\pm(g|a)$ satisfy the relation

$$W_0(g|a) = W_0(g|-a), \qquad W_-(g|a) = W_+(g|-a). \tag{60}$$

It follows from the analogous relation for the matrix elements of $u$ (see (D.5)).

Taking into account (49), we find that the function $W_0(g)$ is related to the second derivative of the determinant (43) as

$$g^2 \partial_g^2 \mathcal{F}_\ell(\alpha) = W_0(g) + \frac{1}{2}\left((\ell+1)^2 + \ell^2\right). \tag{61}$$

This relation is consistent with the relations (60) and (18).

The additional relation between the functions (55) and (58) follows from (50). We use the identity $\partial_g w = -g\partial_g^2 u$ to obtain from (50) the following equations

$$\partial_g W_0(g) = -4g\partial_g \left[ g\cos\alpha \int_0^\infty dz\, \text{Re}\left(q_+(z)q_-(z)\right) z^2 \partial_z \chi(z) \right],$$
$$\partial_g W_-(g) = -4g\partial_g \left[ g\cos\alpha \int_0^\infty dz\, \text{Re}\left(q_+^2(z)\right) z^2 \partial_z \chi(z) \right], \tag{62}$$
$$\partial_g W_+(g) = -4g\partial_g \left[ g\cos\alpha \int_0^\infty dz\, \text{Re}\left(q_-^2(z)\right) z^2 \partial_z \chi(z) \right].$$

In the next section, we construct the solution to (57) and (62) at strong coupling.

The equations (57) and (62) are equivalent to the matrix relations (50) and (51). The rationale behind introducing specific linear combinations of the matrix elements (58) is that, as we show below (see (71)), the asymptotic behaviour of $W_0(g)$ and $W_\pm(g)$ is simpler at strong coupling than that of the matrix $u$.

# 4 Strong coupling expansion

In this section, we derive the strong coupling expansion of the function $\mathcal{F}_\ell(\alpha)$ defined in (43).

The leading behaviour of this function for $g \to \infty$ can be found following [16] as

$$
\mathcal{F}_\ell(\alpha) = -\frac{2g}{\pi} \int_0^\infty dz\, z\, \partial_z \log \det(1 + \chi(z)U) + O(g^0)
$$

$$
= -\frac{2g}{\pi} \int_0^\infty dz\, z\, \partial_z \log\left( \sin^2\alpha + (1 + \chi(z))^2 \cos^2\alpha \right) + O(g^0). \tag{63}
$$

This relation is a generalization of the first Szegő theorem for the matrix Bessel kernel defined in (42) and (34). Its derivation can be found in appendix B. Replacing the function $\chi(z)$ with its expression (17) we obtain from (63)

$$
\mathcal{F}_\ell(\alpha) = \pi\left( 1 - \frac{4\alpha^2}{\pi^2} \right) g - A_\ell(\alpha)\log(8\pi g) + B_\ell(\alpha) + O(1/g), \tag{64}
$$

where $\alpha$ satisfies (19). Here we have explicitly included the subleading $O(g^0)$ corrections which are parameterized by the coefficient functions $A_\ell(\alpha)$ and $B_\ell(\alpha)$.

We show below that the method of differential equations described in the previous section allows us to compute all subleading corrections to (64) except the constant term $B_\ell(\alpha)$. The reason for this is that by replacing (64) with (49), $\operatorname{tr} u$ no longer depends on $B_\ell(\alpha)$. The constant term is calculated in section 5 using a different technique.

## 4.1 Ansatz

To compute the function $\mathcal{F}_\ell(\alpha)$ using the relations (61) and (62), we need the expressions for the functions $q_\pm(z)$ defined in (55). These functions are given by linear combinations of real-valued functions $Q_{ij}(x)$ which are related to the resolvent $1/(1 + \mathcal{H}_\ell \mathcal{X})$ through the relation (45). Following [29] we can exploit this relation to derive integral equations for the matrix $Q(x)$. Solving these integral equations at strong coupling we can derive asymptotic expressions for the functions $Q(x)$ and $q_\pm(z)$. The details of the calculation can be found in appendix C.

The resulting expression for the functions $q_\pm(z)$ at strong coupling is

$$
q_+(z) = \frac{i^\ell}{\sqrt{2g\pi\cos\alpha}} \left[ \frac{z^a e^{2igz}}{\Phi_a(z)} a_+(iz, g) + (-1)^\ell \frac{(g^2 z)^{-a} e^{-2igz}}{\Phi_{-a}(-z)} b_-(iz, g) \right],
$$

$$
q_-(z) = \frac{(-i)^\ell}{\sqrt{2g\pi\cos\alpha}} \left[ \frac{z^{-a} e^{-2igz}}{\Phi_{-a}(-z)} b_+(iz, g) + (-1)^\ell \frac{(g^2 z)^a e^{2igz}}{\Phi_a(z)} a_-(iz, g) \right], \tag{65}
$$

where $a = \alpha/\pi$ and the notation was introduced for the function (see (C.16))

$$
\Phi_a(z) = \frac{\Gamma\left(\frac{1}{2} - a\right)\Gamma\left(1 + \frac{iz}{2\pi}\right)}{\Gamma\left(\frac{1}{2} - a + \frac{iz}{2\pi}\right)}. \tag{66}
$$

In a close analogy with (28), the coefficient functions in (65) are given by transseries that run in powers of $1/g$ and the nonperturbative parameters (7) (see (115) below). The first few terms of their perturbative expansion look as

$$
a_+(iz, g) = 1 + \frac{a^2 - \ell^2}{4igz} + O(1/g^2),
$$

$$
b_-(iz, g) = -\frac{1}{4igz} 4^{-2a} \frac{\Gamma(1 + \ell + a)}{\Gamma(\ell - a)} + O(1/g^2). \tag{67}
$$

The remaining coefficient functions $b_+(iz, g)$ and $a_-(iz, g)$ can be obtained from $a_+(iz, g)$ and $b_-(iz, g)$, respectively, by substituting $z \to -z$ and $a \to -a$ in (67).

The techniques outlined in appendix C enable us to compute the subleading terms in (67). However, the calculations become increasingly complex at higher orders in $1/g$. To determine the subleading corrections to (67) more efficiently, we employ the differential equations (57). By substituting the ansatz (65) into (57) and matching coefficients of the exponential terms $e^{\pm 2igz}$, we can systematically derive the coefficient functions in (67) in terms of the functions $W_0(g)$ and $W_{\pm}(g)$ (see (79) below).

We would like to emphasize that the relation (65) provides asymptotic expressions for the functions $q_{\pm}(z)$ which are valid for large $g$ and fixed $z$ in the upper half-plane $\mathrm{Im}\, z > 0$. The expressions in (65) involve the factors of $z^{\pm a}$, which are multivalued functions of $z$ for arbitrary $a$. Their appearance is a manifestation of the Stokes phenomenon. To extend (65) to the region $\mathrm{Im}\, z < 0$, we take into account that for arbitrary $g$ the functions $q_{\pm}(z)$ satisfy the relation (its derivation can be found in appendix D)

$$q_{\pm}(z) = (-1)^{\ell} e^{-i\alpha} \overline{q}_{\pm}(-z), \tag{68}$$

which connects the functions across the Stokes line at $\mathrm{Im}\, z = 0$ and ensures their proper continuation into the lower half-plane. Given that $a_{\pm}(x, g)$ and $b_{\pm}(x, g)$ are real functions of $x$ (see (C.20)) and $\overline{\Phi}_a(-z) = \Phi_a(z)$, we substitute (65) into the right-hand side of (68). This leads to the result that, in the lower half-plane $\mathrm{Im}\, z < 0$, the functions $q_{\pm}(z)$ are given by the same relation as (65), with the sole modification that the factors $z^{\pm a}$ are replaced with $e^{-i\pi a}(-z)^{\pm a}$, e.g.

$$q_+(z) = \frac{i^{\ell} e^{-i\pi a}}{\sqrt{2g\pi \cos \alpha}} \left[ \frac{(-z)^a e^{2igz}}{\Phi_a(z)} a_+(iz, g) + (-1)^{\ell} \frac{(-g^2 z)^{-a} e^{-2igz}}{\Phi_{-a}(-z)} b_-(iz, g) \right]. \tag{69}$$

This relation holds at large $g$ and fixed $z$ in the lower half-plane $\mathrm{Im}\, z < 0$.

## 4.2 Transseries

Let us examine the leading behaviour of the functions (58) for $g \to \infty$. We find from (61) and (64) that

$$W_0(g) = A_{\ell}(\alpha) - \frac{1}{2}\left((\ell+1)^2 + \ell^2\right) + O(1/g), \tag{70}$$

and, therefore, this function scales as $W_0(g) = O(g^0)$. Replacing the functions $q_{\pm}(z)$ in the differential equations (57) with the ansatz (65) and matching the dependence on $g$ on both sides of (57), we find that $W_-(g) = O(g^{-2a})$ and $W_+(g) = O(g^{2a})$.

Substituting the ansatz (65) into (62) we find that the form of subleading corrections to the functions $W_0(g)$ and $W_{\pm}(g)$ is in one-to-one correspondence with the one to the coefficient functions in (65). Namely, the resulting strong coupling expansion of $W_0$ and $W_{\pm}$ takes the form of transseries

$$W_0(g) = W_0^{(0)}(g) + g^2 \sum_{n,m} \Lambda_-^{2n} \Lambda_+^{2m} W_0^{(n,m)}(g),$$

$$W_{\pm}(g) = g^{\pm 2a} \left[ W_{\pm}^{(0)}(g) + g^2 \sum_{n,m} \Lambda_-^{2n} \Lambda_+^{2m} W_{\pm}^{(n,m)}(g) \right], \tag{71}$$

where the coefficient functions are given by series in $1/g$. The terms involving $W_0^{(0)}(g)$ and $W_{\pm}^{(0)}(g)$ define a 'perturbative part' of the functions. The remaining terms proportional to

$W_0^{(n,m)}(g)$ and $W_\pm^{(n,m)}(g)$ define a 'nonperturbative' part. The additional factor of $g^2$ in front of the sum in (71) is introduced for convenience.

The subleading corrections to the perturbative functions $W_0^{(0)}(g)$ and $W_\pm^{(0)}(g)$ in (71) originate from the $O(1/g)$ corrections to the coefficient functions in (65). At the same time, the corrections to the nonperturbative functions $W_0^{(n,m)}(g)$ and $W_\pm^{(n,m)}(g)$ come from two different sources: from exponentially suppressed corrections to the coefficient functions in (65) and from integrals on the right-hand side of (62) involving rapidly oscillating functions $e^{\pm 4igz}$. These integrals have a general form

$$\text{Re} \int_0^\infty dz\, f(z) \frac{g^{\pm 2a} e^{\pm 4igz}}{[\Phi_{\pm a}(\pm z)]^2} = \frac{1}{2} g^{\pm 2a} \int_{-\infty}^\infty dz\, f(z) \frac{e^{4igz}}{[\Phi_{\pm a}(z)]^2}\,, \tag{72}$$

where $f(z)$ is an even function of $z$ and the function $\Phi_{\pm a}(z)$ is defined in (66). Closing the integration contour to the upper half-plane, the integral (72) can be evaluated by picking up residues at the poles of $1/[\Phi_{\pm a}(z)]^2$ located at $z = 2\pi i(1/2 \mp a + n)$ with $n \geq 0$. Their contribution to (72) is proportional to powers of the parameters $\Lambda_+^2$ and $\Lambda_-^2$ defined in (7).

Combining the relations (61) and (71), we find that the nonperturbative coefficient functions in the transseries (6) for $\mathcal{F}_\ell(\alpha)$ are related to those in (71) as

$$\left(\partial_g + \omega_{n,m}(g)\right)^2 \mathcal{F}_\ell^{(n,m)}(g) = W_0^{(n,m)}(g)\,. \tag{73}$$

Here the notation was introduced for

$$\omega_{n,m}(g) = n\partial_g \log \Lambda_-^2 + m\partial_g \log \Lambda_+^2 = \left(\frac{2a}{g} + 8\pi a\right)(n-m) - 4\pi(n+m)\,. \tag{74}$$

Our next objective is to calculate the coefficient functions $W_0^{(0)}(g)$ and $W_0^{(n,m)}(g)$ in (71). Once these are determined, we can use (61) and (73) to compute the coefficient functions $\mathcal{F}_\ell^{(0)}(g)$ and $\mathcal{F}_\ell^{(n,m)}(g)$.

### 4.3 Perturbative part

Let us start with computing the perturbative function $\mathcal{F}_\ell^{(0)}(g)$.

Neglecting exponentially small corrections in (71) we look for the perturbative functions $W_0^{(0)}(g)$ and $W_\pm^{(0)}(g)$ as series in $1/g$

$$W_0^{(0)}(g) = \sum_{n\geq 0} \frac{w_0^{(n)}}{g^n}\,, \qquad W_\pm^{(0)}(g) = \sum_{n\geq 0} \frac{w_\pm^{(n)}}{g^n}\,. \tag{75}$$

As a function of the angle $a = \alpha/\pi$, the expansion coefficients in (75) satisfy the relation

$$w_0^{(n)}(a) = w_0^{(n)}(-a)\,, \qquad w_-^{(n)}(a) = w_+^{(n)}(-a)\,, \tag{76}$$

where $n \geq 0$. It follows from the analogous relation (60) for the coefficient functions.

Substituting (75) into (61), we can express the perturbative part of the function $\mathcal{F}_\ell(\alpha)$ in terms of the expansion coefficients $w_0^{(n)}(a)$ as

$$\mathcal{F}_\ell(\alpha) = -2g I_0 - \frac{1}{2}\left(2w_0^{(0)} + \ell^2 + (\ell+1)^2\right)\log(8\pi g) + B_\ell(a) + \sum_{n\geq 1} \frac{w_0^{(n)} g^{-n}}{n(1+n)} + \dots\,, \tag{77}$$

where $I_0$ and $B_\ell(a)$ are the integration constants and dots denote nonperturbative terms proportional to $\Lambda_\pm^2$. One of these constants can be found by comparing the leading $O(g)$ term in (77) and (64)

$$I_0 = -\frac{\pi}{2}(1 - 4a^2). \tag{78}$$

The second constant $B_\ell(a)$ is computed in section 5. The remaining terms in the strong coupling expansion (77) can be found in two steps. First, we solve the differential equations (57) and express their solutions $q_+(z)$ and $q_-(z)$ in terms of the coefficients $w_0^{(n)}$ and $w_\pm^{(n)}$. We then substitute these solutions into (62) and obtain a closed system of equations for $w_0^{(n)}$ and $w_\pm^{(n)}$.

Let us substitute the ansatz (65) into the differential equations (57) and replace the functions $W_0(g)$ and $W_\pm(g)$ with (75). By equating the coefficients of $e^{\pm 2igz}$ on both sides of the equation, we obtain a system of differential equations for the coefficient functions $a_\pm(iz, g)$ and $b_\pm(iz, g)$. These can be solved as formal series in $1/g$. Going through the calculation we find

$$a_+(iz, g) = 1 - \frac{i(1 + 4w_0^{(0)})}{16gz} - \frac{1}{512(gz)^2}\left[64izw_0^{(1)} + \frac{16w_-^{(0)}w_+^{(0)}}{1 - 2a} + 8w_0^{(0)}(2w_0^{(0)} + 5) + 9\right]$$
$$+ O(1/g^3),$$

$$b_-(iz, g) = -\frac{iw_-^{(0)}}{4gz(1 + 2a)} + \frac{1}{64(gz)^2}\left[-\frac{8izw_-^{(1)}}{1 + a} + \frac{w_-^{(0)}(8a + 4w_0^{(0)} + 5)}{1 + 2a}\right] + O(1/g^3). \tag{79}$$

The two remaining functions $b_+(iz, g)$ and $a_-(iz, g)$ can be obtained from this relation by replacing $z \to -z$ and $a \to -a$ (see (D.10)). Note that the solutions to (57) are defined up to an overall normalization factor. In relation (79) this factor is chosen to ensure that the leading term in the expansion of $a_+(iz, g)$ coincides with the analogous term in (67).

The coefficients in (79) are given by the sum of terms $p_{n-1}(z)/(gz)^n$ involving polynomials $p_{n-1}(z)$ of degree $(n-1)$ in $z$ and depending on a finite set of the expansion coefficients $w_0^{(k)}$ and $w_\pm^{(k)}$ with $0 \le k \le n-1$. Assigning to these coefficients a fictitious $U(1)$ charge, $q(w_0) = 0$ and $q(w_\pm) = \pm 1$, we find that the functions $a_+(iz, g)$ and $b_-(iz, g)$ carry the total charge 0 and $-1$, respectively.

At the next step, we substitute the relation (65) inside the integrals in (62) and neglect terms involving rapidly oscillating functions $e^{\pm 4igz}$. According to (72), these terms generate exponentially suppressed corrections to (62) and they do not contribute to the perturbative part of the functions $W_0(g)$ and $W_\pm(g)$. In this way, we obtain

$$\partial_g W_0^{(0)}(g) = -4g\,\partial_g\,\text{Re}\left[\int_0^\infty \frac{dz}{\pi}\,(a_+(iz, g)b_+(iz, g) + a_-(iz, g)b_-(iz, g))z\frac{\partial_z\chi_a(z)}{\chi_a(z)}\right],$$

$$\partial_g W_-^{(0)}(g) = -8g\,\partial_g\,\text{Re}\left[g^{-2a}\int_0^\infty \frac{dz}{\pi}\,a_+(iz, g)b_-(iz, g)z\frac{\partial_z\chi_a(z)}{\chi_a(z)}\right], \tag{80}$$

$$\partial_g W_+^{(0)}(g) = -8g\,\partial_g\,\text{Re}\left[g^{2a}\int_0^\infty \frac{dz}{\pi}\,b_+(iz, g)a_-(iz, g)z\frac{\partial_z\chi_a(z)}{\chi_a(z)}\right],$$

where the notation was introduced for a complex valued function depending on the symbol function (17)

$$\chi_a(z) = e^{i\alpha} + \chi(z)\cos\alpha$$
$$= \frac{2\cos\alpha}{z}\Phi_a(z)\Phi_{-a}(-z) = \frac{\cosh(z/2 + i\alpha)}{\sinh(z/2)}. \tag{81}$$

Here on the second line we performed a Wiener-Hopf type decomposition of $\chi_\alpha(z)$ by factorizing it into a product of functions $\Phi_a(z)$ and $\Phi_{-a}(-z)$ analytical in the lower and upper half-planes, respectively (see (66)).

We recall that the functions $b_+(iz, g)$ and $a_-(iz, g)$ in (80) can be obtained from the functions $a_+(-iz, g)$ and $b_-(-iz, g)$, respectively, by replacing $a \to -a$. According to (79), the strong coupling expansion of $a_\pm(iz, g)$ and $b_\pm(iz, g)$ involve the coefficient functions which are polynomials in $1/z$. Substituting them into (80), we encounter integrals of the form

$$I_n(a) = 2\operatorname{Re}\left[\int_0^\infty \frac{dz}{\pi}\left(\frac{i}{z}\right)^n z\partial_z \log \chi_\alpha(z)\right],\qquad(82)$$

where the function $\chi_\alpha(z)$ is defined in (81). Going through their evaluation we find

$$I_0(a) = -\frac{\pi}{2}\left(1 - 4a^2\right),\qquad I_1(a) = 2a,\qquad(83)$$

where $a = \alpha/\pi$. The first relation coincides with (78) because the integral in (63) is proportional to $I_0$.

For $n \geq 2$ the integral (82) diverges at the origin but it can be evaluated using the analytical regularization. Namely, we insert the factor of $z^\varepsilon$ inside the integral (82), carry out the integration and take the limit $\varepsilon \to 0$ afterwards. In this way we arrive at[4]

$$I_n(a) = \frac{1}{(n-2)!\,(2\pi)^{n-1}}\Bigg[\psi^{(n-2)}\left(\frac{1}{2} - a\right) - \psi^{(n-2)}(1)$$
$$+ (-1)^n\left(\psi^{(n-2)}\left(\frac{1}{2} + a\right) - \psi^{(n-2)}(1)\right)\Bigg],\qquad(84)$$

where $\psi^{(n)}(x) = d^n\psi(x)/dx^n$ and $\psi(x) = d\log\Gamma(x)/dx$ is the Euler function. A generating function of the integrals (84) takes a simple form

$$\sum_{n\geq 1}\frac{(ix)^n}{n}I_{n+1}(a) = \log\left(\frac{\Phi_{-a}(-x)}{\Phi_a(x)}\right)$$
$$= \log\left(\frac{\Gamma\left(\frac{1}{2} + a\right)\Gamma\left(1 - \frac{ix}{2\pi}\right)\Gamma\left(\frac{1}{2} - a + \frac{ix}{2\pi}\right)}{\Gamma\left(\frac{1}{2} - a\right)\Gamma\left(1 + \frac{ix}{2\pi}\right)\Gamma\left(\frac{1}{2} + a - \frac{ix}{2\pi}\right)}\right),\qquad(85)$$

where the function $\Phi_a(x)$ is defined in (66). Note that $I_n(-a) = (-1)^n I_n(a)$ and, a consequence, $I_{2n-1}(a)$ and $I_{2n}(a)$ are odd and even functions of $a$, respectively.

Applying (82) we can expand the right-hand side of (80) into a linear combination of the integrals (82). Replacing the functions on the left-hand side of (80) with their series expansion (75), we compare the $O(1/g^n)$ terms on both sides of (80) and obtain the system of equations for the expansion coefficients in (75).

Solving these equations, we can express the perturbative functions (75) in terms of the integrals (84), e.g.

$$W_0^{(0)}(g) = w_0^{(0)} - \frac{4w_0^{(0)} + 1}{8g}I_2 + \frac{3}{64g^2}\left[(4w_0^{(0)} + 1)I_2^2 - \frac{8aw_-^{(0)}w_+^{(0)}}{1 - 4a^2}I_3\right] + O(1/g^3),\qquad(86)$$

where $I_n = I_n(a)$. Note that this relation depends on the coefficients $w_0^{(0)}$ and $w_\pm^{(0)}$. These coefficients are not captured by the differential equations (80) and have to be determined

---

[4]The function $I_{2n}(a)$ has previously appeared in the study of quiver $\mathcal{N} = 2$ super Yang-Mills theories, see (3.27) in [54]. The underlying reason for this is that $\chi_\alpha(x)\chi_{-\alpha}(x) = 1 - s_\alpha\chi_{\text{loc}}(x)$ where $s_\alpha = \cos^2\alpha$ and $\chi_{\text{loc}}(x) = -1/\sinh^2(x/2)$ is the symbol function fixed by the localization.

independently. The expression on the right-hand side of (86) has the following interesting property. Assigning a transcendental weight of $(n-1)$ to the functions $I_n(a)$, the coefficients of $1/g^n$ in (86) are even functions of $a$ of a homogenous weight $n$.

The coefficients $w_0^{(0)}$ and $w_\pm^{(0)}$ define the leading $O(1/g)$ corrections to the functions (79). They can be computed by matching (79) to the analogous relation (67) which was obtained using a technique described in appendix C. This leads to

$$
\begin{aligned}
w_0^{(0)} &= a^2 - \ell^2 - \frac{1}{4}, \\
w_-^{(0)} &= -2^{-4a}(1+2a)\frac{\Gamma(1+\ell+a)}{\Gamma(\ell-a)}, \\
w_+^{(0)} &= -2^{4a}(1-2a)\frac{\Gamma(1+\ell-a)}{\Gamma(\ell+a)}.
\end{aligned}
\tag{87}
$$

Note that $w_-^{(0)}w_+^{(0)} = (1-4a^2)(\ell^2-a^2)$.

Replacing $w_0^{(0)}$ in (77) with its expression (87), we determine the coefficient of $O(\log g)$ term in (64) as

$$
A_\ell(\alpha) = \frac{1}{4} + \ell + a^2.
\tag{88}
$$

Nonzero value of this coefficient is a manifestation of the Fisher-Hartwig singularity of the matrix symbol (37).

## 4.4 Results

Combining together the relations (86) and (87), we obtain the strong coupling expansion of the function $W_0^{(0)}(g)$ and, then, apply (61) to compute the function $\mathcal{F}_\ell(\alpha)$. In this way, we reproduce the first three terms on the right-hand side of (6) and arrive at the following result for the perturbative function

$$
\mathcal{F}_\ell^{(0)}(\alpha) = -\frac{(a^2-\ell^2)}{4g}f_1 + \frac{(a^2-\ell^2)}{32g^2}f_2 - \frac{(a^2-\ell^2)}{192g^3}f_3 + \frac{(a^2-\ell^2)}{1024g^4}f_4 - \frac{(a^2-\ell^2)}{5120g^5}f_5 + O(1/g^6),
\tag{89}
$$

where $f_k = f_k(a)$ are multilinear combinations of the functions $I_n = I_n(a)$ defined in (84)

$$
\begin{aligned}
f_1 &= I_2, \\
f_2 &= I_2^2 + 2aI_3, \\
f_3 &= I_2^3 + 6aI_3I_2 + I_4(5a^2-\ell^2+1), \\
f_4 &= I_2^4 + 12aI_3I_2^2 + 4I_4I_2(5a^2-\ell^2+1) + I_3^2(9a^2-\ell^2+1) + 2aI_5(7a^2-3\ell^2+5), \\
f_5 &= I_2^5 + 20aI_3I_2^3 + 10I_4I_2^2(5a^2-\ell^2+1) + 5I_2I_3^2(9a^2-\ell^2+1) + 10aI_2I_5(7a^2-3\ell^2+5) \\
&\quad + 10aI_3I_4(6a^2-2\ell^2+3) + 2I_6(21a^4-14a^2\ell^2+\ell^4+35a^2-5\ell^2+4).
\end{aligned}
\tag{90}
$$

The expressions for the higher order corrections to (89) become rather lengthy and we do not present them here to save space.

We verified that for $a = 0$ and $\ell = 0, 1$ the relations (89) and (90) correctly reproduce the perturbative part of the strong coupling expansion of the exact expressions (22).

The perturbative function (89) has the following general form

$$
\mathcal{F}_\ell^{(0)}(\alpha) = \sum_{n,\boldsymbol{m}} g^{-n}(a^2-\ell^2)c_{\boldsymbol{m}}(a,\ell^2)I_2^{m_2}(a)I_3^{m_3}(a)\dots I_{n+1}^{m_{n+1}}(a),
\tag{91}
$$

where the sum goes over nonnegative integers $\boldsymbol{m} = (m_2, m_3, \ldots, m_{n+1})$ satisfying the condition $m_2 + 2m_3 + \cdots + nm_{n+1} = n$.

We recall that the function $I_p(a)$ defined in (84) carries the weight $p - 1$ and has a parity $(-1)^p$ under the tranformation $a \to -a$. In each term in the sum (91), the product of the $I$−functions has the total weight $n$ and the parity $(-1)^{m_3+m_5+\cdots}$. Since $\mathcal{F}_\ell^{(0)}(\alpha)$ is an even function of $a$, the polynomial coefficients $c_{\boldsymbol{m}}(a, \ell^2)$ must have the same parity,

$$c_{\boldsymbol{m}}(-a, \ell^2) = (-1)^{m_3+m_5+\cdots} c_{\boldsymbol{m}}(a, \ell^2). \tag{92}$$

This explains why different terms in (90) contain the factors of $a$.

The relation (91) was derived for $0 \le a \le 1/2$ and nonnegative integer $\ell$. Since the coefficients $c_{\boldsymbol{m}}(a, \ell^2)$ in (91) are polynomial in $\ell^2$, they can be analytically continued to arbitrary $\ell$. In particular, for $a = \ell$ the perturbative function (91) vanishes to all orders in $1/g$. This suggests that, in a close analogy with (22), the function $\mathcal{F}_{\ell=a}$ defined in (6) should admit a closed form representation.

The relations (90) involve powers of $I_2$. Furthermore, their dependence on $I_2$ exhibits a remarkable regularity suggesting that the terms in (89) containing this function can be re-summed to all orders in $1/g$. Indeed, by redefining the coupling constant as

$$g' = g + \frac{1}{4} I_2(a), \tag{93}$$

and expanding (89) at large $g'$ we find that the dependence on $I_2$ disappears. The resulting expression of the perturbative function $\mathcal{F}_\ell^{(0)}(\alpha)$ looks as

$$
\begin{aligned}
\mathcal{F}_\ell^{(0)}(\alpha) = (a^2 - \ell^2)\Bigg[ &-\log(g'/g) + \frac{aI_3}{16g'^2} + \frac{I_4\left(-5a^2 + \ell^2 - 1\right)}{192g'^3} \\
&+ \frac{I_3^2\left(9a^2 - \ell^2 + 1\right) + 2aI_5\left(7a^2 - 3\ell^2 + 5\right)}{1024g'^4} \\
&+ \frac{-5aI_3I_4\left(6a^2 - 2\ell^2 + 3\right) - I_6\left(21a^4 - 14a^2\ell^2 + \ell^4 + 35a^2 - 5\ell^2 + 4\right)}{2560g'^5} \\
&+ O(1/g'^6)\Bigg].
\end{aligned}
\tag{94}
$$

Expanding this expression in a power series in $1/g$ results in terms involving powers of $I_2$.

A finite renormalization of the coupling constant (93) is a universal feature of observables in strongly coupled four-dimensional superconformal Yang-Mills theories which admit a dual semiclassical holographic description. The exact form of the shift in $g'$ depends on the specific observable under consideration [4, 16, 22].

The relations (89) and (94) define the perturbative part of the strong coupling expansion (6). Before addressing nonperturbative corrections, it is important to understand the properties of the series (89) and (94). We show below that, for $a \neq \ell$, the coefficients in these series grow factorially at high orders, rendering them divergent and requiring regularization. By applying resurgence techniques to these divergent series, we can extract information about the exponentially suppressed, nonperturbative corrections.

## 5 Double scaling limit

A close examination shows that the strong coupling expansion (89) and (94) is well-defined for $0 \le a < \frac{1}{2}$, away from the end-point $a = \frac{1}{2}$. For $a \to 1/2$ we find from (84) that the

functions $I_n(a)$ develop poles[5]

$$I_n(\alpha) = \frac{1}{(-2\delta)^{n-1}} + O(\delta), \tag{95}$$

where $\alpha = \pi/2 - \delta$. As a result, in the end-point region, the strong coupling expansion (89) and (94) is given by the sum of singular terms of the form $1/(g\delta)^n$.

The appearance of these singularities is an artefact of the strong coupling expansion. For finite coupling constant, the operator (37) vanishes for $\alpha \to \pi/2$ and, as a consequence, the function $\mathcal{F}_\ell(\alpha)$ defined in (43) vanishes as well. In this section, we compute this function in the limit when $a \to 1/2$ and $g \to \infty$ simultaneously.

Since the strong coupling expansion (89) runs in powers of $1/(g\delta)$, it is suggestive to consider the double scaling limit

$$g \to \infty, \qquad \alpha = \frac{\pi}{2} - \delta, \qquad \xi = 4g\delta = \text{fixed}. \tag{96}$$

We apply (89) and (95) to find that the perturbative function $\mathcal{F}_\ell^{(0)}(\alpha)$ is given in this limit by a series in $1/\xi$

$$\begin{aligned}
\mathcal{F}_\ell^{(0)} = &-\frac{4\ell^2 - 1}{8\xi} - \frac{4\ell^2 - 1}{16\xi^2} + \frac{(4\ell^2 - 1)(4\ell^2 - 25)}{384\xi^3} \\
&+ \frac{(4\ell^2 - 1)(4\ell^2 - 13)}{128\xi^4} + O(1/\xi^5) + O(1/g).
\end{aligned} \tag{97}$$

The expansion coefficients are polynomials in $\ell^2$ which vanish for $\ell^2 = \frac{1}{4}$. We demonstrate below that it is possible to sum not only the series (97) to all orders in $1/\xi$ but also to obtain a closed-form expression for the entire function (6) in the double scaling limit (96).

## 5.1 Leading behaviour

In the double scaling limit (96), the operator $\mathcal{X}$ defined in (37) simplifies significantly. In this limit the matrix $U$ can be replaced by its leading behaviour at small $\delta$. Additionally, a close examination shows that the asymptotic behaviour (95) arises from integration in (82) over small $z = O(\delta)$. This suggests that, calculating the determinant (43) in the double scaling limit (96), we can replace the symbol function (17) by its behaviour around the origin

$$\chi(x) \to \frac{2}{x}. \tag{98}$$

This leads to the following simplification of the operator in (43)

$$\langle x | \mathcal{H}_\ell \mathcal{X} | y \rangle \to \xi \begin{bmatrix} 0 & \frac{1}{\sqrt{y}} K_\ell(x, y) \\ -\frac{1}{\sqrt{y}} K_{\ell+1}(x, y) & 0 \end{bmatrix} + O(1/g), \tag{99}$$

where the kernel $K_\ell(x, y)$ is defined in (34).

To find the Fredholm determinant of the integral operator (99), it is sufficient to find its eigenvalues. Denoting the eigenfunctions of (99) as $(\psi_e(x), \psi_o(x))$, we examine the spectral problem

$$\begin{aligned}
\int_0^\infty \frac{dy}{\sqrt{y}} K_\ell(x, y) \psi_o(y) &= i\lambda \psi_e(x), \\
\int_0^\infty \frac{dy}{\sqrt{y}} K_{\ell+1}(x, y) \psi_e(y) &= -i\lambda \psi_o(x).
\end{aligned} \tag{100}$$

---

[5]For even $n$ the correction to (95) scales as $O(\delta^2)$.

The resulting expression for the Fredholm determinant is given by the product over the eigen-values

$$\det(1 + \mathcal{H}_\ell \mathcal{X}) = \prod_a (1 + i\xi\lambda_a) + O(1/g). \tag{101}$$

The system of equations (100) can be solved by using the properties of the Bessel kernel (34) summarized in appendix A. Let us denote by $\mu_a$ (with $a = 1, 2, \dots$) the positive zeros of the Bessel function, $J_\ell(\mu_a) = 0$ and $\mu_a > 0$. Replacing $z = \mu_a^2$ in (A.6), we observe that the resulting relations coincide with (100) upon identification

$$\psi_e(y) = c_e K_\ell(y, \mu_a^2), \qquad \psi_o(y) = c_o K_{\ell+1}(y, \mu_a^2), \qquad \lambda_a^2 = 1/\mu_a^2, \tag{102}$$

where the normalization factors satisfy $c_o/c_e = i\lambda_a\mu_a$.

Replacing $\lambda_a = \pm 1/\mu_a$ in (101), we can express the determinant as the product $\prod_a \left(1 + \xi^2/\mu_a^2\right)$ over positive zeros of the Bessel function $J_\ell(\mu_a) = 0$. This product can be computed in terms of a modified Bessel function of the first kind[6]

$$e^{\mathcal{F}_\ell} = \Gamma(\ell+1)(\xi/2)^{-\ell} I_\ell(\xi) + O(1/g). \tag{103}$$

We would like to emphasize that this relation takes into account both perturbative and non-perturbative corrections to the determinant (43) in the double scaling limit (96).[7]

The above analysis can be extended to compute the subleading $O(1/g)$ correction to (103), see (E.13) in appendix E.

## 5.2 Asymptotic expansion

To establish the relation between the exact formula (103) and the strong coupling expansion (6), we examine the asymptotic expansion of (103) at large $\xi$

$$e^{\mathcal{F}_\ell} = \frac{\Gamma(\ell+1)}{2\sqrt{\pi}}(\xi/2)^{-\ell-\frac{1}{2}} e^\xi \left[z(\xi) - i(-1)^\ell e^{-2\xi} z(-\xi)\right] + O(1/g), \tag{104}$$

where $z(\xi)$ is given by a series in $1/\xi$ with factorially growing coefficients

$$z(\xi) = 1 - \frac{(4\ell^2 - 1)}{8\xi} + \frac{(4\ell^2 - 9)(4\ell^2 - 1)}{128\xi^2} - \frac{(4\ell^2 - 25)(4\ell^2 - 9)(4\ell^2 - 1)}{3072\xi^3} + \dots \tag{105}$$

Taking logarithm on both sides of (104) we obtain

$$\begin{aligned}
\mathcal{F}_\ell = {}&\xi - \left(\ell + \frac{1}{2}\right)\log\xi + \log\left(2^{\ell-1/2}\pi^{-1/2}\Gamma(\ell+1)\right) \\
&+ \left(-\frac{4\ell^2-1}{8\xi} - \frac{4\ell^2-1}{16\xi^2} + \frac{(4\ell^2-1)(4\ell^2-25)}{384\xi^3} + \frac{(4\ell^2-1)(4\ell^2-13)}{128\xi^4} + \dots\right) \\
&+ (-1)^\ell i e^{-2\xi}\left(1 + \frac{4\ell^2-1}{4\xi} + \frac{(4\ell^2-1)^2}{32\xi^2} + \frac{(4\ell^2-1)\left(16\ell^4-16\ell^2+51\right)}{384\xi^3} + \dots\right) \\
&+ e^{-4\xi}\left(\frac{1}{2} + \frac{4\ell^2-1}{4\xi} + \frac{(4\ell^2-1)^2}{16\xi^2} + \frac{(4\ell^2-1)(32\ell^4-20\ell^2+27)}{192\xi^3} + \dots\right) \\
&+ O(e^{-6\xi}) + O(1/g).
\end{aligned} \tag{106}$$

---

[6]We thank Benjamin Basso and Gerald Dunne for informing us of the independent derivation of relation (103) in [55].

[7]It is interesting to note that the relation (103) emerged in the study of superconformal $\mathcal{N} = 2$ long circular quiver theories [54].

The expression on the second line of (106) is given by $\log z(\xi)$ and it coincides with the perturbative function (97). The remaining terms in (106) provide the nonperturbative corrections to (28). In distinction to (104), they contain an arbitrary power of $e^{-2\xi}$.

Let us compare (106) with the strong coupling expansion (6). We verify that, in the limit (96), the $O(\xi)$ and $O(\log \xi)$ terms in (106) coincide, respectively, with the $O(g)$ and $O(\log g)$ terms in (6). Furthermore, comparing the constant $O(\xi^0)$ and $O(g^0)$ terms in the two relations we obtain

$$B_\ell(a) = \left(\ell + \frac{1}{2}\right)\log(1/\delta) + \log\left((4\pi)^\ell \Gamma(\ell + 1)\right) + O(\delta), \tag{107}$$

where $a = 1/2 - \delta/\pi$. This relation defines the leading asymptotic behaviour of the Widom-Tracy constant for $a \to 1/2$.

The nonperturbative corrections to (6) depend on the exponentially small parameters (7). In the double scaling limit (96), we have $\Lambda_-^2 \sim e^{-2\xi}$ and $\Lambda_+^2 \to 0$. Thus, the terms in (106) proportional to $e^{-2\xi n}$ should be compared with the analogous terms in (6) proportional to $\Lambda_-^{2n}$. This leads to the following result for the leading behaviour of the coefficient functions $\mathcal{F}_\ell^{(n,0)}$ for $a \to 1/2$

$$\Lambda_-^2 \mathcal{F}^{(1,0)}(g) = -i(-1)^\ell e^{-2\xi} z(-\xi)/z(\xi),$$

$$\mathcal{F}^{(n,0)}(g) = \frac{(-1)^{n-1}}{n}\left[\mathcal{F}^{(1,0)}(g)\right]^n. \tag{108}$$

We show below that the second relation holds for arbitrary $a$.

It follows from the relation (104) that the nonperturbative correction to $Z_\ell = e^{\mathcal{F}_\ell}$ are much simpler than those to (106). Indeed, comparing (104) with (8) we find that all nonperturbative coefficient functions in (8) except $Z^{(1,0)}$ vanish in the double scaling limit (96).

## 5.3 Widom-Dyson constant

As mentioned above, the constant term $B_\ell(a)$ in (6) cannot be determined by the method of differential equations. In this subsection, we compute $B_\ell(a)$ by exploiting its behaviour in the two different limits, $a \to 0$ and $a \to 1/2$. The additional condition for $B_\ell(a)$ follows from the relation (18). Being combined with (6), this relation implies that $B_\ell(a)$ is an even function of $a$.

We recall that for $a = 0$, the function $\mathcal{F}_\ell(0)$ equals the sum (20) of functions defined in (21). For $\ell = 0$ and $\ell = 1$, these functions are given by (22) for any coupling constant. Matching their expansion at large $g$ to (6), we get

$$B_{\ell=0}(0) = 0, \qquad B_{\ell=1}(0) = \log 8. \tag{109}$$

For arbitrary $\ell$, the Widom-Dyson constant $B_\ell(0)$ is given by the sum of the analogous constants coming from the two functions on the right-hand side of (20). Using the expressions for these constants obtained in [17, 22], we find

$$e^{B_\ell(0)} = (4\pi)^\ell \Gamma(\ell + 1)\frac{G^2\left(\frac{3}{2}\right)G^2(\ell + 1)}{\sqrt{\pi}G\left(\ell + \frac{1}{2}\right)G\left(\ell + \frac{3}{2}\right)}, \tag{110}$$

where $G(x)$ is the Barnes function.

For $a \to 1/2$ the leading behaviour of $B_\ell(a)$ is given by (107). This relation was obtained by first computing the function $\mathcal{F}_\ell$ in the double scaling limit (103) and matching its expansion to

a general expression (6). The expansion of $B_\ell(a)$ around $a = 1/2$ can be derived by computing the subleading corrections to (103) suppressed by powers of $1/g$. This leads to

$$e^{B_\ell(a)} = \frac{(4\pi)^\ell \Gamma(\ell+1)}{\delta^{\ell+1/2}} \left[ 1 + \frac{\delta}{\pi}\left( \log\left(\frac{\delta}{\pi}\right) - \psi\left(\ell+\frac{1}{2}\right)\right) + O(\delta^2) \right], \qquad (111)$$

where $\delta = \pi/2 - \alpha$ and $a = \alpha/\pi$. The calculation of the $O(\delta)$ correction to the expression inside the brackets can be found in appendix E.

Thus, calculating the Widom-Dyson constant requires finding an even function $B_\ell(a)$ that satisfies (110) for $a = 0$ and has a prescribed behaviour (111) around $a = 1/2$. With some guesswork we arrived at the following relation

$$e^{B_\ell(a)} = (4\pi)^\ell \Gamma(\ell+1) \left[ \frac{\Gamma\left(\frac{1}{2}+a\right)}{\Gamma(\frac{1}{2})} \right]^{\ell-a} \left[ \frac{\Gamma\left(\frac{1}{2}-a\right)}{\Gamma(\frac{1}{2})} \right]^{\ell+a}$$
$$\times \frac{G\left(\frac{3}{2}-a\right) G\left(\frac{3}{2}+a\right) G(\ell-a+1) G(\ell+a+1)}{\sqrt{\pi} G\left(\ell+\frac{1}{2}\right) G\left(\ell+\frac{3}{2}\right)}. \qquad (112)$$

This relation can be tested by computing numerical values of $B_\ell(a)$ for various values of $a$ and $\ell$. To this end, we truncated the semi-infinite matrix in (3) to a sufficiently large size $n, m \le N_{\max}$ with $N_{\max} \sim 10^2$ and computed numerically the determinant (3) for various values of the coupling $g$ and the parameters $a$ and $\ell$. Comparing numerical values of the function $\mathcal{F}_\ell$ for $g = O(1)$ with the strong coupling expansion (6), we extracted the values of the constant $B_\ell(a)$ and observed an agreement with (112).

## 6 Nonperturbative corrections

In the previous sections, we derived the first four terms of the strong coupling expansion (6), including the perturbative function (89). We now extend this analysis to determine the nonperturbative coefficient functions $\mathcal{F}_\ell^{(n,m)}(\alpha, g)$, which multiply powers of nonperturbative parameters $\Lambda_-^{2n} \Lambda_+^{2m}$ in (6).

To calculate nonperturbative corrections to (28), we employ the method outlined in [28, 29]. We begin by solving the differential equations (57). We argue in appendix C that their solutions $q_\pm(z)$ must be entire functions of $z$. However, we show below that, for the functions $W_0(g)$ and $W_\pm(g)$ given by the transseries (71) with arbitrary coefficient functions, the solutions to (57) exhibit spurious poles at finite $z = z_\star$ (to be specified below). The requirement for the function $q_+(z)$ to be free of these poles can be expressed as

$$\lim_{z \to z_\star} (z - z_\star) q_+(z) = 0. \qquad (113)$$

Since the function $q_-(z)$ can be derived from $q_+(z)$ by replacing $z \to -z$ and $a \to -a$ (see (D.8)), this relation ensures that $q_-(z)$ is also free of spurious poles.

Replacing the function $q_+(z)$ in (113) with its asymptotic expression (65) and (69), we find that the quantization condition (114) assumes different form depending on whether $z_\star$ lies in the upper or lower half-plane

$$\lim_{z \to z_\star} (z - z_\star) \left[ b_-(iz, g) + (-1)^\ell (gz)^{2a} e^{4igz} \frac{\Phi_{-a}(-z)}{\Phi_a(z)} a_+(iz, g) \right] = 0, \quad \operatorname{Im} z_\star > 0,$$

$$\lim_{z \to z_\star} (z - z_\star) \left[ a_+(iz, g) + (-1)^\ell (-gz)^{-2a} e^{-4igz} \frac{\Phi_a(z)}{\Phi_{-a}(-z)} b_-(iz, g) \right] = 0, \quad \operatorname{Im} z_\star < 0. \quad (114)$$

Here we took into account that the function (66) is analytical in the lower half-plane.

The poles on the left-hand side of (114) originate from both the coefficient functions, $a_\pm(iz, g)$ and $b_\pm(iz, g)$, and the zeros of the $\Phi$−functions. The former functions depend in a nontrival way on the nonperturbative functions in (71). We demonstrate below that the relation (114) uniquely determines $W_0^{(n,m)}(g)$ and $W_\pm^{(n,m)}(g)$ in terms of the perturbative functions $W_0^{(0)}(g)$ and $W_\pm^{(0)}(g)$. Combined with (73), this enables us to compute the nonperturbative functions $\mathcal{F}_\ell^{(n,m)}$.

The solutions to the differential equations (57) take a general form (65). We look for the coefficients in (65) as transseries

$$
\begin{aligned}
a_+(iz, g) = {} & a_+^{(0)}(iz, g) + \Lambda_-^2 a_+^{(1,0)}(iz, g) + \Lambda_+^2 a_+^{(0,1)}(iz, g) \\
& + \Lambda_-^4 a_+^{(2,0)}(iz, g) + \Lambda_-^2 \Lambda_+^2 a_+^{(1,1)}(iz, g) + \Lambda_+^4 a_+^{(0,2)}(iz, g) + \dots, \\
b_-(iz, g) = {} & b_-^{(0)}(iz, g) + \Lambda_-^2 b_-^{(1,0)}(iz, g) + \Lambda_+^2 b_-^{(0,1)}(iz, g) \\
& + \Lambda_-^4 b_-^{(2,0)}(iz, g) + \Lambda_-^2 \Lambda_+^2 b_-^{(1,1)}(iz, g) + \Lambda_+^4 b_-^{(0,2)}(iz, g) + \dots,
\end{aligned}
\tag{115}
$$

where $\Lambda_\pm^2$ are defined in (7). The perturbative functions in this relation, $a_+^{(0)}(ix, g)$ and $b_-^{(0)}(ix, g)$, are given by (79) and (87). We recall that the two remaining functions $b_+(iz, g)$ and $a_-(iz, g)$ in (65) can be obtained from (115) by replacing $z \to -z$ and $a \to -a$.

## 6.1 Linear terms

Let us start with the nonperturbative corrections to (115) linear in the parameters $\Lambda_+^2$ and $\Lambda_-^2$.

Substituting (65) and (115) into (57), we compare the coefficients in front of $e^{\pm 2igz}$ and $\Lambda_\pm^2$ on both sides of (57) to obtain differential equations for the coefficient functions $a_\pm^{(1-n,n)}(iz, g)$ and $b_\pm^{(1-n,n)}(iz, g)$ (with $n = 0, 1$). They can be solved perturbatively in powers of $1/g$ leading to

$$
\begin{aligned}
a_+^{(1,0)}(iz, g) &= -\frac{i W_0^{(1,0)}(g)}{16((1-2a)\pi(z + i(1-2a)\pi))} + O(1/g), \\
a_+^{(0,1)}(iz, g) &= -\frac{i W_0^{(0,1)}(g)}{16((1+2a)\pi(z + i(1+2a)\pi))} + O(1/g), \\
b_-^{(1,0)}(iz, g) &= -\frac{i W_-^{(1,0)}(g)}{16(1-2a)\pi(z - i(1-2a)\pi)} + O(1/g), \\
b_-^{(0,1)}(iz, g) &= -\frac{i W_-^{(0,1)}(g)}{16(1+2a)\pi(z - i(1+2a)\pi)} + O(1/g),
\end{aligned}
\tag{116}
$$

where the nonperturbative functions $W_0^{(1-n,n)}(g)$ and $W_-^{(1-n,n)}(g)$ are defined in (71).

We observe that the functions (116) develop poles at $z = z_\star$ for $z_\star \in \{-i(1\pm 2a)\pi, i(1\pm 2a)\pi\}$. Let us substitute the above relations into (114) and examine the limit $z \to z_\star$.

- For $z = -i\pi(1-2a)$ and $z = i\pi(1+2a)$, the functions $\Phi_{-a}(-z)$ and $\Phi_a(z)$ take finite values and the relations (114) – (116) lead to

$$
\mathrm{res}_{z=-i\pi(1-2a)} \, a_+^{(1,0)}(iz, g) = \mathrm{res}_{z=i\pi(1+2a)} \, b_-^{(0,1)}(iz, g) = 0.
\tag{117}
$$

- For $z = -i\pi(1+2a)$ and $z = i\pi(1-2a)$ one of the functions, $\Phi_{-a}(-z)$ and $\Phi_a(z)$, respectively, vanishes and the relation (114) translates to

$$
\begin{aligned}
\mathrm{res}_{z=i\pi(1-2a)} \, b_-^{(1,0)}(iz, g) &= (-1)^\ell z^{1+2a} a_+^{(0)}(iz, g) \Big|_{z=i\pi(1-2a)}, \\
\mathrm{res}_{z=-i\pi(1+2a)} \, a_+^{(0,1)}(iz, g) &= (-1)^{\ell+1}(-z)^{1-2a} b_-^{(0)}(iz, g) \Big|_{z=-i\pi(1+2a)}.
\end{aligned}
\tag{118}
$$

Taking into account (116) and replacing the perturbative functions $a_+^{(0)}(iz, g)$ and $b_-^{(0)}(iz, g)$ by their expressions (79), we find from (117) and (118)

$$
\begin{aligned}
W_0^{(1,0)}(g) &= O(1/g), & W_0^{(0,1)}(g) &= O(1/g), \\
W_-^{(1,0)}(g) &= 16(-1)^\ell (i\pi(1-2a))^{2+2a}, & W_-^{(0,1)} &= O(1/g).
\end{aligned}
\tag{119}
$$

It is straightforward to compute the subleading $O(1/g)$ corrections to (116) by using the differential equations (57). These corrections involve high order poles which are located at the same values of $z$ as in (116). The relation (114) implies that the residues at these poles have to vanish while the residues at simple poles have to satisfy (117) and (118). Solving these relations, we can determine the subleading corrections to (119). Going through the calculation, we find

$$
\begin{aligned}
W_0^{(1,0)}(g) &= -e^{i\pi a}(-1)^\ell \frac{(4\pi(1-2a))^{1+2a}\Gamma(\ell-a+1)}{\Gamma(\ell+a)}\left[\frac{1}{g} - \frac{\frac{1}{4}(1-2a)I_2 + \frac{a^2-\ell^2}{2\pi(1-2a)}}{g^2} + O(1/g^3)\right], \\
W_0^{(0,1)}(g) &= -e^{-i\pi a}(-1)^\ell \frac{(4\pi(1+2a))^{1-2a}\Gamma(\ell+a+1)}{\Gamma(\ell-a)}\left[\frac{1}{g} - \frac{\frac{1}{4}(1+2a)I_2 + \frac{a^2-\ell^2}{2\pi(1+2a)}}{g^2} + O(1/g^3)\right],
\end{aligned}
\tag{120}
$$

together with

$$
\begin{aligned}
W_-^{(1,0)}(g) &= -16e^{i\pi a}(-1)^\ell (\pi(1-2a))^{2+2a} \\
&\quad \times \left[1 - \frac{a^2-\ell^2}{2\pi(1-2a)g} + \frac{(a^2-\ell^2)\bigl(2\pi(1-2a)I_2 + 2(a^2-\ell^2) - 2a - 1\bigr)}{16(\pi(1-2a))^2 g^2} + O(1/g^3)\right], \\
W_-^{(0,1)}(g) &= -e^{-i\pi a}(-1)^\ell \frac{(16\pi(1+2a))^{-2a}\Gamma^2(\ell+a+1)}{\Gamma^2(\ell-a)} \\
&\quad \times \left[\frac{1}{g^2} - \frac{a^2-\ell^2 + \pi(1+2a)^2 I_2}{2\pi(1+2a)g^3} + O(1/g^4)\right],
\end{aligned}
\tag{121}
$$

where the function $I_2 = I_2(a)$ is defined in (84). Note that the strong coupling expansion of $W_-^{(1,0)}(g)$ and $W_-^{(0,1)}(g)$ starts at order $O(g^0)$ and $O(1/g^2)$, respectively.

Viewed as functions of $a$, the two functions in (120) are related through the transformation $a \to -a$. In virtue of (60), the analogous relation holds between the functions $W_+$ and $W_-$

$$
\begin{aligned}
W_0^{(1,0)}(g|a) &= W_0^{(0,1)}(g|-a), \\
W_\mp^{(1,0)}(g|a) &= W_\pm^{(0,1)}(g|-a).
\end{aligned}
\tag{122}
$$

The four expressions in (120) and (121) involve factors of $e^{i\pi a}$ and $e^{-i\pi a}$. When substituted into (71), the resulting expressions for $W_0(g)$ and $W_\pm(g)$ become complex-valued for generic $a$. This contradicts their expected properties as real-valued functions of the coupling $g$. This discrepancy stems from the formal nature of the representation (71). The strong coupling expansions of the perturbative functions $W_0^{(0)}(g)$ and $W_\pm^{(0)}(g)$ have factorially growing coefficients and suffer from Borel singularities. These singularities require regularization, introducing an imaginary part proportional to powers of the non-perturbative parameters (7). Crucially, as we show in the next section, this imaginary part precisely cancels the imaginary part arising from the non-perturbative functions in (71), ensuring that their sum remains a real function of $g$.

The coefficients in (120) exhibit poles at $a = 1/2$. In the double scaling limit (96), these poles transform into powers of $1/\xi$. As we will demonstrate below (see (139)), the resulting expression for the nonperturbative function $\mathcal{F}^{(1,0)}$ coincides with the corresponding expression in (108).

We previously observed that the perturbative function (89) vanishes at $\ell = a$. It follows from (120) and (121) that the same behavior extends to the nonperturbative functions. Specifically, for $\ell = a$, the functions $W_0^{(0,1)}(g)$ and $W_-^{(0,1)}(g)$ vanish to all orders in $1/g$, while $W_-^{(1,0)}(g)$ receives only an $O(g^0)$ correction.

## 6.2 Quadratic terms

Repeating the previous analysis, we can use the differential equations (57) to compute the coefficient functions $a_+^{(2-n,n)}(iz, g)$ and $b_-^{(2-n,n)}(iz, g)$ (for $n = 0, 1, 2$) appearing in (115) in terms of the nonperturbative functions $W_0^{(2-n,n)}(g)$ and $W_\pm^{(2-n,n)}(g)$ defined in (71). Analogously to (116), we observe that these coefficient functions develop poles in $z$.

The poles of the $a_+$−functions are located at

$$
\begin{aligned}
a_+^{(2,0)}(iz, g): \quad & z_\star = \{-i\pi(1-2a), -2i\pi(1-2a)\}, \\
a_+^{(0,2)}(iz, g): \quad & z_\star = \{-i\pi(1+2a), -2i\pi(1+2a)\}, \\
a_+^{(1,1)}(iz, g): \quad & z_\star = \{-i\pi(1+2a), -i\pi(1-2a), -2i\pi\},
\end{aligned}
\tag{123}
$$

and the poles of the $b_-$−functions at

$$
\begin{aligned}
b_-^{(2,0)}(iz, g): \quad & z_\star = \{i\pi(1-2a), 2i\pi(1-2a)\}, \\
b_-^{(0,2)}(iz, g): \quad & z_\star = \{i\pi(1+2a), 2i\pi(1+2a)\}, \\
b_-^{(1,1)}(iz, g): \quad & z_\star = \{i\pi(1+2a), i\pi(1-2a), 2i\pi\}.
\end{aligned}
\tag{124}
$$

Some of these poles are at the same values of $z$ as in (116). We verified that for these poles the relations (114) are automatically satisfied on shell of the relations (120). The quantization condition (114) becomes nontrivial for the remaining poles at $z = z_\star$ with $z_\star \in \{-2i\pi(1 \pm 2a), 2i\pi(1 \pm 2a), \pm 2i\pi\}$. We verify using (66) that for general $a$, the functions $\Phi_{-a}(-z)$ and $\Phi_a(z)$ are different from zero for $z \to z_\star$. As a result, the relation (114) simplifies to a form analogous to (117)

$$
\mathrm{res}_{z=-2i\pi(1-2a)}\, a_\pm^{(2,0)}(iz, g) = \mathrm{res}_{z=-2i\pi(1+2a)}\, a_\pm^{(0,2)}(iz, g) = \mathrm{res}_{z=-2i\pi}\, a_\pm^{(1,1)}(iz, g) = 0,
$$

$$
\mathrm{res}_{z=2i\pi(1-2a)}\, b_\pm^{(2,0)}(iz, g) = \mathrm{res}_{z=2i\pi(1+2a)}\, b_\pm^{(0,2)}(iz, g) = \mathrm{res}_{z=2i\pi}\, b_\pm^{(1,1)}(iz, g) = 0.
\tag{125}
$$

By solving these equations perturbatively, we can compute the nonperturbative functions $W_0^{(2-n,n)}(g)$ and $W_\pm^{(2-n,n)}(g)$ (for $n = 0, 1, 2$) as series in $1/g$.

The resulting expressions for the functions $W_0^{(2-n,n)}(g)$ are

$$
W_0^{(2,0)}(g) = -e^{2i\pi a} \frac{2(4\pi(1-2a))^{4a}\Gamma^2(\ell-a+1)}{\Gamma^2(\ell+a)}
\tag{126}
$$
$$
\times \left[ \frac{1}{g^2} - \frac{\pi(1-2a)^2 I_2 + 2(a^2-\ell^2)+1-2a}{2\pi g^3(1-2a)} + O(1/g^4) \right],
$$
$$
W_0^{(1,1)}(g) = -16\pi^2(1-2a)^{1+2a}(1+2a)^{1-2a}
$$
$$
\times \left[ 1 - \frac{a^2-\ell^2}{\pi g(1-4a^2)} + \frac{(a^2-\ell^2)\big(\pi\big(1-4a^2\big)I_2 + 2(a^2-\ell^2)\big)}{4\pi^2 g^2(1-4a^2)^2} + O(1/g^3) \right].
$$

The remaining function $W_0^{(0,2)}(g)$ satisfies the relation similar to (122)

$$
W_0^{(0,2)}(g|a) = W_0^{(2,0)}(g|-a).
\tag{127}
$$

The expressions (126) have many properties in common with the analogous expressions in (120). They simplify at $\ell = a$ and develop poles at $a = 1/2$.

The functions $W_0^{(2-n,n)}$ are directly used to calculate the nonperturbative functions $\mathcal{F}_\ell^{(2-n,n)}$ using (73). The functions $W_\pm^{(2-n,n)}(g)$, on the other hand, serve an auxiliary role in our analysis. They are necessary for computing $W_0^{(p-n,n)}(g)$ for higher $p \geq 3$. To save space we do not present their explicit expressions.

## 6.3 High order terms

High-order nonperturbative corrections to the coefficient functions (115) involve terms proportional to $\Lambda_-^{2n}\Lambda_+^{2m}$. The corresponding coefficient functions $a_+^{(n,m)}(iz, g)$ and $b_-^{(n,m)}(iz, g)$ are meromorphic functions of $z$. In a close analogy with (123) and (124), they inherit the poles of the same functions for smaller $n$ and $m$. In addition, they exhibit new poles located at

$$a_+^{(n,m)}(iz, g): \qquad z_\star = -i\pi(n + m - 2a(n - m)),$$

$$b_-^{(n,m)}(iz, g): \qquad z_\star = i\pi(n + m - 2a(n - m)). \tag{128}$$

The quantization conditions (114) are automatically satisfied for all poles except (128).

As explained above, the form of the relation (114) depends on whether the functions $\Phi_a(z)$ and $\Phi_{-a}(-z)$ vanish as $z \to z_\star$. Given (66), this occurs for $a_+^{(n,m)}(iz, g)$ when $m = n + 1$ and for $b_-^{(n,m)}(iz, g)$ when $n = m + 1$. In these two cases, the quantization conditions (114) take the following form

$$\operatorname{res}_{z=i\pi(2n+1-2a)} b_-^{(n+1,n)}(iz, g) = (-1)^\ell \left[ \frac{\Gamma(n - a + \frac{1}{2})}{\Gamma(\frac{1}{2} - a)\Gamma(n+1)} \right]^2 z^{1+2a} a_+^{(0)}(iz, g) \Big|_{z=i\pi(2n+1-2a)},$$

$$\operatorname{res}_{z=-i\pi(2n+1+2a)} a_+^{(n,n+1)}(iz, g) = (-1)^{\ell+1} \left[ \frac{\Gamma(n + a + \frac{1}{2})}{\Gamma(\frac{1}{2} + a)\Gamma(n+1)} \right]^2 \tag{129}$$
$$\times (-z)^{1-2a} b_-^{(0)}(iz, g) \Big|_{z=-i\pi(2n+1+2a)},$$

where the perturbative functions $a_+^{(0)}(iz, g)$ and $b_-^{(0)}(iz, g)$ given by (79).

For $n = 0$ the relations (129) coincide with (118). For generic values of $n$ and $m$, different from those in (129), the relation (114) simplifies as

$$\operatorname{res}_{z=-i\pi(n+m-2a(n-m))} a_+^{(n,m)}(iz, g) = 0, \qquad m \neq n + 1,$$

$$\operatorname{res}_{z=i\pi(n+m-2a(n-m))} b_-^{(n,m)}(iz, g) = 0, \qquad m \neq n - 1. \tag{130}$$

Solving (129) and (130) we can determine the nonperturbative functions $W_0^{(n,m)}$ and $W_\pm^{(n,m)}$ for arbitrary $n$ and $m$.

## 6.4 Summary

Substituting the obtained expressions for the functions $W_0^{(n,m)}$ into (73), we can compute the nonperturbative functions $\mathcal{F}_\ell^{(n,m)}$ for various $n$ and $m$. We present below the explicit expressions for these functions and discuss their properties.

## Functions $\mathcal{F}_\ell^{(n,0)}$ and $\mathcal{F}_\ell^{(0,n)}$

These functions are not independent. As follows from the relations (27) and (28), the functions $\mathcal{F}^{(0,n)}$ can be derived from $\mathcal{F}^{(n,0)} = \mathcal{F}^{(n,0)}(g|a)$ by a simple substitution $a \to -a$

$$\mathcal{F}^{(0,n)}(g|a) = \mathcal{F}^{(n,0)}(g|-a). \tag{131}$$

Consequently, we only present the results for $\mathcal{F}^{(n,0)}$ below.

The leading function $\mathcal{F}_\ell^{(1,0)}$ is given by

$$\mathcal{F}^{(1,0)}(g) = -e^{i\pi a}(-1)^\ell (4\pi(1-2a))^{-1+2a}\frac{\Gamma(\ell-a+1)}{\Gamma(\ell+a)}\left[\frac{1}{g} + \frac{f_1^{(1,0)}}{g^2} + \frac{f_2^{(1,0)}}{g^3} + \frac{f_3^{(1,0)}}{g^4} + O(1/g^5)\right], \tag{132}$$

where the coefficients are multi-linear combinations of the functions $I_n = I_n(a)$ defined in (84)

$$f_1^{(1,0)} = -\frac{(1-a)^2-\ell^2}{2\pi(1-2a)} - \frac{1}{4}(1-2a)I_2, \tag{133}$$

$$f_2^{(1,0)} = \frac{1}{16}\left(3(1-a)a+\ell^2-1\right)I_3 + \frac{(1-a)\left((1-a)^2-\ell^2\right)}{4\pi(1-2a)}I_2$$

$$+\frac{1}{16}(1-a)(1-2a)I_2^2 + \frac{\left((1-a)^2-\ell^2\right)\left(3-(3-a)a-\ell^2\right)}{8\pi^2(1-2a)^2},$$

$$f_3^{(1,0)} = \frac{(a-1)(2a-3)\left((1-a)^2-\ell^2\right)}{32\pi(2a-1)}I_2^2 - \frac{\left((1-a)^2-\ell^2\right)\left(a(3a-5)-\ell^2+3\right)}{32\pi(2a-1)}I_3$$

$$+\frac{(2a-3)\left((1-a)^2-\ell^2\right)\left((a-3)a-\ell^2+3\right)}{32(\pi-2\pi a)^2}I_2 + \frac{1}{192}(a-1)(2a-3)(2a-1)I_2^3$$

$$+\frac{1}{96}(2a-1)\left(5(a-1)a-3\ell^2+3\right)I_4 - \frac{1}{64}(2a-3)\left(3(a-1)a-\ell^2+1\right)I_2I_3$$

$$+\frac{\left((1-a)^2-\ell^2\right)((2-a)^2-\ell^2)\left(2(a-3)a-2\ell^2+9\right)}{96\pi^3(2a-1)^3}.$$

Having computed the functions $\mathcal{F}_\ell^{(n,0)}$ for $n = 2,3,\dots$ we found that they are related to $\mathcal{F}^{(1,0)}(g)$ by a remarkably simple relation

$$\mathcal{F}^{(n,0)}(g) = \frac{(-1)^{n-1}}{n}\left[\mathcal{F}^{(1,0)}(g)\right]^n. \tag{134}$$

This relation holds for arbitrary angle $0 \le a < 1/2$. For $a \to 1/2$ it coincides with the analogous relation in (108).

Combining together the relations (134) and (131), we find that all terms in (28) proportional to powers of either $\Lambda_-^2$ or $\Lambda_+^2$ can be summed to all orders, e.g.

$$\sum_{n\ge 1}\Lambda_-^{2n}\mathcal{F}_\ell^{(n,0)} = \log\left(1 + \Lambda_-^2\mathcal{F}^{(1,0)}\right). \tag{135}$$

Taking this relation into account, the function (28) can be rewritten as

$$e^{\Delta\mathcal{F}_\ell} = \left(1 + \Lambda_-^2\mathcal{F}^{(1,0)}\right)\left(1 + \Lambda_+^2\mathcal{F}^{(0,1)}\right)\exp\left(\mathcal{F}_\ell^{(0)} + \sum_{n,m\ge 1}\Lambda_-^{2n}\Lambda_+^{2m}\mathcal{F}^{(n,m)}\right), \tag{136}$$

where the exponent only contains mixed terms proportional to $\Lambda_-^2\Lambda_+^2$.

The expansion coefficients (133) of the nonperturbative function $\mathcal{F}^{(1,0)}$ look similar to the analogous coefficients (90) of the perturbative function $\mathcal{F}^{(0)}$. We recall that all terms in $\mathcal{F}^{(0)}$ proportional to $I_2$ can be absorbed into redefinition of the coupling constant (93). The same property holds for the nonperturbative function (135), e.g.

$$\Lambda_-^2 \mathcal{F}^{(1,0)} = -(-1)^\ell e^{i\pi a} \frac{\Gamma(\ell - a + 1)}{\Gamma(\ell + a)} e^{-4\pi g(1-2a)}$$
$$\times (4\pi g'(1-2a))^{-1+2a} \left[ 1 + \frac{f_1^{(1,0)}}{g'} + \frac{f_2^{(1,0)}}{g'^2} + \frac{f_3^{(1,0)}}{g'^3} + O(1/g'^4) \right] \Bigg|_{I_2=0}, \quad (137)$$

where $g'$ is given by (93) and the expression inside the brackets is obtained from (133) by setting $I_2$ equal to zero.

It is instructive to examine the relation (136) in the double scaling limit (96). In this limit, $\Lambda_+^2$ vanishes and the relation (136) simplifies,

$$e^{\Delta \mathcal{F}_\ell} \overset{a \to \frac{1}{2}}{=} \left( 1 + \Lambda_-^2 \mathcal{F}^{(1,0)} \right) e^{\mathcal{F}_\ell^{(0)}}. \quad (138)$$

Here the perturbative function is given by (97) and it is related to the function $z(\xi)$ defined in (105) as $e^{\mathcal{F}_\ell^{(0)}} = z(\xi)$. Furthermore, we use (137) to obtain in the limit (96)

$$\Lambda_-^2 \mathcal{F}^{(1,0)} = -i(-1)^\ell e^{-2\xi} \left[ 1 + \frac{4\ell^2 - 1}{4\xi} + \frac{(4\ell^2 - 1)^2}{32\xi^2} + \frac{(4\ell^2 - 1)(16\ell^4 - 16\ell^2 + 51)}{384\xi^3} + O(1/\xi^4) \right]. \quad (139)$$

The expression inside the brackets coincides with the large $\xi$ expansion of the ratio of functions $z(-\xi)/z(\xi)$. In this way, we verify that (139) agrees with the first relation in (108). This provides a rigorous check of the techniques described above.

## Function $\mathcal{F}_\ell^{(1,1)}$

This function is accompanied in (6) by the factor of $\Lambda_-^2 \Lambda_+^2 = e^{-8\pi g}$. Similar to (137) all terms in the strong coupling expansion of $\mathcal{F}_\ell^{(1,1)}$ proportional to $I_2(a)$ can be absorbed into redefinition of the coupling (93).

The contribution of $\mathcal{F}_\ell^{(1,1)}$ to (6) at large $g' = g + I_2(a)/4$ looks as

$$\Lambda_-^2 \Lambda_+^2 \mathcal{F}^{(1,1)}(g) = -\frac{1}{4}(1 - 2a)^{1+2a}(1 + 2a)^{1-2a} e^{-8\pi g}$$
$$\times \left[ 1 + (a^2 - \ell^2)\left( \frac{f_1^{(1,1)}}{g'} + \frac{f_2^{(1,1)}}{g'^2} + \frac{f_3^{(1,1)}}{g'^3} + \frac{f_4^{(1,1)}}{g'^4} \right) + O(1/g'^5) \right], \quad (140)$$

where the coefficient functions are independent of $I_2(a)$ and are given by

$$f_1^{(1,1)} = \frac{1}{\pi(4a^2 - 1)},$$
$$f_2^{(1,1)} = -\frac{2a^2 + 2\ell^2 - 1}{4\pi^2(4a^2 - 1)^2},$$
$$f_3^{(1,1)} = \frac{a}{8\pi(4a^2 - 1)} I_3 + \frac{20a^4 + 20a^2\ell^2 - 19a^2 + 8\ell^4 - 13\ell^2 + 5}{48\pi^3(4a^2 - 1)^3},$$
$$f_4^{(1,1)} = -\frac{(5a^2 - \ell^2 + 1)}{64\pi(4a^2 - 1)} I_4 - \frac{a(5a^2 + 3\ell^2 - 2)}{32\pi^2(4a^2 - 1)^2} I_3$$
$$- \frac{22a^6 + 18a^4\ell^2 - 26a^4 + 6a^2\ell^4 - 15a^2\ell^2 + 13a^2 + 2\ell^6 - 7\ell^4 + 8\ell^2 - 3}{48\pi^4(4a^2 - 1)^4}. \quad (141)$$

Analogous to the behaviour observed in (89), the subleading terms in (140) are proportional to $(a^2 - \ell^2)$. This hints at the absence of the corrections to (140) when $a$ and $\ell$ coincide.

Similar to (133), the coefficients (141) have poles at $a = 1/2$. An important difference to (139) is that, as explained above, the expression (140) vanishes in the double scaling limit (96) and does not contribute to (104).

It is interesting to compare the properties of the perturbative coefficients (90) with their nonperturbative counterparts (133) and (141). Recalling the parity of the functions (84), $I_n(-a) = (-1)^n I_n(a)$, we observe that all coefficients are even functions of $a$.

Furthermore, the perturbative coefficients (90) exhibit homogeneous weight $w(f_n) = n$ under the weight assignment $w(I_n) = n - 1$. In contrast, the nonperturbative coefficients (133) and (141) possess a more intricate structure. For instance, the coefficient $f_1^{(1,0)}$ in (133) is a linear combination of $I_2$ and a term proportional to $1/\pi$. Assigning the weight $w(1/\pi) = 1$, we find that all terms in (133) and (141) share the same homogeneous weight $w(f_n^{(1,0)}) = w(f_n^{(1,1)}) = n$.

Using the techniques described above, we can compute $\mathcal{F}_\ell^{(n,m)}$ for any $n$ and $m$. The explicit expressions for these functions for $n + m = 3$ and $n + m = 4$ are given in appendix F. We used these expressions to verify that for $a = 1/4$ the strong coupling expansion of the function $\Gamma_{\alpha=\pi/4}$ defined in (11) and (12) coincides with the known result for the cusp anomalous dimension [4].

## Functions $Z^{(n,m)}$

The method of differential equations is effective for computing the function $\mathcal{F}_\ell$. However, analyzing nonperturbative corrections becomes simpler when considering its exponential form, $Z_\ell = e^{\mathcal{F}_\ell(\alpha)}$. Indeed, the definition (3) of $Z_\ell$ as a determinant suggests its holographic interpretation as a partition function. Nonperturbative corrections to $Z_\ell$ can then be attributed to contributions from distinct saddle points.

The relation between the nonperturbative corrections to $\mathcal{F}_\ell$ and $Z_\ell$ follows from (8) and (136)

$$1 + \sum_{n,m \geq 0} \Lambda_-^{2n} \Lambda_+^{2m} Z^{(n,m)}(g) = \left(1 + \Lambda_-^2 \mathcal{F}^{(1,0)}(g)\right)\left(1 + \Lambda_+^2 \mathcal{F}^{(0,1)}(g)\right) e^{\sum_{n,m \geq 1} \Lambda_-^{2n} \Lambda_+^{2m} \mathcal{F}^{(n,m)}(g)}. \quad (142)$$

Comparing the coefficients of powers of $\Lambda_-^2$ and $\Lambda_+^2$ on both sides of this relation, we can express $Z^{(n,m)}$ in terms of the functions $\mathcal{F}^{(n',m')}$ for $n' \leq n$ and $m' \leq m$.

From terms linear in $\Lambda_-^2$ and $\Lambda_+^2$ we get

$$Z^{(1,0)}(g) = \mathcal{F}^{(1,0)}(g), \qquad Z^{(0,1)}(g) = \mathcal{F}^{(0,1)}(g). \quad (143)$$

We observe that the right-hand side of (142) does not contain $O(\Lambda_-^{2n})$ and $O(\Lambda_+^{2n})$ terms with $n \geq 2$. As a consequence, the corresponding $Z$−functions have to vanish

$$Z^{(n,0)}(g) = Z^{(0,n)}(g) = 0 \qquad (n \geq 2). \quad (144)$$

This property is a direct consequence of the relations (134) and (131).

In a similar manner, we can replace the functions $\mathcal{F}^{(n,m)}$ in (142) with their strong coupling expansions and compute the functions $Z^{(n,m)}(g)$. Surprisingly, we found that, in addition to (144), many other $Z$−functions also vanish, e.g.

$$Z^{(3+n,1)}(g) = Z^{(4+n,2)}(g) = Z^{(6+n,3)}(g) = Z^{(7+n,4)}(g) = \cdots = 0 \qquad (n \geq 1). \quad (145)$$

The same relations hold for the functions $Z^{(n,m)}$ with the indices $n$ and $m$ interchanged. We verified the relations (145) for $n \le 10$ and we expect them to hold for arbitrary $n$. Combined with (142), the relations (145) yield nonlinear constraints between the functions $\mathcal{F}^{(n,m)}$. These constraints are significantly more intricate than (145), making the $Z-$functions more suitable for analyzing the nonperturbative corrections.

The nonzero $Z$-functions can be parameterized by a pair of integers $(p,q)$ satisfying $p \ge 2$ and $q \ge 0$. They take the form $Z^{(n,n+q)}$ and $Z^{(n+q,n)}$, where the positive integer $n$ is constrained as

$$\frac{1}{2}p(p-1) \le n < \frac{1}{2}p(p+1), \qquad 0 \le q \le p. \tag{146}$$

These functions are given by series in $1/g$ with the leading term given by

$$Z^{(n,n+q)}(g) = O(1/g^{q^2}), \qquad Z^{(n+q,n)}(g) = O(1/g^{q^2}). \tag{147}$$

For $q > p \ge 2$ and $n$ satisfying (146), the functions $Z^{(n,n+q)}$ and $Z^{(n+q,n)}$ vanish.

It is interesting to compare (147) with the analogous relations for the functions $\mathcal{F}^{(n,m)}$. We find that $\mathcal{F}^{(n,n)}$ and $\mathcal{F}^{(n+1,n)}$ have the same behaviour in $1/g$ as $Z^{(n,n)}$ and $Z^{(n+1,n)}$, respectively, whereas $\mathcal{F}^{(n+2,n)} = O(1/g^2)$ is enhanced by the factor of $g^2$ relative to $Z^{(n+2,n)}(g)$. Consequently, the coefficient functions of the strong coupling expansion (8) are expected to be smaller than the analogous functions entering the expansion of $\mathcal{F}_\ell = \log Z_\ell$.

Applying the relation (145), the function $Z_\ell$ in (8) simplifies as

$$Z_\ell = Z^{(0)}\Bigg( 1 + \Lambda_-^2 Z^{(1,0)} + \Lambda_+^2 Z^{(0,1)} + \Lambda_-^2 \Lambda_+^2 Z^{(1,1)} \tag{148}$$

$$+ \Lambda_-^4 \Lambda_+^2 Z^{(2,1)} + \Lambda_-^2 \Lambda_+^4 Z^{(1,2)} + \Lambda_-^6 \Lambda_+^2 Z^{(3,1)} + \Lambda_-^2 \Lambda_+^6 Z^{(1,3)} + \Lambda_-^4 \Lambda_+^4 Z^{(2,2)} + \dots \Bigg),$$

where dots denote the subleading corrections proportional to $\Lambda_+^{2n} \Lambda_-^{2m}$ with $n + m > 4$.

According to (6), the perturbative function in (148) is given by

$$Z^{(0)} = \exp\left( \pi(1 - 4a^2)g - \left(\tfrac{1}{4} + \ell + a^2\right)\log(8\pi g) + B_\ell(a) + \mathcal{F}^{(0)}(g) \right), \tag{149}$$

where $B_\ell(a)$ and $\mathcal{F}_\ell^{(0)}$ are defined in (112) and (89), respectively.

The nonperturbative coefficient functions in (148) have different behaviour at strong coupling (147). Their contribution to (148) is given by

$$\Lambda_-^2 Z^{(1,0)} = -e^{i\pi a}(-1)^\ell (1-2a)^{2a-1} \frac{\Gamma(\ell - a + 1)}{\Gamma(\ell + a)}$$

$$\times (4\pi g')^{2a-1} e^{-4\pi g(1-2a)} \left[ 1 - \frac{(1-a)^2 - \ell^2}{2\pi g'(1-2a)} + O(1/g'^2) \right],$$

$$\Lambda_-^2 \Lambda_+^2 Z^{(1,1)} = -\frac{1}{4}(1-2a)^{1+2a}(1+2a)^{1-2a}$$

$$\times e^{-8\pi g}\left[ 1 + (a^2 - \ell^2)\left( \frac{1}{\pi g'(4a^2 - 1)} - \frac{a^2 + \ell^2 - 1}{2(\pi g')^2 (4a^2 - 1)^2} + O(1/g'^3) \right) \right],$$

$$\Lambda_-^4 \Lambda_+^2 Z^{(2,1)} = -\frac{1}{4}e^{i\pi a}(-1)^\ell (1-2a)^2 (3-2a)^{-1+2a} \frac{\Gamma(\ell - a + 1)}{\Gamma(\ell + a)}$$

$$\times (4\pi g')^{2a-1} e^{-4\pi g(3-2a)} \left[ 1 - \frac{(1-a)^2 - \ell^2}{2\pi g'(3-2a)} + O(1/g'^2) \right],$$

$$\Lambda_-^6 \Lambda_+^2 Z^{(3,1)} = -e^{2i\pi a}(1-2a)^{2a-1}(3-2a)^{2a-3}\frac{\Gamma(\ell-a+1)}{\Gamma(\ell+a)}\frac{\Gamma(\ell-a+2)}{\Gamma(\ell+a-1)}$$

$$\times (4\pi g')^{4a-4}e^{-16\pi g(1-a)}\left[1-\frac{2(1-a)((2-a)^2-\ell^2)}{\pi g'(3-2a)(1-2a)}+O(1/g'^2)\right],$$

$$\Lambda_-^4 \Lambda_+^4 Z^{(2,2)} = -\frac{1}{64}(1+2a)^{1-2a}(1-2a)^{1+2a}e^{-16\pi g}$$

$$\times \left[(1+2a)^{1+2a}(3+2a)^{1-2a}\left(1-\frac{2(a^2-\ell^2)}{\pi(1-2a)(3+2a)g'}+O(1/g'^2)\right)\right.$$

$$\left.+ (1-2a)^{1-2a}(3-2a)^{1+2a}\left(1-\frac{2(a^2-\ell^2)}{\pi(1+2a)(3-2a)g'}+O(1/g'^2)\right)\right], \quad (150)$$

where the coupling $g'$ is defined in (93). The remaining functions $Z^{(n,n+p)}$ (for $p = 1, 2$) are related to $Z^{(n+p,n)}$ by the transformation $a \to -a$. We recall that the transformation $a \to -a$ interchanges $\Lambda_-^{2n}\Lambda_+^{2m} Z^{(n,m)}$ and $\Lambda_-^{2m}\Lambda_+^{2n} Z^{(m,n)}$. Consequently, $\Lambda_-^{2n}\Lambda_+^{2n} Z^{(n,n)}$ is even in $a$.

Expressions for the coefficient functions (149) and (150), including subleading $1/g$ corrections, are provided in a Mathematica notebook attached to the submission.

## 7 Resurgence relations

In this section, we employ the relation (148) to compute the determinant (3) for various values of the parameters $a$ and $\ell$, as well as the coupling constant $g = \sqrt{\lambda}/(4\pi)$. To check these analytical results, we perform numerical computations of (3) by truncating the semi-infinite $\mathbb{K}$-matrix to a sufficiently large finite size, $N_{\max} = 200$, and evaluating its determinant.

For our purposes it is convenient to rewrite (148) as

$$Z_\ell = Z^{(0)}\left(1 + \Lambda_-^2 Z^{(1,0)} + \Lambda_+^2 Z^{(0,1)} + \Lambda_-^2\Lambda_+^2 Z^{(1,1)} + \Delta Z_{\text{npt}}\right), \quad (151)$$

where $\Delta Z_{\text{npt}}$ describes the subleading nonperturbative corrections to (148). The coefficient functions in (151) are defined in (149) and (150). These functions satisfy (147) and have the general form

$$Z^{(n,m)} = \frac{1}{g^{q^2-1}}\sum_{k\geq 0}\frac{z_k}{g^{k+1}}, \quad (152)$$

where $q = n - m$ and the expansion coefficients $z_k$ depend on $n$ and $m$. As was mentioned earlier, these coefficients grow factorially at large orders $z_k \sim k!$ and, therefore, the series requires a regularization.

To this end, we employ a lateral Borel resummation $S^\pm$ defined as

$$S^\pm[Z^{(n,m)}](g) = \frac{1}{g^{q^2-1}}\int_0^{\infty e^{\pm i\epsilon}} ds\, e^{-gs}\mathcal{B}(s), \quad (153)$$

where the Borel transform is given by

$$\mathcal{B}(s) = \sum_{k\geq 0}\frac{z_k}{k!}s^k. \quad (154)$$

The factorial grows of $z_k$ at large $k$ generates singularities of $\mathcal{B}(s)$ for positive $s$. To avoid these singularities, the integration contour in (153) is slightly shifted above ($S^+$) or below ($S^-$) the real axis. The corresponding functions $S^\pm[Z^{(n,m)}](g)$ correspond to two distinct branches

$Z^{(n,m)}(g \mp i0)$ of the function (152). The difference between these functions yields a discontinuity of $Z^{(n,m)}(g)$ for $g > 0$.

The choice of integration contour in (153) introduces an ambiguity in defining the coefficient functions in (151). We demonstrate below that this ambiguity is exponentially suppressed at large $g$ and scales with powers of $\Lambda_-^2$ and $\Lambda_+^2$. Given that the observable in (151) should be independent of the integration contour choice in (153), all ambiguities must cancel on the right-hand side of (151). We verify below this property for the first few terms in the expansion (151).

Using the technique described above, we can compute an arbitrary *finite* number $M$ of terms in the series (152). Substituting them into (154), we obtain $\mathcal{B}(s)$ as a polynomial of degree $M$. To estimate the high-order corrections to (153), we use this polynomial to construct its diagonal $[M/2, M/2]$ Padé approximant, which is then substituted into (153). Performing the integration in (153), we can compute the coefficient functions for arbitrary coupling constant and subsequently calculate the function (151). Using the independence of (151) from the choice of the integration contour in (153), we adopt the lateral Borel resummation $S^+$.

The results of the analytical calculation of $Z_\ell$ for different values of $a$ and $\ell$, along with their comparison to the numerical evaluation of the determinant (3) over a wide range of the coupling constant are presented in figure 1. We observe that the relation (151) holds for sufficiently small values of the coupling, within the radius of convergence $g_\star = 1/4$ of the weak coupling expansion.

## 7.1 Numerical checks

To investigate the impact of nonperturbative corrections in (151), we numerically compute the determinant (3) for a specific set of reference parameter values and compare it with the contributions from individual terms on the right-hand side of (151).

Setting $\ell = 0$, $a = 1/10$ and $g = 1/4$, we truncate the semi-infinite matrix in (3) to a finite size of $N_{\text{max}} = 30$ and obtain

$$Z_{\ell=0} = 1.347427760461422966. \tag{155}$$

To compute the coefficient functions on the right-hand side of (151), we took into account the first $M = 100$ terms in (152) and used them to prepare the diagonal $[50, 50]$ Padé approximant of (154). We then applied the lateral Borel resummation $S^+$ and got from (153) the following results

$$S^+[Z^{(0)}] = 1.352200416619164396 + 0.001756591700125033\, i,$$
$$\Lambda_-^2 S^+[Z^{(1,0)}] = -0.003560800304735243 - 0.001284424510108616\, i,$$
$$\Lambda_+^2 S^+[Z^{(0,1)}] = 0.000438288762713771 - 0.000010368432362404\, i,$$
$$\Lambda_-^2 \Lambda_+^2 S^+[Z^{(1,1)}] = -0.000408005421013918 + 0.000000530004795297\, i. \tag{156}$$

Unlike (155), these coefficient functions acquire a nonzero imaginary part. For $S^+[Z^{(0)}]$, this imaginary part arises from integration near the Borel singularities in (153). For the other functions, additional contributions to the imaginary part stem from the $e^{\pm i\pi a}$ factors in (150). If we had employed the lateral Borel resummation $S^-$, the resulting expressions would instead be replaced by their complex conjugates.

By substituting (156) into (151), we observe that the imaginary part of $Z_{\ell=0}$ decreases rapidly as additional terms are included on the right-hand side of (151). Moreover, by comparing (151) with the exact numerical value (155), we can extract the contribution of subleading nonperturbative correction to (151)

$$\Delta Z_{\text{npt}} = -7.12551447633506 \times 10^{-7} - 2.11424375622143 \times 10^{-7}\, i. \tag{157}$$

The dominant contribution to this relation should come from the term $\Lambda_-^4\Lambda_+^2 Z^{(2,1)}$ in (148). By retaining only the leading term within the brackets in the expression for this function in (150), we obtain:

$$\Lambda_-^4\Lambda_+^2 Z^{(2,1)} = -7.53556353767215 \times 10^{-7} - 2.44845301560420 \times 10^{-7} i. \tag{158}$$

As expected, this result closely matches the value given in (157).

We would like to emphasize that the non-perturbative corrections in (156) exhibit a natural hierarchy of magnitudes. After subtracting $S^+[Z^{(0)}]$ from $Z_{\ell=0}$, the remainder is precisely at the order of $\Lambda_-^2$. Subtracting $\Lambda_-^2 S^+[Z^{(1,0)}]$ further reduced the remainder to the order of the next correction, $\Lambda_+^2 S^+[Z^{(0,1)}]$, and so on. If any non-perturbative term had been computed incorrectly, subsequent subtractions would not have led to a further decrease, and the remainder would have stagnated at an incorrect order. The observed successive reduction of the remainders at each order confirms the accuracy of all tested non-perturbative corrections, up to $M = 100$ terms, providing strong support for our calculations.

## 7.2 Borel singularities

The observable (151) has the form of a transseries in which both perturbative, $Z^{(0)}$, and non-perturbative coefficient functions, $Z^{(n,m)}$, are given by Borel nonsummable asymptotic power series in $1/g$. The properties of these series can be analyzed using generalized Borel transformation described in appendix G. We demonstrate below that the analytical structure of the Borel transform of the perturbative function carries information on the non-perturbative corrections and the various sectors in (151) labelled by a pair of indices $(n,m)$ are interrelated by resurgence relations.

Discussing the resurgent properties of (151) it is convenient to introduce the following transseries

$$\mathcal{Z} = \mathcal{Z}^{(0,0)} + e^{i\pi a}\Lambda_-^2\,\mathcal{Z}^{(1,0)} + e^{-i\pi a}\Lambda_+^2\,\mathcal{Z}^{(0,1)} + \Lambda_-^2\Lambda_+^2\,\mathcal{Z}^{(1,1)} + \dots, \tag{159}$$

where the coefficient functions are related to those in (151) as

$$\mathcal{Z}^{(0,0)} = e^{\mathcal{F}^{(0)}(g)}, \qquad \mathcal{Z}^{(n,m)} = e^{-i\pi a(n-m)}e^{\mathcal{F}^{(0)}(g)}Z^{(n,m)}. \tag{160}$$

The function (159) differs from (151) by the factor of $Z^{(0)}e^{-\mathcal{F}^{(0)}(g)}$ (see (149)), which does not affect the resurgent properties of (151). We recall that the functions (150) take complex values due to the factors containing power of $e^{i\pi a}$. The same factors were introduced in (159) to ensure that all terms in the expansion of $\mathcal{Z}^{(n,m)}$ are real.

The functions $\mathcal{Z}^{(n,m)}$ have the same form as (152)

$$\mathcal{Z}^{(n,m)} = \frac{1}{g^{(n-m)^2-1}}\sum_{k\geq 0}\frac{f_k^{(n,m)}}{g^{k+1}}, \tag{161}$$

with the only difference that the real coefficient $f_k^{(n,m)}$ receive the additional contribution from the expansion of $e^{\mathcal{F}^{(0)}(g)} = 1 + O(1/g)$. We recall that the functions $\mathcal{Z}^{(n,m)}$ and $\mathcal{Z}^{(m,n)}$ are related to each other through the transformation $a \to -a$. The same property applies to the coefficients $f_k^{(n,m)}$.

The expansion coefficients in (161) grow factorially at large $k$ and have the following general form

$$f_k^{(n,m)} = \frac{1}{\pi}\sum_I\left(c_{0,I}\frac{\Gamma(k+\lambda_I)}{(4\pi A_I)^{k+\lambda_I}} + c_{1,I}\frac{\Gamma(k+\lambda_I-1)}{(4\pi A_I)^{k+\lambda_I-1}} + c_{2,I}\frac{\Gamma(k+\lambda_I-2)}{(4\pi A_I)^{k+\lambda_I-2}} + \dots\right), \tag{162}$$

where the factors of $\pi$ were introduced for convenience. It is parameterized by the parameters $A_I$ and $\lambda_I$, along with an infinite set of coefficients $c_{n,I}$. To avoid unnecessary clutter, we do not display their dependence on $n$ and $m$. Each term in the sum (162) generates a singularity $\mathcal{B}(s) \sim (s - 4\pi A_I)^{-\lambda_I}$ of the Borel transform (154) (see (G.6)).

Due to the presence of Borel singularities, the function (161) is not well-defined for real $g$. To define it properly, the coupling constant must be slightly shifted off the real axis. The relation (159) holds for Im $g < 0$, corresponding to the lateral Borel resummation $S^+$. If we used instead the $S^-$–resummation in (161), the coefficients in (159) had to be replaced with complex conjugated expressions,

$$
\begin{aligned}
\mathcal{Z} &= S^+[\mathcal{Z}^{(0,0)}] + e^{i\pi a}\Lambda_-^2 S^+[\mathcal{Z}^{(1,0)}] + e^{-i\pi a}\Lambda_+^2 S^+[\mathcal{Z}^{(0,1)}] + \Lambda_-^2\Lambda_+^2 S^+[\mathcal{Z}^{(1,1)}] + \dots \\
&= S^-[\mathcal{Z}^{(0,0)}] + e^{-i\pi a}\Lambda_-^2 S^-[\mathcal{Z}^{(1,0)}] + e^{i\pi a}\Lambda_+^2 S^-[\mathcal{Z}^{(0,1)}] + \Lambda_-^2\Lambda_+^2 S^-[\mathcal{Z}^{(1,1)}] + \dots
\end{aligned}
\tag{163}
$$

The function $\mathcal{Z}$ is independent of the choice of the regularization. However, the individual terms in its expansion are regularization-dependent.

The difference between the two expressions, $S^+[\mathcal{Z}^{(n,m)}]$ and $S^-[\mathcal{Z}^{(n,m)}]$, is exponentially suppressed at large $g$ (see (G.7))

$$
\Lambda_-^{2n}\Lambda_+^{2m} S^+[\mathcal{Z}^{(n,m)}] - \Lambda_-^{2n}\Lambda_+^{2m} S^-[\mathcal{Z}^{(n,m)}] \sim 2i\pi g^{\lambda_I - 1} e^{-4\pi g A_I},
\tag{164}
$$

where $n, m \geq 0$. For this ambiguity to cancel in the sum (159), the expression on the right-hand side of (164) must exhibit the same dependence on the coupling constant as the other subleading terms in (159). This requirement suggests that the parameters $A_I$ and $\lambda_I$ in (162) and (164) take the following form

$$
A_I = p(1 - 2a) + q(1 + 2a), \qquad \lambda_I = 1 + 2a(p - q) - (p - q)^2,
\tag{165}
$$

where $I = (p, q)$ is a composed index with $p \geq n$, $q \geq m$ and $p + q > n + m$.

To verify the relations (164) and (165), we investigated the analytical properties of a (generalized) Borel transform for the specific value of the angle $a = 1/25$ and $\ell = 0$. The location of the Borel singularities (164) on the real axis depends on $a$. The value of $a$ was chosen to separate the leading Borel singularities and, thus, avoid overlaps between the various nonperturbative terms. We computed numerically the first 400 coefficients in $1/g$ expansion of the functions (161) for $n + m \leq 2$ and constructed the Borel transform (154) in the form of partial sums with 400 terms. We then applied the technique described in appendix G to compute the difference of functions on the left-hand side of (164). Finally, we verify that this difference coincides with the subleading nonperturbative functions and, thus, satisfies the resurgence relations. We present below the details of the calculation for various functions in (159).

## Perturbative function $\mathcal{Z}^{(0,0)}$

We applied (89) and (160), computed the first 400 coefficients of the $1/g$ expansion of $\mathcal{Z}^{(0,0)}$ and prepared the diagonal $[200, 200]$ Padé approximant $\mathcal{B}_{\text{app}}(s)$ of the Borel transform (154). Examining its poles we found that the Borel transform of $\mathcal{Z}^{(0,0)}$ has cuts on the real axis (see figure 2). For positive and negative $s$, the closest to the origin cut starts at $s = 4\pi A$ with $A = 1 - 2a$ and $A = -2$, respectively. The former value coincides with (165) for $p = 1$ and $q = 0$.

To identify the subleading Borel singularities, we used the conformal mapping to map the whole complex Borel plane to the unit circle (see (G.10) in appendix G). This transformation effectively isolates the various branch cuts, allowing for their individual analysis. Through

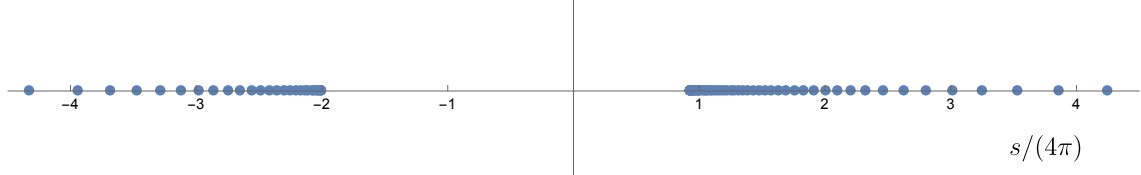

Figure 2: Poles of the diagonal Padé approximant for the Borel transform of $\mathcal{Z}^{(0,0)}$ for $\ell = 0$ and $a = 1/25$. The two accumulation points are located at $s/(4\pi) = -2$ and $s/(4\pi) = 1 - 2a$.

this approach, we determined that, within the achieved numerical accuracy, the perturbative function $\mathcal{Z}^{(0,0)}$ exhibits cuts along the positive semi-axis, originating at $s = 4\pi A^{(0,0)}$ where[8]

$$A^{(0,0)} = \{1 - 2a, 1 + 2a, 2, 3 - 2a, 3 + 2a\}. \tag{166}$$

These values have the expected form (165) and correspond to

$$I = (p, q) = \{(1, 0), (0, 1), (1, 1), (1, 2), (2, 1)\}. \tag{167}$$

Additionally, we confirmed that the associated values of the parameter $\lambda^{(0,0)}$ coincide with those given by (165).

Each of the cuts in (166) can be analyzed using the method described in appendix G. Their contribution to (164) is given by (see (G.7))

$$S^+[\mathcal{Z}^{(0,0)}](g) - S^-[\mathcal{Z}^{(0,0)}](g) = 2ig^\lambda e^{-4\pi gA} \sum_{n \geq 0} \frac{c_n}{g^{n+1}}, \tag{168}$$

where $A$, $\lambda$ and $c_n$ (with $n \geq 0$) define large order behaviour (162) of the expansion coefficients $f_k^{(0,0)}$ generated by the cut in (166). The values of $A$ and $\lambda$ were specified above. The coefficients $c_n$ can be found by comparing the asymptotic behaviour of the Padé approximant $\mathcal{B}_{\text{app}}(s)$ in the vicinity of $s = 4\pi A$ with its expected form (see (G.9)).

For the leading cut in (166) we have $A = 1 - 2a$ and $\lambda = 2a$. Its contribution to the Borel ambiguity (168) takes the form

$$S^+[\mathcal{Z}^{(0,0)}](g) - S^-[\mathcal{Z}^{(0,0)}](g) = 2i\Lambda_-^2 \sum_{n \geq 0} \frac{c_n^{(1,0)}}{g^{n+1}}, \tag{169}$$

where $\Lambda_-^2$ is defined in (7) and the superscript in $c_n^{(1,0)}$ refers to the values of the index $I$ in (166). This ambiguity must cancel in the difference of the two expressions in (163) against the contribution of the $O(\Lambda_-^2)$ term. The latter is given by $(e^{i\pi a} - e^{-i\pi a})\Lambda_-^2 \mathcal{Z}^{(1,0)}$. In this way, we obtain the relation between the perturbative function on the right-hand side of (169) and the nonperturbative function $\mathcal{Z}^{(1,0)}$ (169)

$$\sum_{n \geq 0} \frac{c_n^{(1,0)}}{g^{n+1}} = \sin(\pi a)\mathcal{Z}^{(1,0)}. \tag{170}$$

Replacing $\mathcal{Z}^{(1,0)}$ with its $1/g$ expansion (161), we find the resurgence relation between the perturbative and nonperturbative coefficients

$$c_k^{(1,0)} = \sin(\pi a)f_k^{(1,0)}. \tag{171}$$

---

[8]The function $\mathcal{Z}^{(0,0)}$ might possess additional subleading cuts other than those specified in (166), but they remain undetectable with the available 400 terms of the $1/g$ expansion of $\mathcal{Z}^{(0,0)}$.

We used the first 400 terms of the $1/g$ expansion of the perturbative function $\mathcal{Z}^{(0,0)}$ to compute the first few terms in (169). These were then combined with the leading terms of the nonperturbative function $Z^{(1,0)}$ from (150) to verify the relation (171) for $k = 0, 1, 2, \ldots$, achieving accuracies of $10^{-43}, 10^{-40}, 10^{-37}, \ldots$

For the second cut in (165) we have $A = 1 + 2a$ and $\lambda = -2a$. The contribution of this cut to (168) is

$$S^+[\mathcal{Z}^{(0,0)}](g) - S^-[\mathcal{Z}^{(0,0)}](g) = 2i\Lambda_+^2 \sum_{n \geq 0} \frac{c_n^{(0,1)}}{g^{n+1}}. \tag{172}$$

It must cancel in the difference of the two expressions in (163) against the contribution of the $O(\Lambda_+^2)$ term which is given by $(e^{-i\pi a} - e^{i\pi a})\Lambda_+^2 \mathcal{Z}^{(0,1)}$. This leads to the resurgence relation analogous to (171)

$$c_k^{(0,1)} = -\sin(\pi a) f_k^{(0,1)}. \tag{173}$$

The calculation of the perturbative coefficients $c_k^{(0,1)}$ is more involved as compared with $c_k^{(1,0)}$ because the cut at $s = 4\pi(1 + 2a)$ is hidden behind the leading cut at $s = 4\pi(1 - 2a)$. To isolate this cut, we use the conformal mapping (G.10) to map the branching points (166) to points on the unit circle. Expanding the Borel transform in the vicinity of the image of $s = 4\pi(1 + 2a)$ on the unit circle and matching this expansion to (G.9), we can compute the expansion coefficients $c_n^{(0,1)}$. In this way, we verified the resurgence relation (173) for $k = 0, 1, 2, \ldots$ with the accuracies of $10^{-23}, 10^{-19}, 10^{-18}, \ldots$

For the third cut in (166) we have $A = 2$ and $\lambda = 1$. According to (164), its contribution to (168) is proportional to $\Lambda_+^2 \Lambda_-^2 = e^{-8\pi g}$. There are two distinct sources of this contribution. The first, 'direct' source arises from the large-order behavior of the expansion coefficients (162). Similar to (169) and (172), this contribution to (168) is *pure imaginary*. The second, 'indirect' source is related to Borel singularities of the series on the right-hand side of (169) and (172). These singularities emerge from the factorial growth of the coefficients $c_n^{(1,0)}$ and $c_n^{(0,1)}$ at large $n$. Note that, due to the resurgence relations (171) and (173), the series in (169) and (172) coincide, up to a normalization factor, with the nonperturbative functions $\mathcal{Z}^{(0,1)}$ and $\mathcal{Z}^{(1,0)}$. As we show below, the Borel singularities produce a pure imaginary contribution to these functions, proportional to the product of $\Lambda_\pm^2$ and the nonperturbative function $\mathcal{Z}^{(1,1)}$. Taking this into account, we find the relations (169) and (172) acquire an additional *real* contribution, proportional to $\Lambda_+^2 \Lambda_-^2 \mathcal{Z}^{(1,1)}$.

The contribution of the cut at $A = 2$ to (168) takes the form

$$S^+[\mathcal{Z}^{(0,0)}](g) - S^-[\mathcal{Z}^{(0,0)}](g) = 2i e^{-8\pi g} \sum_{n \geq 0} \frac{c_n^{(1,1)}}{g^n}. \tag{174}$$

We expect that it is related, via a resurgence relation, to the $O(\Lambda_+^2 \Lambda_-^2)$ term in (159). Since this term is a real function of the coupling, the coefficients $c_n^{(1,1)}$ must be pure imaginary, indicating that they receive nonvanishing contribution only from the indirect source. Consequently, the resulting resurgence relations take the form

$$i \sum_{n \geq 0} \frac{c_n^{(1,1)}}{g^n} = 2\sin^2(\pi a) \mathcal{Z}^{(1,1)}, \tag{175}$$

or equivalently

$$c_k^{(1,1)} = -2i \sin^2(\pi a) f_k^{(1,1)}. \tag{176}$$

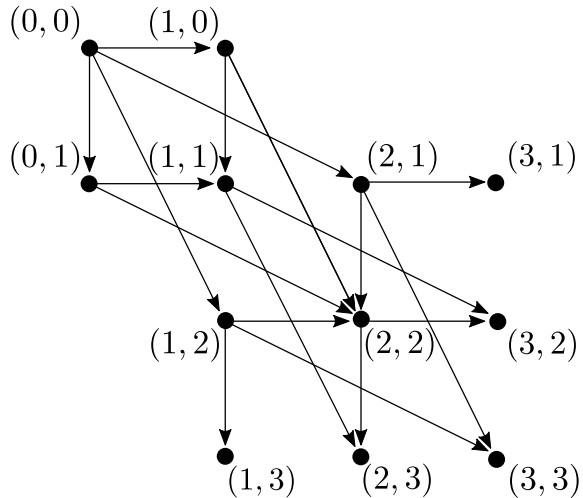

Figure 3: Diagrammatic representation of the resurgence relations. The node with indices $(n, m)$ represents the function $\mathcal{Z}^{(n,m)}$. Arrows connecting the nodes signify direct resurgence relations. The diagram continues indefinitely towards the southeast, our analysis is restricted to $0 \le n, m \le 2$.

The expressions on the right-hand side receive two contributions: one from the cut of the Borel transform of $\mathcal{Z}^{(1,0)}$ at $s = 4\pi(1 + 2a)$ and the other from the cut of $\mathcal{Z}^{(0,1)}$ at $s = 4\pi(1 - 2a)$. Both contributions are equal and proportional to $\mathcal{Z}^{(1,1)}$.

We verified the relation (176) through an explicit calculation of the coefficients $c_n^{(1,1)}$ for $n = 0, 1, 2, \ldots$ The analysis follows the same procedure used for calculating the coefficients $c_n^{(0,1)}$. Our results show that the real part of $c_n^{(1,1)}$ vanishes within the achieved numerical accuracy, while the imaginary part is in agreement with (176).

### Leading non-perturbative function $\mathcal{Z}^{(1,0)}$

As in the previous case, we computed the first 400 terms in $1/g$ expansion of the function $\mathcal{Z}^{(1,0)}$ and constructed the diagonal $[200, 200]$ Padé approximant $\mathcal{B}_{\text{app}}^{(1,0)}(s)$ of the Borel transform (154). By inspecting its poles, we found that the closest to the origin cuts are located at $s = -4\pi(1 - 2a)$ and $s = 4\pi(1 + 2a)$ for negative and positive $s$, respectively.

The cut at $s = -4\pi(1 - 2a)$ carries information about the perturbative function $\mathcal{Z}^{(0,0)}$. On the positive semi-axis, the leading cut starts at $s = 4\pi(1 + 2a)$. To identify the location of subleading cuts, we applied a conformal mapping (G.10). Surprisingly, we found the subleading cut at $s = 4\pi A$ for $A = 2(1+2a)+(1-2a)$ and no cuts at $A = 2(1+2a)$ or $A = (1+2a)+(1-2a)$. This means that the branching points of $\mathcal{Z}^{(1,0)}$ satisfy (165) for

$$A^{(1,0)} = \{1 + 2a, 3 + 2a\}, \qquad I = (p, q) = \{(0, 1), (1, 2)\}. \tag{177}$$

Hence, the function $\mathcal{Z}^{(1,0)}$ is related by the resurgence relations to the functions $\mathcal{Z}^{(1,1)}$ and $\mathcal{Z}^{(2,2)}$, but not to $\mathcal{Z}^{(1,2)}$ or $\mathcal{Z}^{(2,1)}$ (see figure 3).

The leading cut in (177) can be investigated as above. For this cut we have $A = 1 + 2a$ and $\lambda = -2a$. Expanding the Padé approximant $\mathcal{B}_{\text{app}}^{(1,0)}(s)$ around $s = 4\pi(1 + 2a)$ we can apply (G.9) and compute the corresponding coefficients $c_n^{(0,1)}$. These coefficients are related to the nonperturbative function $\mathcal{Z}^{(1,1)}$ by the resurgence relations

$$\sum_{n \ge 0} \frac{c_n^{(0,1)}}{g^n} = -\sin(\pi a)\mathcal{Z}^{(1,1)}, \tag{178}$$

which are analogous to (170). Replacing $\mathcal{Z}^{(1,1)}$ with its expansion (161) we get

$$c_k^{(0,1)} = -\sin(\pi a)f_k^{(1,1)}. \tag{179}$$

Performing numerical analysis we found that this relation is satisfied for $k = 0, 1, 2, \dots$ with the accuracies of $10^{-53}, 10^{-50}, 10^{-48}, \dots$

### Non-perturbative function $\mathcal{Z}^{(0,1)}$

Going along the same lines as before, we found that the diagonal $[200, 200]$ Padé approximant $\mathcal{B}_{\text{app}}^{(0,1)}(s)$ has the leading cuts at $s = -4\pi(1+2a)$ and $s = 4\pi(1-2a)$ for negative and positive $s$, respectively.

The cut at $s = -4\pi(1+2a)$ corresponds to $\mathcal{Z}^{(0,0)}$. Applying the optimal conformal mapping (G.10), we identified the cuts at positive $s = 4\pi A$ as

$$A^{(0,1)} = \{1 - 2a, 3 - 2a\}, \qquad I = (p, q) = \{(1, 0), (2, 1)\}. \tag{180}$$

Hence, the function $\mathcal{Z}^{(1,0)}$ is related by the resurgence relations to the functions $\mathcal{Z}^{(1,1)}$ and $\mathcal{Z}^{(2,2)}$ (see figure 3). By repeating the analysis, we found that for the functions $\mathcal{Z}^{(1,1)}$ these relations look as

$$c_k^{(1,0)} = \sin(\pi a)f_k^{(1,1)}. \tag{181}$$

We verified this relation for $k = 0, 1, 2, \dots$ with the accuracies of $10^{-63}, 10^{-60}, 10^{-58}, \dots$

### Non-perturbative function $\mathcal{Z}^{(1,1)}$

We used the diagonal $[200, 200]$ Padé approximant $\mathcal{B}_{\text{app}}^{(1,1)}(s)$ to verify that it has cuts on the negative semi-axis located at $s = 4\pi A$ and $A \in \{-2, -(1+2a), -(1-2a)\}$. These cuts establish the connections between $\mathcal{Z}^{(1,1)}$ and the functions $\mathcal{Z}^{(0,0)}$, $\mathcal{Z}^{(0,1)}$ and $\mathcal{Z}^{(1,0)}$.

On the positive semi-axis, the leading cut is located at $s = 4\pi(1 + 2a + 2(1-2a))$. As a result, $\mathcal{Z}^{(1,1)}$ is related through resurgence to the functions $\mathcal{Z}^{(2,3)}$ and $\mathcal{Z}^{(3,2)}$ but not to $\mathcal{Z}^{(1,2)}$, $\mathcal{Z}^{(2,1)}$ and $\mathcal{Z}^{(2,2)}$ (see figure 3).

It would be interesting to better understand the resurgent properties of higher order functions $\mathcal{Z}^{(n,m)}$ in the transseries (159), as it was done in [56] for the free energy problem in two dimensional integrable models. The analysis presented in this section confirms that the Borel ambiguities cancel in the sum of all terms in (159), ensuring that $\mathcal{Z}$ is a well-defined function of the coupling constant.

## Acknowledgments

We would like to thank Benjamin Basso, Gerald Dunne, Yizhuang Liu and Alessandro Testa for very interesting discussions. We are also grateful to Benjamin Basso for helpful comments on a draft of this paper. ZB and GK thank the Galileo Galilei Institute for Theoretical Physics for the hospitality and the INFN for partial support during the workshop "Resurgence and Modularity in QFT and String Theory".

**Funding information** The research was supported by the Doctoral Excellence Fellowship Programme funded by the National Research Development and Innovation Fund of the Ministry of Culture and Innovation and the Budapest University of Technology and Economics, under a grant agreement with the National Research, Development and Innovation Office (NK-FIH). It was also supported by the research Grant No. K134946 of NKFIH and by the French National Agency for Research grant "Observables"(ANR-24-CE31-7996).

# A  Derivation of the differential equations

In this appendix, we derive the differential equations (49) – (51). The derivation relies on the properties of the Bessel kernel (34), which are summarized below. Our analysis closely follows the approach outlined in [14].

**Properties of the Bessel kernel**

The Bessel kernel $K_\ell(x, y) = K_\ell(y, x)$ is defined in (34). It satisfies the following relations

$$
\begin{aligned}
\left(x\partial_x + y\partial_y + 1\right)K_\ell(x, y) &= \frac{1}{4}J_\ell(\sqrt{x})J_\ell(\sqrt{y}), \\
(x - y)K_\ell(x, y) &= J_\ell(\sqrt{x})y\partial_y J_\ell(\sqrt{y}) - x\partial_x J_\ell(\sqrt{x})J_\ell(\sqrt{y}) \\
&= \frac{1}{2}\left[\sqrt{x}J_{\ell+1}(\sqrt{x})J_\ell(\sqrt{y}) - \sqrt{y}J_{\ell+1}(\sqrt{y})J_\ell(\sqrt{x})\right] \\
&= \frac{1}{2}\left[\sqrt{y}J_\ell(\sqrt{x})J_{\ell-1}(\sqrt{y}) - \sqrt{x}J_\ell(\sqrt{y})J_{\ell-1}(\sqrt{x})\right].
\end{aligned}
\tag{A.1}
$$

The first relation follows from the integral representation (34). The second is a direct consequence of the Christoffel-Darboux formula applied to the Bessel function and its differential equation.

The Bessel function $J_\ell(x)$ has infinitely many zeros, denoted by $\mu_a$ (where $a = 1, 2, \dots$). Setting $x = \mu_a^2$ in the second relation of (A.1) yields

$$
K_\ell(\mu_a^2, y) = \frac{\mu_a J_{\ell+1}(\mu_a)J_\ell(\sqrt{y})}{2(\mu_a^2 - y)}, \qquad K_{\ell+1}(\mu_a^2, y) = \frac{J_{\ell+1}(\mu_a)\sqrt{y}J_\ell(\sqrt{y})}{2(\mu_a^2 - y)}.
\tag{A.2}
$$

In addition, for $y = \mu_b^2$ and $J_\ell(\mu_b) = 0$ we find

$$
K_\ell(\mu_a^2, \mu_b^2) = K_{\ell+1}(\mu_a^2, \mu_b^2) = \frac{1}{4}J_{\ell+1}^2(\mu_a)\delta_{ab}.
\tag{A.3}
$$

The orthogonality condition (16) implies that the Bessel kernel (34) possesses the properties of a projector

$$
\int_0^\infty dy\, K_\ell(x, y)K_\ell(y, z) = K_\ell(x, z).
\tag{A.4}
$$

Furthermore, the normalized Bessel functions (15) satisfy a recurrence equation

$$
\frac{\psi_n(x)}{\sqrt{x}} = \frac{1}{2\sqrt{n}}\left[\frac{\psi_{n+1}(x)}{\sqrt{n+1}} + \frac{\psi_{n-1}(x)}{\sqrt{n-1}}\right],
\tag{A.5}
$$

which leads to

$$
\begin{aligned}
\int \frac{dy}{\sqrt{y}}K_\ell(x, y)K_{\ell+1}(y, z) &= \frac{1}{\sqrt{z}}K_\ell(x, z) - \frac{\psi_{\ell+1}(x)\psi_\ell(z)}{2\sqrt{\ell(\ell+1)}}, \\
\int \frac{dy}{\sqrt{y}}K_{\ell+1}(x, y)K_\ell(y, z) &= \frac{1}{\sqrt{z}}K_{\ell+1}(x, z).
\end{aligned}
\tag{A.6}
$$

**Differential equations**

Let us examine a derivative of (43) with respect to the coupling constant. On the right-hand side of (43), the dependence on $g$ resides in the operator $\mathcal{X}$ defined in (37). Since the function $\chi$ in (37) depends on the ratio $\sqrt{x}/(2g)$, this operator satisfies the relation

$$g\partial_g\mathcal{X} = -2[x\partial_x,\mathcal{X}]. \tag{A.7}$$

Taking this relation into account, we get from (43)

$$\begin{aligned}
g\partial_g\mathcal{F}_\ell &= \text{Tr}\left(\frac{1}{1+\mathcal{H}_\ell\mathcal{X}}\mathcal{H}_\ell g\partial_g\mathcal{X}\right)\\
&= -2\,\text{Tr}\left(\frac{1}{1+\mathcal{H}_\ell\mathcal{X}}\mathcal{H}_\ell[x\partial_x,\mathcal{X}]\right) = 2\,\text{Tr}\left(\mathcal{X}\frac{1}{1+\mathcal{H}_\ell\mathcal{X}}[x\partial_x,\mathcal{H}_\ell]\right),
\end{aligned} \tag{A.8}$$

where in the last relation we used the cyclic property of the trace. Here the trace is taken over both matrix indices and the linear space on which $\mathcal{H}_\ell$ acts.

The integral operator $\mathcal{H}_\ell$ has a matrix kernel (42). The kernel of the commutator $[x\partial_x,\mathcal{H}_\ell]$ is $(x\partial_x + y\partial_y + 1)\mathcal{H}_\ell(x,y)$. Applying the first relation in (A.1), we find that this kernel is a diagonal matrix with nonzero entries $J_{\ell+i}(\sqrt{x})J_{\ell+i}(\sqrt{y})/4$ for $i=0,1$. Returning to operator form and using the notation (44), we find

$$[x\partial_x,\mathcal{H}_\ell] = \frac{1}{4}\begin{bmatrix} |\phi_\ell\rangle\langle\phi_\ell| & 0 \\ 0 & |\phi_{\ell+1}\rangle\langle\phi_{\ell+1}| \end{bmatrix} = \frac{1}{4}|\Phi_\ell\rangle\rangle\langle\langle\Phi_\ell|. \tag{A.9}$$

We apply this relation to get from (A.8)

$$g\partial_g\mathcal{F}_\ell = \frac{1}{2}\text{tr}\left[\langle\langle\Phi_\ell|\mathcal{X}\frac{1}{1+\mathcal{H}_\ell\mathcal{X}}|\Phi_\ell\rangle\rangle\right] = -\frac{1}{2}\text{tr}\,u, \tag{A.10}$$

where a $2\times 2$ matrix $u$ is defined in (47). This yields the equation (49).

Let us differentiate both sides of (47) with respect to the coupling constant

$$\begin{aligned}
g\partial_g u &= -\langle\langle\Phi_\ell|g\partial_g\mathcal{X}\frac{1}{1+\mathcal{H}_\ell\mathcal{X}}|\Phi_\ell\rangle\rangle + \langle\langle\Phi_\ell|\frac{1}{1+\mathcal{X}\mathcal{H}_\ell}\mathcal{X}\mathcal{H}_\ell g\partial_g\mathcal{X}\frac{1}{1+\mathcal{H}_\ell\mathcal{X}}|\Phi_\ell\rangle\rangle\\
&= -\langle\langle\Phi_\ell|\frac{1}{1+\mathcal{X}\mathcal{H}_\ell}g\partial_g\mathcal{X}\frac{1}{1+\mathcal{H}_\ell\mathcal{X}}|\Phi_\ell\rangle\rangle\\
&= 2\int_0^\infty dx\, x\partial_x\chi\left(\frac{\sqrt{x}}{2g}\right)\bar{Q}(x)UQ(x),
\end{aligned} \tag{A.11}$$

where in the last relation we applied (A.7), (37) and (45) and introduced notation for

$$\bar{Q}(x) = \langle\langle\Phi_\ell|\frac{1}{1+\mathcal{X}\mathcal{H}_\ell}|x\rangle = \left(\langle x|\frac{1}{1+\mathcal{H}_\ell^t\mathcal{X}^t}|\Phi_\ell^t\rangle\rangle\right)^t. \tag{A.12}$$

Here the superscript '$t$' indicates the transpose of a $2\times 2$ matrix. The matrices (42) and (44) are diagonal and, therefore, $\mathcal{H}_\ell^t = \mathcal{H}_\ell$ and $\Phi_\ell^t = \Phi_\ell$. At the same time, as follows from (37), $\mathcal{X}^t$ differs from $\mathcal{X}$ in that its off-diagonal elements flip the sign. As a result, the two operators are related to each other as

$$\mathcal{X}^t = \sigma_3\mathcal{X}\sigma_3, \tag{A.13}$$

where $\sigma_3$ is the Pauli matrix. Together with (A.12) this leads to

$$\bar{Q}(x) = \sigma_3 Q^t(x)\sigma_3. \tag{A.14}$$

Substituting this relation into (A.11) we arrive at (50).

To derive the differential equation (51), we use the identity

$$\left[x\partial_x + \frac{1}{2}g\partial_g, \frac{1}{1+\mathcal{H}_\ell\mathcal{X}}\right] = -\frac{1}{1+\mathcal{H}_\ell\mathcal{X}}[x\partial_x, \mathcal{H}_\ell]\mathcal{X}\frac{1}{1+\mathcal{H}_\ell\mathcal{X}}$$

$$= -\frac{1}{4}\left(\frac{1}{1+\mathcal{H}_\ell\mathcal{X}}|\Phi_\ell\rangle\rangle\langle\langle\Phi_\ell|\mathcal{X}\frac{1}{1+\mathcal{H}_\ell\mathcal{X}}\right), \qquad \text{(A.15)}$$

where we took into account (A.7) and (A.9).

Let us apply both sides of (A.15) to the states $|\Phi_\ell\rangle\rangle$ and $|\dot{\Phi}_\ell\rangle\rangle \equiv x\partial_x|\Phi_\ell\rangle\rangle$ and introduce notation for $2 \times 2$ matrices

$$P(x) = \langle x|\frac{1}{1+\mathcal{H}_\ell\mathcal{X}}|\dot{\Phi}_\ell\rangle\rangle,$$

$$v = -\langle\langle\Phi_\ell|\mathcal{X}\frac{1}{1+\mathcal{H}_\ell\mathcal{X}}|\dot{\Phi}_\ell\rangle\rangle. \qquad \text{(A.16)}$$

We take into account the relations (45) and (47) to find from (A.15)

$$\left(x\partial_x + \frac{1}{2}g\partial_g\right)Q(x) = \frac{1}{4}Q(x)u + P(x),$$

$$\left(x\partial_x + \frac{1}{2}g\partial_g\right)P(x) = \frac{1}{4}Q(x)v + \langle x|\frac{1}{1+\mathcal{H}_\ell\mathcal{X}}(x\partial_x)^2|\Phi_\ell\rangle\rangle. \qquad \text{(A.17)}$$

The last term in (A.17) can be simplified by leveraging the differential equation for the Bessel function

$$(x\partial_x)^2|\Phi_\ell\rangle\rangle = \frac{1}{4}\begin{pmatrix}(\ell^2-x)|\phi_\ell\rangle & 0 \\ 0 & ((\ell+1)^2-x)|\phi_{\ell+1}\rangle\end{pmatrix} = \frac{1}{4}\left(|\Phi_\ell\rangle\rangle L^2 - x|\Phi_\ell\rangle\rangle\right), \qquad \text{(A.18)}$$

where the matrix $L^2$ is defined in (51). In this way, we get

$$\langle x|\frac{1}{1+\mathcal{H}_\ell\mathcal{X}}(x\partial_x)^2|\Phi_\ell\rangle\rangle = \frac{1}{4}\left(Q(x)L^2 - \langle x|\frac{1}{1+\mathcal{H}_\ell\mathcal{X}}x|\Phi_\ell\rangle\rangle\right)$$

$$= \frac{1}{4}\left(Q(x)(L^2-x) - \langle x|\frac{1}{1+\mathcal{H}_\ell\mathcal{X}}[x, \mathcal{H}_\ell]\mathcal{X}\frac{1}{1+\mathcal{H}_\ell\mathcal{X}}|\Phi_\ell\rangle\rangle\right)$$

$$= \frac{1}{4}\left(Q(x)(L^2-x) + Q(x)\bar{v} - P(x)u\right), \qquad \text{(A.19)}$$

where $u$ is given by (47) and $\bar{v}$ is defined as

$$\bar{v} = -\langle\langle\dot{\Phi}_\ell|\mathcal{X}\frac{1}{1+\mathcal{H}_\ell\mathcal{X}}|\Phi_\ell\rangle\rangle = -\langle\langle\Phi_\ell|\mathcal{X}\frac{1}{1+\mathcal{H}_\ell^t\mathcal{X}}|\dot{\Phi}_\ell\rangle\rangle = \sigma_3 v\sigma_3. \qquad \text{(A.20)}$$

In deriving the last relation in (A.19) we used the identity

$$\langle x|[x, \mathcal{H}_\ell]|y\rangle = (x-y)\mathcal{H}_\ell(x,y) = \Phi_\ell(x)\dot{\Phi}_\ell(y) - \dot{\Phi}_\ell(x)\Phi_\ell(y), \qquad \text{(A.21)}$$

which follows from (42) and the second relation in (A.1).

Combining together (A.17) and (A.19), we can exclude $P(x)$ to get a differential equation for the matrix $Q(x)$

$$\left(g\partial_g + 2x\partial_x\right)^2 Q(x) = Q(x)\left(L^2 - x + \frac{1}{2}g\partial_g u + v + \bar{v} + \frac{1}{4}u^2\right). \qquad \text{(A.22)}$$

The right-hand side contains the sum of matrices $v + \bar{v}$ defined in (A.16) and (A.20). It can be determined by evaluating the matrix elements of both sides of (A.15) over the states $\langle\langle\Phi_\ell|\mathcal{X}$ and $|\Phi_\ell\rangle\rangle$. Repeating the above analysis, we obtain after some algebra

$$\left(\frac{1}{2}g\partial_g - 1\right)u = v + \bar{v} + \frac{1}{4}u^2. \qquad \text{(A.23)}$$

Substituting this relation into (A.22) we arrive at the differential equation (51).

# B  Matrix generalization of the first Szegő theorem

At strong coupling, the leading asymptotic behaviour of the Fredholm determinants of the truncated Bessel operators (36) is given by

$$F_\ell = -\frac{2g}{\pi} \int_0^\infty dz\, z\, \partial_z \log(1 + \chi(z)) + O(g^0). \tag{B.1}$$

In mathematical literature this relation is known as the first Szegő theorem [40]. In this appendix, we follow [16] and generalize the relation (B.1) to the determinant (43) involving matrix generalization of the Bessel operators.

We start with the first relation in (A.8) and rewrite it as

$$g\, \partial_g \mathcal{F}_\ell = \int_0^\infty dx\, \mathrm{tr}(\gamma(x)U)\, g\, \partial_g \chi\left(\frac{\sqrt{x}}{2g}\right), \tag{B.2}$$

where the notation was introduced for a $2 \times 2$ matrix

$$\gamma(x) = \langle x| \frac{1}{1 + \mathcal{H}_\ell \mathcal{X}} \mathcal{H}_\ell |x\rangle$$

$$= \mathcal{H}_\ell(x, x) - \int_0^\infty dy\, \chi\left(\frac{\sqrt{y}}{2g}\right) \mathcal{H}_\ell(x, y) U \mathcal{H}_\ell(y, x) + \dots \tag{B.3}$$

Here in the second relation we expanded the resolvent in powers of the operators and replaced the product of operators by a convolution of their integral kernels (37) and (42).

At strong coupling, the dominant contribution to the integrals in (B.2) and (B.3) comes from $x$ and $y$ being of order $O(g^2)$. This allows us to replace the kernel $\mathcal{H}_\ell(x, y)$ by its leading behaviour at large $x$ and $y$. Replacing the Bessel functions in (34) and (42) with their asymptotic behaviour at infinity we find

$$K_\ell(x, y) \sim \frac{1}{2\pi(xy)^{1/4}} \left[ \frac{\sin\left(\sqrt{x} - \sqrt{y}\right)}{\sqrt{x} - \sqrt{y}} - (-1)^\ell \frac{\cos\left(\sqrt{x} + \sqrt{y}\right)}{\sqrt{x} + \sqrt{y}} \right]. \tag{B.4}$$

The second term inside the brackets is a rapidly oscillating function and its contribution to (B.2) and (B.3) is subleading as compared to the first term. This leads to

$$\mathcal{H}_\ell(x, y) \sim \frac{1}{2\pi(xy)^{1/4}} \frac{\sin\left(\sqrt{x} - \sqrt{y}\right)}{\sqrt{x} - \sqrt{y}} \times \mathbf{1}. \tag{B.5}$$

We apply this relation to verify that for $g \to \infty$ and an arbitrary test function $f(x)$

$$\int_0^\infty dy\, \chi\left(\frac{\sqrt{y}}{2g}\right) \mathcal{H}_\ell(x, y) U f(y) = \chi\left(\frac{\sqrt{x}}{2g}\right) U f(x) + \dots, \tag{B.6}$$

where dots denote terms subleading at large $g$. Taking this relation into account, we find from (B.3)

$$\gamma(x) = \frac{1}{2\pi\sqrt{x} \left[ 1 + \chi\left(\frac{\sqrt{x}}{2g}\right) U \right]} + O(1/g). \tag{B.7}$$

The underlying reason for such a simplification is that at large $g$ the integrals in (B.3) are localized at $y_i = x$.

Substituting the relation (B.7) into (B.2) and changing the integration variable as $z = \sqrt{x}/(2g)$ we finally obtain

$$
\begin{aligned}
g\partial_g \mathcal{F}_\ell &= -\frac{2g}{\pi}\int_0^\infty dz\, z\, \partial_z \operatorname{tr}\log(1+\chi(z)U) + O(g^0)\\
&= -\frac{2g}{\pi}\int_0^\infty dz\, z\, \partial_z \log\det(1+\chi(z)U) + O(g^0)\\
&= -\frac{2g}{\pi}\int_0^\infty dz\, z\, \partial_z \log\Big(\sin^2\alpha + (1+\chi(z))^2\cos^2\alpha\Big) + O(g^0).
\end{aligned}
\tag{B.8}
$$

Here in the last relation we replaced the matrix $U$ with its expression (37). Comparing the last relation with (B.1), we conclude that the leading asymptotic behaviour of the Fredholm determinant (43) is given by the relation (B.1) in which $\log(1+\chi(x))$ is replaced by its matrix counterpart $\log\det(1+\chi(x)U)$.

The relation (B.7) leads to a nontrivial relation for the matrices $Q(x)$ defined in (46) and (45). To show this we introduce an auxiliary function

$$
\gamma(x,y) = \langle x|\frac{1}{1+\mathcal{H}_\ell \mathcal{X}}\mathcal{H}_\ell|y\rangle,
\tag{B.9}
$$

which is different from (B.3) in that the matrix element is not diagonal. Obviously, the function (B.3) can be obtained from (B.9) by going to the limit $y \to x$. The function (B.9) satisfies the following relation

$$
(x-y)\gamma(x,y) = \langle x|[x,\frac{1}{1+\mathcal{H}_\ell \mathcal{X}}\mathcal{H}_\ell]|y\rangle = \langle x|\frac{1}{1+\mathcal{H}_\ell \mathcal{X}}[x,\mathcal{H}_\ell]\frac{1}{1+\mathcal{X}\mathcal{H}_\ell}|y\rangle.
\tag{B.10}
$$

Replacing the commutator with (A.21) we get

$$
\gamma(x,y) = \frac{Q(x)\bar{P}(y) - P(x)\bar{Q}(y)}{x-y} = \frac{Q(x)\sigma_3 P^t(y)\sigma_3 - P(x)\sigma_3 Q^t(y)\sigma_3}{x-y},
\tag{B.11}
$$

where the matrices $Q(x)$, $\bar{Q}(x)$ and $P(x)$ are defined in (45), (A.12) and (A.16). The matrix $\bar{P}(x)$ is given by

$$
\bar{P}(x) = \langle\langle\dot{\Phi}_\ell|\frac{1}{1+\mathcal{H}_\ell \mathcal{X}}|x\rangle = \sigma_3 P^t(x)\sigma_3.
\tag{B.12}
$$

The second relation follows from (A.13) and its derivation is analogous to that of (A.14).

Going to the limit $y \to x$ in (B.11) we obtain

$$
x\gamma(x) = x\partial_x Q(x)\sigma_3 P^t(x)\sigma_3 - x\partial_x P(x)\sigma_3 Q^t(x)\sigma_3.
\tag{B.13}
$$

We can apply the first relation in (A.17) to exclude $P(x)$ from this relation and express the function $x\gamma(x)$ in terms of $Q(x)$ only. Then, we apply the differential operator $\left(x\partial_x + \frac{1}{2}g\partial_g\right)$ to both sides of the last relation and take into account the differential equation (51) to find after some algebra

$$
\left(x\partial_x + \frac{1}{2}g\partial_g\right)(x\gamma(x)) = \frac{x}{4}Q(x)\sigma_3 Q^t(x)\sigma_3.
\tag{B.14}
$$

Finally, replacing $\gamma(x)$ with its leading asymptotic behaviour (B.7) we obtain the following relation for a bilinear product of the $Q$−matrices

$$
Q(x)\sigma_3 Q^t(x)\sigma_3 = \frac{1}{\pi\sqrt{x}\left[1+\chi(\frac{\sqrt{x}}{2g})U\right]} + O(1/g).
\tag{B.15}
$$

At strong coupling, it is convenient to change variable as $x = (2gz)^2$ and switch to the functions $q_\pm(z)$ defined in (55). The matrix equation (B.15) leads to the following relations

$$q_+(z)q_-(z) = \frac{1}{\pi g z \left(e^{i\alpha} + \chi(z)\cos\alpha\right)} + \dots,$$

$$q_+(z)\bar{q}_-(z) + q_-(z)\bar{q}_+(z) = 0 + \dots, \qquad \text{(B.16)}$$

where dots denote rapidly oscillating terms and as well as terms suppressed by powers of $1/g$. For $g \to \infty$ the leading contribution to $q_+(z)$ and $q_-(z)$ comes from the first term inside the brackets in (C.21). We verified using (C.7) and (C.16) that these functions satisfy (B.16) for arbitrary $\beta(g)$.

## C Riemann-Hilbert problem

In this appendix, we derive the asymptotic expansion (65) of the functions $q_\pm(z)$ at strong coupling. Following [29], we reformulate the definition (45) as a Riemann-Hilbert problem for the matrix $Q(x)$ and proceed to solve it. The functions $q_\pm(z)$ are given by linear combinations (55) of the matrix elements $Q_{ij}(x)$.

The definition of the states (15) and (38) involves the square root of $x$. This implies that the function $Q(x)$ possesses a square root branch cut for complex values of $x$. This branch cut can be eliminated by introducing the change of variable $x = z^2$. Before proceeding further, let us investigate the properties of $Q(x)$ under the transformation that interchanges the two branches of the square root function, namely, $\sqrt{x} \to -\sqrt{x}$ or, equivalently, $z \to -z$.

Taking into account that $J_n(-z) = (-1)^n J_n(z)$ we find from (34), (42) and (44) that the matrices $\mathcal{H}_\ell(x, y)$ and $\Phi_\ell(x)$ are transformed under this transformation as

$$\mathcal{H}_\ell(x, y) \to (-1)^\ell \sigma_3 \mathcal{H}_\ell(x, y), \qquad \Phi_\ell(x) \to (-1)^\ell \sigma_3 \Phi_\ell(x). \qquad \text{(C.1)}$$

As a consequence of (53), the matrix $Q(x)$ is transformed in the same way. In components this reads

$$Q_{1i}(x) \to (-1)^\ell Q_{1i}(x), \qquad Q_{2i}(x) \to -(-1)^\ell Q_{2i}(x), \qquad \text{(C.2)}$$

where $i = 1, 2$.

It follows from the definition (45) that the function $Q(x)$ satisfies the relation

$$(1 + \mathcal{H}_\ell \mathcal{X})|Q\rangle\rangle = |\Phi_\ell\rangle\rangle. \qquad \text{(C.3)}$$

Projecting both sides on a reference state $\langle\Psi|\mathcal{H}_\ell$ and taking into account the second relation in (41), we obtain

$$\langle\Psi|\mathcal{H}_\ell(1 + \mathcal{X})|Q\rangle\rangle = \langle\Psi|\mathcal{H}_\ell|\Phi_\ell\rangle\rangle = 0. \qquad \text{(C.4)}$$

The vanishing of the matrix element on the right-hand side can be demonstrated by using (A.9). Indeed, upon taking the expectation value of both sides of (A.9) over the state $\mathcal{H}_\ell|\Psi\rangle$ we find

$$\frac{1}{4}\langle\Psi|\mathcal{H}_\ell|\Phi_\ell\rangle\rangle\langle\langle\Phi_\ell|\mathcal{H}_\ell|\Psi\rangle = \langle\Psi|\mathcal{H}_\ell[x\partial_x, \mathcal{H}_\ell]\mathcal{H}_\ell|\Psi\rangle = 0, \qquad \text{(C.5)}$$

where the last relation follows from $\mathcal{H}_\ell\mathcal{H}_\ell = \mathcal{H}_\ell$. Given that the matrix elements on the left-hand side of (C.5) are complex conjugated to one another, it follows that they must be equal to zero.

Choosing $|\Psi\rangle$ to be a linear combination of the states (38) with arbitrary coefficients, we get from (C.4)

$$\int_0^\infty dx\, \Psi_n(x)\left(1 + \chi\left(\frac{\sqrt{x}}{2g}\right)U\right)Q(x) = 0 \qquad (n \geq \ell + 1). \qquad \text{(C.6)}$$

For $n \leq \ell$ we find from (41) that $\langle \Psi_n | \mathcal{H}_\ell = 0$ and, therefore, the relation (C.4) is automatically satisfied.

We can simplify (C.6) by diagonalizing the matrix $U$ defined in (37) and decomposing $Q(x)$ over its eigenstates. Changing variable as $z = \sqrt{x}$ and introducing linear combinations of the matrix elements of $Q(x)$

$$
\Omega_1(z) = z \chi_\alpha\left(\frac{z}{2g}\right)\left[Q_{11}(x) + i Q_{21}(x)\right],
$$
$$
\Omega_2(z) = z \chi_\alpha\left(\frac{z}{2g}\right)\left[Q_{12}(x) + i Q_{22}(x)\right],
$$
$$
\chi_\alpha(z) = e^{i\alpha} + \chi(z)\cos\alpha = \frac{\cosh(z/2 + i\alpha)}{\sinh(z/2)}, \tag{C.7}
$$

we find from (C.6) that these functions satisfy a system of homogenous integral equations

$$
\int_0^\infty \frac{dz}{z} J_{2n-1+\ell}(z)\left[e^{-i\alpha}\Omega_j(z) + e^{i\alpha}\bar{\Omega}_j(z)\right] = 0,
$$
$$
\int_0^\infty \frac{dz}{z} J_{2n+\ell}(z)\left[e^{-i\alpha}\Omega_j(z) - e^{i\alpha}\bar{\Omega}_j(z)\right] = 0, \tag{C.8}
$$

where $n \geq 1$ and $j = 1, 2$.

The functions (C.7) are closely related to the functions $q_\pm(z)$ defined in (55)

$$
\Omega_1(2gz) = gz\chi_\alpha(z)e^{i\alpha/2}\left[q_+(z) + q_-(z)\right],
$$
$$
\Omega_2(2gz) = gz\chi_\alpha(z)e^{i\alpha/2}\left[q_+(z) - q_-(z)\right]. \tag{C.9}
$$

Having constructed the solutions to (C.8), we can apply these relations to determine $q_\pm(z)$ and, then, substitute them into the first relation in (62) to compute $\operatorname{tr} u = -2g\partial_g \mathcal{F}_\ell(\alpha)$.

We observe that the equations (C.8) are invariant under transformation $\Omega_i(z) \to \Lambda_{ij}(g)\Omega_j(z)$. In the similar manner, the first relation in (62) is invariant under

$$
q_+(z) \to e^{\beta(g)}q_+(z), \qquad q_-(z) \to e^{-\beta(g)}q_-(z), \tag{C.10}
$$

where $\beta(g)$ is an arbitrary function of the coupling constant. In virtue of (C.9) the two transformations are closely related to each other

$$
\Omega_1(z) \to \cosh\beta\,\Omega_1(z) + \sinh\beta\,\Omega_2(z),
$$
$$
\Omega_2(z) \to \sinh\beta\,\Omega_1(z) + \cosh\beta\,\Omega_2(z). \tag{C.11}
$$

To fix this ambiguity, we have to impose the additional conditions on the functions $q_\pm(z)$. These conditions are derived in appendix B, see (B.16).

### Properties of $\Omega$—functions

To solve the integral equations (C.8), we have to specify the analytic properties of the functions $\Omega_j(z)$. These properties can be inferred from the definition (45) of $Q(x)$,

$$
Q(x) = \Phi_\ell(x) - \langle x| \frac{\mathcal{H}_\ell \mathcal{X}}{1 + \mathcal{H}_\ell \mathcal{X}} |\Phi_\ell\rangle = \Phi_\ell(x) - \sum_{n \geq \ell+1} \Psi_n(x) R_{nm} \langle \Psi_m | \mathcal{X} | \Phi_\ell\rangle. \tag{C.12}
$$

Here in the second relation we replaced $\mathcal{H}_\ell$ with its general expression (41), applied (39) and introduced the semi-infinite matrix $R_{nm}$ which is the inverse to the matrix $(1 + K)_{nm}$ with $n, m \geq \ell + 1$.

According to (C.12), the matrix elements $Q_{ij}(x)$ are given by Neumann series over the Bessel functions (38) and (44) with real coefficients. Assuming that the sum in (C.12) is uniformly convergent for $x > 0$, we find that $Q_{ij}(x)$ are real-valued entire functions. Being combined with (C.2) and (C.7), this implies that the functions (C.7) satisfy the reality condition

$$\overline{\Omega}_j(z) = (-1)^\ell \, \Omega_j(-z). \tag{C.13}$$

At small $z$ we take into account the property of the Bessel function $J_n(z) = O(z^n)$ to deduce from (C.12) that $Q(x) = O(x^{\ell/2})$ for $x = z^2$. This leads to

$$\Omega_j(z) = O(z^\ell). \tag{C.14}$$

Finally, the factor of $\chi_\alpha(z/(2g))$ in (C.7) determines the location of zeros and poles of the functions $\Omega_j(z)$[9] Both of them are evenly spaced along the imaginary axis of the complex $z-$plane

$$\Omega_j(z) = O(z - 4\pi i g x_n), \qquad \Omega_j(z) \sim \frac{1}{z - 4\pi i g y_n}, \tag{C.15}$$

where $x_n = n + 1/2 - a$ and $y_n = n$ for $n = 0, \pm 1, \pm 2, \dots$

The integral equations (C.8), supplemented with the additional conditions (C.14) and (C.15), can be solved using the technique described in [4, 15, 23]. We refer the interesting reader to these papers and formulate below an outcome.

**Solution**

To construct the solution to (C.8), it is convenient to introduce the Wiener-Hopf type decomposition of the function (C.7)

$$\chi_\alpha(x) = \frac{\kappa}{x} \Phi_a(x) \Phi_{-a}(-x),$$

$$\Phi_a(x) = \frac{\Gamma\left(\frac{1}{2} - a\right) \Gamma\left(1 + \frac{ix}{2\pi}\right)}{\Gamma\left(\frac{1}{2} - a + \frac{ix}{2\pi}\right)}, \tag{C.16}$$

where $\kappa = 2\cos\alpha$ and $a = \alpha/\pi$. The function $\Phi_a(x)$ is analytical in the lower-half plane. It vanishes at $x = 2\pi i(n + \frac{1}{2} - a)$ (for $n = 0, 1, \dots$) and satisfies the reality condition

$$\overline{\Phi}_a(x) = \Phi_a(-x). \tag{C.17}$$

For the purpose of computing the functions $q_\pm(z)$ from (C.9), it suffices to obtain the strong coupling expansion of $\Omega_j(2gz)$. Going through the steps described in [4, 15, 23], we found a particular solution to (C.8)

$$\Omega_1(2gz) + \Omega_2(2gz) = 2i^\ell \sqrt{\frac{g\kappa}{\pi}} \left[ (igz)^a e^{2igz} \Phi_{-a}(-z) a_+(iz, g) + (-1)^\ell (-igz)^{-a} e^{-2igz} \Phi_a(z) b_-(iz, g) \right],$$

$$\Omega_1(2gz) - \Omega_2(2gz) = 2i^\ell \sqrt{\frac{g\kappa}{\pi}} \left[ (igz)^a e^{2igz} \Phi_{-a}(-z) a_-(iz, g) + (-1)^\ell (-igz)^{-a} e^{-2igz} \Phi_a(z) b_+(iz, g) \right], \tag{C.18}$$

where the normalization $z-$independent factors are inserted for convenience and the coefficient functions $a_\pm(iz, g)$ and $b_\pm(iz, g)$ are given by series in $1/g$. The important difference between the two relations in (C.18) is that these functions satisfy different conditions,

$$a_+(iz, g)/b_-(iz, g) = O(g), \qquad a_-(iz, g)/b_+(iz, g) = O(1/g). \tag{C.19}$$

---

[9]In addition, the function $\Omega_j(z)$ has an infinite set of zeros arising from linear combinations of $Q-$functions in (C.7). However their position is not prescribed.

The functions $b_+(iz, g)$ and $a_-(iz, g)$ are not independent. They can be obtained from $a_+(iz, g)$ and $b_-(iz, g)$, respectively, by substituting $a \to -a$ and $z \to -z$, see (D.10).

Substituting (C.18) the reality condition (C.13) we find that $a_\pm(x, g)$ and $b_\pm(x, g)$ are real-valued functions of $x$

$$\overline{a_\pm(x, g)} = a_\pm(x, g), \qquad \overline{b_\pm(x, g)} = b_\pm(x, g). \tag{C.20}$$

At strong coupling, these functions are given by series in $1/g$.

We recall that any linear combination (C.11) of the functions (C.18) verifies the integral equations (C.8). We combine together the relations (C.9) and (C.18) and apply the transformation (C.10) to find a general expression for the functions $q_\pm(z)$ at strong coupling

$$q_+(z) = i^\ell \frac{e^{\beta(g)}}{\sqrt{g\pi\kappa}} \left[ \frac{(gz)^a e^{2igz}}{\Phi_a(z)} a_+(iz, g) + (-1)^\ell \frac{(gz)^{-a} e^{-2igz}}{\Phi_{-a}(-z)} b_-(iz, g) \right],$$

$$q_-(z) = (-i)^\ell \frac{e^{-\beta(g)}}{\sqrt{g\pi\kappa}} \left[ \frac{(gz)^{-a} e^{-2igz}}{\Phi_{-a}(-z)} b_+(iz, g) + (-1)^\ell \frac{(gz)^a e^{2igz}}{\Phi_a(z)} a_-(iz, g) \right], \tag{C.21}$$

where $\kappa$ is defined in (C.16) and $\beta(g)$ is an arbitrary function of the coupling.

To determine the normalization factors in (C.21), we substitute the expressions for $q_\pm(z)$ into (62) and compare the coupling constant dependence on both sides. As shown in section 4, the functions on the left-hand side of (62) exhibit strong coupling behaviour: $W_0 = O(g^0)$ and $W_\pm = O(g^{\pm 2a})$. This fixes the large $g$ behavior of the product of the $q_\pm$−functions on the right-hand side of (62)

$$q_+(z)q_-(z) = O(g^{-1}), \qquad q_+^2(z) = O(g^{-2-2a}), \qquad q_-^2(z) = O(g^{-2+2a}). \tag{C.22}$$

Substituting (C.21) into these relations, neglecting rapidly oscillating terms, and taking into account (C.19) yields $a_+(iz, g) b_+(iz, g) = O(g^0)$ and $e^{\beta(g)} = O(g^{-a})$.

To fix the normalization of the coefficient functions in (C.21), we choose $e^{\beta(g)} = g^{-a}$, leading to the relation (65). Applying the relation (B.16), we find

$$a_+(iz, g) b_+(iz, g) = 1 + O(1/g). \tag{C.23}$$

The first few terms of the expansions of $a_+(iz, g)$ and $b_-(iz, g)$ are given by equation (67). The functions $b_+(iz, g)$ and $a_-(iz, g)$ can be obtained from (67) by substituting $a \to -a$ and $z \to -z$.

# D  Angular dependence

In this appendix, we discuss the dependence of various quantities introduced above on the angle $\alpha$.

According to (18), the function $\mathcal{F}_\ell(\alpha)$ is invariant under $\alpha \to -\alpha$. This property can be derived using the representation (43) by taking into account that the operator $\mathcal{X} = \mathcal{X}(\alpha)$ defined in (37) satisfies

$$\mathcal{X}^t(\alpha) = \mathcal{X}(-\alpha) = \sigma_3 \mathcal{X}(\alpha) \sigma_3, \tag{D.1}$$

where transposition acts on the $2 \times 2$ matrix $U$ in (37). The operator $\mathcal{H}_\ell$ defined in (41) and (42) satisfies analogous relation

$$\mathcal{H}_\ell^t = \mathcal{H}_\ell = \sigma_3 \mathcal{H}_\ell \sigma_3. \tag{D.2}$$

The relation (18) follows from the invariance of the determinant in (43) under replacing $\mathcal{H}_\ell$ and $\mathcal{X}$ with the transposed operators.

Let us apply the transposition to the matrix (47). Using the relations (D.1) and (D.2), we obtain from (47)

$$u^t = -\langle\langle\Phi_\ell| \frac{1}{1+\mathcal{X}^t\mathcal{H}_\ell} \mathcal{X}^t|\Phi_\ell\rangle\rangle = -\sigma_3\langle\langle\Phi_\ell| \frac{1}{1+\mathcal{X}\mathcal{H}_\ell} \mathcal{X}|\Phi_\ell\rangle\rangle\sigma_3 = \sigma_3 u \sigma_3, \qquad \text{(D.3)}$$

where we took into account that $\Phi_\ell^t = \Phi_\ell = \sigma_3\Phi_\ell\sigma_3$, see (44). This leads the following relation for the matrix $u = u(\alpha)$

$$u^t(\alpha) = u(-\alpha) = \sigma_3 u(\alpha)\sigma_3. \qquad \text{(D.4)}$$

Being written in components, this relation looks as

$$u_{11}(\alpha) = u_{11}(-\alpha), \qquad u_{22}(\alpha) = u_{22}(-\alpha), \qquad u_{21}(\alpha) = u_{12}(-\alpha) = -u_{12}(\alpha). \qquad \text{(D.5)}$$

Thus, the matrix $u(\alpha)$ only has three independent components. Its diagonal and off-diagonal elements are, respectively, even and odd functions of $\alpha$.

In the similar manner, we apply the relations (D.1) and (D.2) to get for the matrix $Q(x|\alpha) = \langle x|Q\rangle\rangle$ defined in (45)

$$Q(x|-\alpha) = \langle x| \frac{1}{1+\mathcal{H}_\ell\mathcal{X}^t} |\Phi_\ell\rangle\rangle = \sigma_3\langle x| \frac{1}{1+\mathcal{H}_\ell\mathcal{X}} |\Phi_\ell\rangle\rangle\sigma_3 = \sigma_3 Q(x|\alpha)\sigma_3. \qquad \text{(D.6)}$$

As in the previous case, this relation implies that the diagonal and off-diagonal matrix elements of (45) are, respectively, even and odd functions of $\alpha$.

We can use (D.6) to derive a relation between the functions $q_\pm = q_\pm(z|\alpha)$ defined in (55). We start with a linear combination of matrix elements in the expression for $q_+(z|\alpha)$ and apply consequently the transformations $\alpha \to -\alpha$ and $\sqrt{x} \to -\sqrt{x}$

$$Q_{11}(x) + Q_{12}(x) + iQ_{21}(x) + iQ_{22}(x) \to Q_{11}(x) - Q_{12}(x) - iQ_{21}(x) + iQ_{22}(x) \qquad \text{(D.7)}$$
$$\to (-1)^\ell [Q_{11}(x) - Q_{12}(x) + iQ_{21}(x) - iQ_{22}(x)].$$

Here in the first relation we applied (D.6) and in the second one (C.2). The expression on the second line matches a linear combination of the matrix elements in the definition of $q_-(z|\alpha)$ (see (55)). In this way, we arrive at

$$q_+(-z|-\alpha) = e^{i\alpha}(-1)^\ell q_-(z|\alpha). \qquad \text{(D.8)}$$

In a similar manner, we apply a complex conjugation to both sides of (55) to get

$$\overline{q}_\pm(z|\alpha) = e^{i\alpha/2}[Q_{11}(x) \pm Q_{12}(x) - iQ_{21}(x) \mp iQ_{22}(x)] = (-1)^\ell e^{i\alpha}q_\pm(-z|\alpha), \qquad \text{(D.9)}$$

where in the second relation we took into account (C.2).

The relations (D.8) and (D.9) follow from the definition of the $q_\pm$–functions and hold for arbitrary $g$ and $z$. For any complex $z$, the functions on both sides of (D.8) and (D.9) are evaluated at points on opposite sides of the real $z$–axis. Due to the Stokes phenomenon, the asymptotic expressions for these functions take different forms. For $z$ in the upper half-plane, the functions $q_\pm(z)$ are given by (C.21). To obtain the asymptotic expansion of these functions in the lower half-plane, it is sufficient to replace $(gz)^{\pm a} \to e^{-i\pi a}(-gz)^{\pm a}$ in (C.21). It is easy to check that the resulting expressions for $q_\pm(z)$ satisfy the relations (D.8) and (D.9), provided that $\beta(g)$ is an odd function of $\alpha$ and the coefficient functions in (C.21) are related by the transformation $\alpha \to -\alpha$

$$b_+(iz, g|\alpha) = a_+(-iz, g|-\alpha), \qquad a_-(iz, g|\alpha) = b_-(-iz, g|-\alpha). \qquad \text{(D.10)}$$

# E   Expansion around $\alpha = \pi/2$

In this appendix, we compute the determinant (43) in the double scaling limit (96) and derive the relation (111). A key observation that facilitates the subsequent analysis is that the asymptotic behaviour of the determinant (43) in the double scaling limit (96) is governed by the behaviour of the symbol function (17) around the origin.

One way to see this is to note that the $\alpha$−dependence of the coefficient functions in the strong coupling expansion (89) is carried by the functions $I_n(a)$ defined in (84). For $\alpha = \pi/2 - \delta$ and $\delta \to 0$, these functions behaves as (95). A close examination of the integral in (82) shows that for $n \geq 1$ this behaviour results from integration over the region $z = O(\delta)$, where the symbol function (17) can be replaced as $\chi(z) \sim 2/z$. At the same time, the contribution to the integral (82) from large $z$ scales as $O(\delta)$ and it is subleading for $n \geq 1$. For $n = 0$ the situation is different. For $\alpha = \pi/2 - \delta$ the corresponding function (78) is given by

$$I_0 = -2\delta + \frac{2\delta^2}{\pi} \, . \tag{E.1}$$

One can verify that the first and second terms in this relation can be obtained from (82) by replacing the function $\chi(z)$ by its leading behaviour $\chi(z) \sim 2/z$ and $\chi(z) \sim 2e^{-z}$ for small and large $z$ respectively.

We recall that (E.1) defines the coefficient of the leading $O(g)$ term in (6) and (77). This suggests that, in the double scaling limit (96), the function (6) can be split into the sum of two terms originating from small and large $z$, respectively,

$$\mathcal{F}_\ell = \mathcal{F}_{\ell,0} - \frac{\xi^2}{4\pi g} + O(1/g^2) \, . \tag{E.2}$$

Here the first term is given by (43) with the symbol function $\chi(z)$ replaced by its small $z$ expansion $\chi_0(z) = 2/z - 1/2 + O(z)$. The second term in (E.2) arises from the $O(\delta^2)$ term in (E.1).

Replacing $\chi(z) \to \chi_0(z)$ in the definition (37) of the operator $\mathcal{X}$, we can simplify its product with the operator (42) as

$$\mathcal{H}_\ell \mathcal{X}_0 = \mathcal{V}^{(0)}(\xi) + \frac{1}{g}\mathcal{V}^{(1)}(\xi) + O(1/g^2) \, . \tag{E.3}$$

The integral operators $\mathcal{V}^{(0)}(\xi)$ and $\mathcal{V}^{(1)}(\xi)$ are defined as

$$\langle x|\mathcal{V}^{(0)}(\xi)|y\rangle = \xi \begin{bmatrix} 0 & K_\ell(x,y) \\ -K_{\ell+1}(x,y) & 0 \end{bmatrix} \frac{1}{\sqrt{y}} \, ,$$
$$\langle x|\mathcal{V}^{(1)}(\xi)|y\rangle = \frac{\xi^2}{4} \begin{bmatrix} K_\ell(x,y) & 0 \\ 0 & K_{\ell+1}(x,y) \end{bmatrix} \frac{1}{\sqrt{y}} - \frac{\xi}{4} \begin{bmatrix} 0 & K_\ell(x,y) \\ -K_{\ell+1}(x,y) & 0 \end{bmatrix} , \tag{E.4}$$

where the Bessel kernel $K_\ell(x,y)$ is given by (34).

Substituting (E.3) into (43), we can expand the corresponding function $\mathcal{F}_{\ell,0}$ at large $g$ as

$$\mathcal{F}_{\ell,0} = \log\det(1 + \mathcal{V}^{(0)}) + \frac{1}{g}\mathrm{Tr}\left[\frac{1}{1 + \mathcal{V}^{(0)}}\mathcal{V}^{(1)}\right] + O(1/g^2) \, . \tag{E.5}$$

The two terms on the right-hand side can be computed by first diagonalizing the operator $\mathcal{V}^{(0)}$. We show in section 5 that the eigenspectrum of this operator can be parameterized by the positive zeros $\mu_a$ of the Bessel function, $J_\ell(\mu_a) = 0$,

$$\mathcal{V}^{(0)}|\Psi_a^\pm\rangle = \pm\frac{i\xi}{\mu_a}|\Psi_a^\pm\rangle \, , \qquad \Psi_a^\pm(x) = c_a\begin{pmatrix} K_\ell(x,\mu_a) \\ \pm iK_{\ell+1}(x,\mu_a) \end{pmatrix} . \tag{E.6}$$

The normalization factor $c_a$ can be fixed by requiring the eigenstates to be orthonormal. We have

$$\langle \Psi_a^+ | \Psi_b^\pm \rangle = c_a c_b \int_0^\infty dx \, (K_\ell(\mu_a, x) K_\ell(x, \mu_b) \pm K_{\ell+1}(\mu_a, x) K_{\ell+1}(x, \mu_b))$$
$$= c_a c_b (K_\ell(\mu_a, \mu_b) \pm K_{\ell+1}(\mu_a, \mu_b)), \tag{E.7}$$

where in the second relation we applied (A.6). It follows from (A.3) that $\langle \Psi_a^+ | \Psi_b^- \rangle = 0$. Requiring $\langle \Psi_a^+ | \Psi_b^+ \rangle = \delta_{ab}$ we obtain

$$c_a^2 = \frac{2}{J_{\ell+1}^2(\mu_a)}. \tag{E.8}$$

The first term in (E.5) is given by a product over eigenvalues of the operator $\mathcal{V}^{(0)}$ and it can be expressed in terms of the modified Bessel function, see (101) and (103). The second term in (E.5) can be expressed as the sum over the eigenstates (E.6)

$$\mathrm{Tr}\left[ \frac{1}{1+\mathcal{V}^{(0)}} \mathcal{V}^{(1)} \right] = \sum_a \frac{\langle \Psi_a^+ | \mathcal{V}^{(1)} | \Psi_a^+ \rangle}{1 + i\xi/\mu_a} + \frac{\langle \Psi_a^- | \mathcal{V}^{(1)} | \Psi_a^- \rangle}{1 - i\xi/\mu_a}. \tag{E.9}$$

Taking into account (E.4) and (E.6), the matrix elements in this relation can be evaluated as

$$\langle \Psi_a^\pm | \mathcal{V}^{(1)} | \Psi_a^\pm \rangle = \frac{\xi^2}{4} \int_0^\infty \frac{dx}{\sqrt{x}} \left[ K_\ell^2(x, \mu_a^2) + K_{\ell+1}^2(x, \mu_a^2) \mp \frac{2i\sqrt{x}}{\xi} K_\ell(x, \mu_a^2) K_{\ell+1}(x, \mu_a^2) \right]. \tag{E.10}$$

We combine the last two relations and use (A.2) to obtain

$$\mathrm{Tr}\left[ \frac{1}{1+\mathcal{V}^{(0)}} \mathcal{V}^{(1)} \right] = \int_0^\infty \frac{dx}{8\sqrt{x}} J_\ell^2(\sqrt{x}) s(x), \tag{E.11}$$

where the notation was introduced for the infinite sum over positive zeros of the Bessel function $J_\ell(z)$

$$s(x) = \sum_a \frac{2\xi^2}{(1+\xi^2/\mu_a^2)(\mu_a^2 - x)} = \xi^2 \oint \frac{dz}{2\pi i} \frac{\frac{d}{dz}\log J_\ell(z)}{(1+\xi^2/z^2)(z^2 - x)}$$
$$= \frac{\xi^2}{\xi^2 + x} \left( \frac{\xi I_{\ell+1}(\xi)}{I_\ell(\xi)} + \frac{\sqrt{x} J_{\ell+1}(\sqrt{x})}{J_\ell(\sqrt{x})} \right). \tag{E.12}$$

Here in the first relation the integration contour encircles all zeros ($\pm \mu_a$) of the Bessel function and the additional factor of $1/2$ is inserted to avoid a double counting. In the second relation, we blow up the integration contour and pick up the residues at $z = \pm\sqrt{x}$ and $z = \pm i\xi$.

Combining the above relations, we arrive at

$$\mathcal{F}_\ell = \log\left( \Gamma(\ell+1)(\xi/2)^{-\ell} I_\ell(\xi) \right) + \frac{1}{g} \mathcal{F}_\ell^{(1)}(\xi) + O(1/g^2), \tag{E.13}$$

where the leading term is given by (103) and the subleading term takes the form

$$\mathcal{F}_\ell^{(1)} = -\frac{\xi^2}{4\pi} + \int_0^\infty \frac{dx \, J_\ell^2(x)}{4(1+x^2/\xi^2)} \left( \frac{\xi I_{\ell+1}(\xi)}{I_\ell(\xi)} + \frac{x J_{\ell+1}(x)}{J_\ell(x)} \right) \tag{E.14}$$
$$= \frac{\xi^2}{\pi} \left[ -\frac{1}{4} + \frac{\xi I_{\ell+1}(\xi)}{I_\ell(\xi)} \frac{{}_2F_3\left(1,1;\frac{3}{2},\frac{3}{2}-\ell,\frac{3}{2}+\ell;\xi^2\right)}{(4\ell^2-1)} + \frac{{}_2F_3\left(1,1;\frac{3}{2},\frac{1}{2}-\ell,\frac{3}{2}+\ell;\xi^2\right)}{2(2\ell+1)} \right].$$

The relations (E.13) and (E.14) hold in the double scaling limit (96) for arbitrary $\xi$.

At large $\xi$, the first term in (E.13) is given by the transseries (106). To derive the large $\xi$ expansion of the function $\mathcal{F}_\ell^{(1)}$, it is convenient to use the integral representation (E.14). Replacing $1/(1 + x^2/\xi^2)$ on the first line in (E.14) with its Mellin-Barnes representation, we carry out the $x$−integration and obtain $\mathcal{F}_\ell^{(1)}$ as an integral of the form $\int dj\, \xi^{-2j}(\dots)$, where the integration goes parallel to the imaginary axis slightly to the left from the origin. Deforming the integration contour to the right and picking up the residues at the poles $j = 0, 1, \ldots$, we can expand (E.14) in powers of $1/\xi$ leading to

$$\mathcal{F}_\ell^{(1)} = -\frac{\xi^2}{4\pi} + \frac{\xi}{4\pi}\left[\log(2\xi) - \psi\left(\ell + \tfrac{1}{2}\right)\right] - \frac{1}{8\pi} + \frac{4\ell^2 - 5}{64\pi\xi} + \frac{4\ell^2 - 3}{32\pi\xi^2} \tag{E.15}$$
$$- \frac{48\ell^4 - 568\ell^2 + 331}{2048\pi\xi^3} + O(1/\xi^4),$$

where $\psi(x) = d\log\Gamma(x)/dx$ is the Euler function.

The relation (E.13) has to be compared with the strong coupling expansion (6). By replacing $\Delta\mathcal{F}_\ell$ in (6) with the perturbative function (89) and taking the double scaling limit (96), we can expand $\mathcal{F}_\ell$ in powers of $1/g$ as in (E.13) and identify the coefficient functions in front of powers of $1/g$. These functions depend on the Widom-Dyson constant $B_\ell(a)$. For $\alpha = \pi/2 - \delta$ it is given by

$$B_\ell(a) = B_\ell^{(0)} + \delta\, B_\ell^{(1)} + O(\delta^2), \tag{E.16}$$

where the coefficients depend on $\log\delta$. Comparing the leading $O(g^0)$ coefficient function in the expansion of (6) with the large $\xi$ expansion of the first term of (E.13), we computed the leading coefficient $B_\ell^{(0)}$ (see (107)) and verified that the two series coincide. For the subleading coefficient function we obtain from (6)

$$\mathcal{F}_\ell^{(1)} = -\frac{\xi^2}{4\pi} + \frac{\xi}{4}\left[\frac{1}{\pi}\log\left(\frac{2\pi\xi}{\delta}\right) + B_\ell^{(1)}\right] - \frac{1}{8\pi} + \frac{4\ell^2 - 5}{64\pi\xi} + \frac{4\ell^2 - 3}{32\pi\xi^2} \tag{E.17}$$
$$- \frac{48\ell^4 - 568\ell^2 + 331}{2048\pi\xi^3} + O(1/\xi^4).$$

This relation correctly reproduces the large $\xi$ expansion (E.15), enabling us to identify the correction to the Widom-Dyson constant (E.16)

$$B^{(1)}(\delta) = \frac{1}{\pi}\left(\log(\delta/\pi) - \psi\left(\ell + \tfrac{1}{2}\right)\right). \tag{E.18}$$

# F  Nonperturbative functions

It is convenient to organize the nonperturbative functions in (28) according to the total power of the nonperturbative parameters $\Lambda_-^2$ and $\Lambda_+^2$. For this purpose we can assign a degree $h = n + m$ to the function $\mathcal{F}^{(n,m)}(g)$. These functions satisfy a reality condition

$$\mathcal{F}^{(n,m)}(g|a) = \mathcal{F}^{(m,n)}(g|-a), \tag{F.1}$$

which allows us to restrict the consideration to $n \geq m \geq 0$.

Furthermore, it is advantageous to redefine the coupling constant (93) and to expand the functions $\mathcal{F}^{(n,m)}(g)$ in $1/g'$. We present below the explicit expressions of $\Lambda_-^{2n}\Lambda_+^{2m}\mathcal{F}^{(n,m)}(g)$ for lowest values of $h = n + m$:

- At degree $h = 1$, or equivalently $n = 1$ and $m = 0$, the corresponding function $\mathcal{F}^{(1,0)}(g)$ is given by (132).

- At degree $h = 2$, we have two functions, $\mathcal{F}^{(2,0)}(g)$ and $\mathcal{F}^{(1,1)}(g)$ which are given by (134) and (140), respectively. Note that the function $\mathcal{F}^{(2,0)}(g)$ is proportional to $e^{2i\pi a}$ and, therefore, it takes complex values for the generic $a$.

- At degree $h = 3$, we have two functions $\mathcal{F}^{(3,0)}(g)$ and $\mathcal{F}^{(2,1)}(g)$. The former function can be found from (134) and (137). The contribution of the latter function to (28) is given by

$$\Lambda_-^4 \Lambda_+^2 \mathcal{F}^{(2,1)}(g) = -\frac{1}{4}(-1)^\ell (1-2a)^2 \frac{\Gamma(\ell-a+1)}{\Gamma(a+\ell)} e^{i\pi a} (4\pi g')^{2a-1} e^{-4\pi g(3-2a)} \qquad \text{(F.2)}$$
$$\times \left[ (3-2a)^{-1+2a} \left( 1 - \frac{(a-1)^2 - \ell^2}{2\pi(3-2a)g'} + O(1/g'^2) \right) \right.$$
$$+ (1-2a)^{-2(1-2a)}(1+2a)^{1-2a}$$
$$\left. \times \left( 1 - \frac{2a^3 - a^2 + 1 - (2a+3)\ell^2}{2\pi(1-4a^2)g'} + O(1/g'^2) \right) \right],$$

where $g' = g + I_2(a)/4$. As expected, the coefficients of $1/g'$ are independent of $I_2(a)$. The expression on the right-hand side is proportional to $e^{i\pi a}$.

- At degree $h = 4$, we have three functions $\mathcal{F}^{(4,0)}(g)$, $\mathcal{F}^{(3,1)}(g)$ and $\mathcal{F}^{(2,2)}(g)$. As before, the function $\mathcal{F}^{(4,0)}(g)$ can be found from (134) and (137). The two remaining functions are

$$\Lambda_-^6 \Lambda_+^2 \mathcal{F}^{(3,1)}(g) = -\frac{1}{4}(1-2a)^2 \frac{\Gamma^2(\ell-a+1)}{\Gamma^2(a+\ell)} e^{2i\pi a} (4\pi g')^{4a-2} e^{-16\pi g(1-a)} \qquad \text{(F.3)}$$
$$\times \left[ (1-2a)^{-1+2a}(3-2a)^{-1+2a} \right.$$
$$\left. + (1-2a)^{-3(1-2a)}(1+2a)^{1-2a} + O(1/g') \right],$$
$$\Lambda_-^4 \Lambda_+^4 \mathcal{F}^{(2,2)}(g) = -\frac{1}{64}(1+2a)^{1-2a}(1-2a)^{1+2a} e^{-16\pi g}$$
$$\times \left[ (1+2a)^{1+2a}(3+2a)^{1-2a} \left( 1 - \frac{2(a^2 - \ell^2)}{\pi(1-2a)(3+2a)g'} + O(1/g'^2) \right) \right.$$
$$+ (1-2a)^{1-2a}(3-2a)^{1+2a} \left( 1 - \frac{2(a^2 - \ell^2)}{\pi(1+2a)(3-2a)g'} + O(1/g'^2) \right)$$
$$\left. + 2(1-2a)^{1+2a}(1+2a)^{1-2a} \left( 1 - \frac{2(a^2 - \ell^2)}{\pi(1-4a^2)g'} + O(1/g'^2) \right) \right].$$

For generic $n$ and $m$, the expression for $\Lambda_-^{2n} \Lambda_+^{2m} \mathcal{F}^{(n,m)}(g)$ is proportional to $e^{i\pi(n-m)}$.

We can use the above relations to calculate the nonperturbative corrections to (28) for $h \leq 4$. We have checked that for $a = 0$ and $\ell = 0, 1$ the resulting expression for the function (28) agrees with the exact expression (22).

The expressions for the functions $\mathcal{F}^{(n,m)}(g)$ are more complicated as compared to the analogous expressions (150) for the functions $Z^{(n,m)}(g)$. It follows from (142) that the two sets of

functions are related to each other as

$$\mathcal{F}^{(2,1)} = Z^{(2,1)} + Z^{(0,1)}(Z^{(1,0)})^2 - Z^{(1,1)}Z^{(1,0)}, \tag{F.4}$$

$$\mathcal{F}^{(3,1)} = Z^{(3,1)} - Z^{(1,0)}\left(Z^{(0,1)}(Z^{(1,0)})^2 - Z^{(1,1)}Z^{(1,0)} + Z^{(2,1)}\right),$$

$$\mathcal{F}^{(2,2)} = Z^{(2,2)} - \frac{3}{2}(Z^{(0,1)}Z^{(1,0)})^2 - Z^{(1,2)}Z^{(1,0)} - \frac{1}{2}(Z^{(1,1)})^2 + Z^{(0,1)}\left(2Z^{(1,0)}Z^{(1,1)} - Z^{(2,1)}\right).$$

Furthermore, the vanishing of the $Z-$functions (see (145)) leads to nontrivial relations between the $\mathcal{F}-$functions, e.g. $Z^{(4,1)} = \mathcal{F}^{(1,0)}\mathcal{F}^{(3,1)} + \mathcal{F}^{(4,1)} = 0$.

# G  Generalized Borel transformation

In this appendix, we outline various techniques for investigating the analytic properties of asymptotic series (for a review, see e.g. [57, 58]).

The coefficient functions (152) are expressed as divergent series

$$f(g) = \sum_{k \geq 0} \frac{f_k}{g^{k+1}}, \tag{G.1}$$

where the expansion coefficients $f_k$ grow factorially at large orders in $1/g$. These coefficients exhibit the following asymptotic behavior for large $k$

$$f_k = \frac{1}{\pi}\left[c_0 \frac{\Gamma(k+\lambda)}{(4\pi A)^{k+\lambda}} + c_1 \frac{\Gamma(k+\lambda-1)}{(4\pi A)^{k+\lambda-1}} + c_2 \frac{\Gamma(k+\lambda-2)}{(4\pi A)^{k+\lambda-2}} + \dots\right], \tag{G.2}$$

where the dots indicate subleading terms as $k \to \infty$. The asymptotic expression (G.2) depends on two parameters, $\lambda$ and $A$, along with an infinite set of coefficients $c_n$ (with $n \geq 0$). In general, the coefficients $f_k$ are given by linear combinations of the expressions (G.2) depending on different sets of parameters.

We outline below a method to determine the coefficients in (G.2) by utilizing a sufficiently large but finite number of terms from the strong-coupling expansion (G.1).

**Borel singularities.**  To regularize the series (G.1), it is convenient to introduce a generalized Borel transform

$$\mathcal{B}_p(s) = \sum_{k \geq 0} f_k \frac{s^{k+p}}{\Gamma(k+p+1)}, \tag{G.3}$$

where $p$ is an auxiliary parameter satisfying $p > -1$. For $p = 0$ this relation reduces to (154). By selecting an appropriate value of $p$, we can simplify the analysis of the Borel singularities of the series (G.1).

The function (G.1) admits an integral representation

$$f(g) = g^p \int_0^\infty ds\, e^{-gs}\mathcal{B}_p(s). \tag{G.4}$$

Substituting (G.2) into (G.3) we find that $\mathcal{B}_p(s)$ exhibits singularities at $s = A$

$$\mathcal{B}_p(s) = \frac{1}{\pi}\sum_{n \geq 0} c_n \Gamma(\lambda - p - n)(4\pi A - s)^{p+n-\lambda} + \dots, \tag{G.5}$$

where dots denote terms that are analytical at $s = 4\pi A$. The behaviour of $\mathcal{B}_p(s)$ around $s = 4\pi A$ depends on the value of $p - \lambda$. For integer $p - \lambda$, the function $\mathcal{B}_p(s)$ develops a logarithmic

branch cut at $s = 4\pi A$. For half-integer $p - \lambda$ it exhibits a square-root singularity. Due to these singularities, the integral in (G.4) is not well-defined but it can be regularized using the lateral Borel resummation $S^{\pm}$ defined in (153).

By slightly deforming the integration contour in (G.4) above or below the real axis, we obtain two distinct expressions, $S^+[f](g)$ and $S^-[f](g)$. The difference between these two functions originates from the discontinuity of the Borel transform (G.5) across the branch cut at $s > A$

$$\mathcal{B}_p(s + i\epsilon) - \mathcal{B}_p(s - i\epsilon) = 2i\theta(s - 4\pi A)\sum_{n\geq 0} c_n \frac{(s - 4\pi A)^{n+p-\lambda}}{\Gamma(1 + n + p - \lambda)}. \tag{G.6}$$

Taking this relation into account, we obtain from (G.4) and (153):

$$S^+[f](g) - S^-[f](g) = 2ig^{\lambda}e^{-4\pi gA}\sum_{n\geq 0} \frac{c_n}{g^{n+1}}. \tag{G.7}$$

Note that the expression on the right-hand side is purely imaginary for real parameters, independent of $p$ as expected, and exponentially suppressed at large $g$.

We conclude that the factorial growth of the expansion coefficients (G.2) leads to an inherent ambiguity in defining the function (G.1), manifesting as nonperturbative corrections of order $O(e^{-4\pi gA})$. The converse is also true: by using a dispersive relation for the function (G.1), one can demonstrate that the discontinuity (G.7) directly translates into the large-order behaviour of the coefficients (G.2).

An observable, such as (151), is expressed as a transseries involving coefficient functions of a form similar to (G.1) and (G.2), though with different values for the parameters $\lambda$ and $A$, which will be specified below. These functions exhibit Borel ambiguities (G.7), but these ambiguities should cancel out in their sum (151). This imposes nontrivial relationships between the various coefficient functions appearing in (151).

**Padé approximants.** Deriving the strong-coupling expansion of the function (G.1), we can only determine the first few expansion coefficients. By substituting these coefficients into (G.3), we can approximate the Borel transform (G.3) using a partial sum. The analytical properties of this partial sum differ from those of the infinite series in (G.3). To approximate the subleading corrections in (G.3), we use the first $M$ terms in (G.3) to construct a diagonal $[M/2, M/2]$ Padé approximant

$$\mathcal{B}_{\mathrm{app},p}(s) = s^p P(s)/Q(s), \tag{G.8}$$

where $P(s)$ and $Q(s)$ are polynomials of degree $M/2$. This function has $M/2$ poles coming from zeros of $Q(x)$. Based on (G.5), we expect that at large $M$ these poles have to condense on the real axis around $s = 4\pi A$.

As an example, we show in figure 2 the position of the poles of the Padé approximant for the perturbative function $\mathcal{F}_{\ell}^{(0)}$, defined in (89), for $\ell = 0$ and $a = 1/25$. We observe that the poles condense near $s = 4\pi(1 - 2a)$ and $s = -8\pi$. This approach is highly effective for identifying the location of the leading Borel singularities closest to the origin. However, it is less suited for detecting subleading branch cuts that appear at larger values of $s$. To identify these subleading cuts, we apply a conformal mapping, as described below.

To extract the value of the coefficients $c_n$ in (G.5), it is convenient to set $p = \lambda + 1/2$ in (G.5) and replace $s = 4\pi A(1 - z^2)$. This transformation maps the square-root singularities of $\mathcal{B}_{p=\lambda+1/2}(s)$ at $s = 4\pi A$ to terms involving odd powers of $z$.

$$\mathcal{B}_{\lambda+1/2}(4\pi A(1 - z^2)) = \sum_{n\geq 0}(-1)^{n+1}c_n \frac{(4\pi A)^{n+1/2}}{\Gamma(n + 3/2)}z^{2n+1} + \dots, \tag{G.9}$$

where dots denote terms with even powers of $z$. This expression should be compared with the analogous expansion of the Padé approximant (G.8).

Since the Padé approximant (G.8) is only reliable away from the singularities of the Borel transform, we cannot simply substitute $s = 4\pi A(1 - z^2)$ into (G.8) and directly expand $\mathcal{B}_{\text{app},\lambda+1/2}(s)$ around $z = 0$. To address this, we first perform the substitution $z = w + 1/\sqrt{2}$ and expand $\mathcal{B}_{\text{app},\lambda+1/2}(s)$ around $w = 0$, ensuring the expansion occurs away from the singularities. Next, we construct the diagonal Padé approximant of this expansion and re-express it in terms of powers of $z = w + 1/\sqrt{2}$. Finally, by matching the terms involving odd powers of $z$ with the expression in (G.9), we can extract the values of the coefficients $c_n$.

**Conformal mapping.** As mentioned earlier, the Borel transform (G.3) generally exhibits an (infinite) set of Borel singularities distributed along the real $s-$axis. The positions of the leading singularities, located at $s = A_-$ (on the negative axis) and $s = A_+$ (on the positive axis), can be readily identified by analyzing the poles of the Padé approximant (G.8).

To identify the subleading poles, it is useful to apply the conformal mapping

$$z = \frac{\sqrt{1 - s/A_-} - \sqrt{1 - s/A_+}}{\sqrt{1 - s/A_-} + \sqrt{1 - s/A_+}}, \tag{G.10}$$

where $A_- < 0$ and $A_+ > 0$. This transformation maps the complex $s-$plane onto the unit disk $|z| \leq 1$. The two branch cuts, $(-\infty, A_-]$ and $[A_+, \infty)$, are mapped to two arcs on the unit circle, originating from $z = -1$ and $z = 1$, respectively. Furthermore, each subleading cut at $s = s_{\text{cut}}$ is mapped to a distinct point $z = z_{\text{cut}}$ on the unit circle $|z_{\text{cut}}| = 1$.

Another advantage of the transformation (G.10) is that it resolves the leading square-root branch cut of $\mathcal{B}_{\lambda_++1/2}(s(z))$, where $s(z)$ is the inverse of the conformal mapping (G.10) and $\lambda_+$ parameterizes the large order behaviour (G.2) of the perturbative coefficients associated with the Borel singularity at $s = A_+$. Specifically, this function becomes analytic in the vicinity of $z = 1$, effectively eliminating the leading cut at $s = A_+$. However, the cut at $z = -1$ as well as the subleading cuts, which originate at distinct points $z = z_{\text{cut}}$ on the unit circle, remain.[10] Once the subleading cuts are well separated on the circle, each can be individually analyzed using the same technique previously applied to the leading cut.

For each subleading cut at $z = z_{\text{cut}}$, we associate the corresponding values of the parameters $A$ and $\lambda$. As before, we choose $p = \lambda + 1/2$ so that the Borel transform (G.5) exhibits a square-root branch cut at $s = s(z_{\text{cut}})$. In the vicinity of this cut, we introduce a parametrization $\tilde{s}(t) = s(z_{\text{cut}}(1 - t^2))$ and expand $\mathcal{B}_{\lambda+1/2}(\tilde{s}(t))$ in powers of $t$. The square-root singularities of the Borel transform $\mathcal{B}_{\lambda+1/2}(s)$ at $s = s(z_{\text{cut}})$ map to terms in the expansion of $\mathcal{B}_{\lambda+1/2}(\tilde{s}(t))$ involving odd powers of $t$. These terms can be expressed as linear combinations of the coefficients $c_n$, which parameterize the singularities of the Borel transform (G.5).

To determine the coefficients $c_n$, we first use (G.8) to construct $\mathcal{B}_{\text{app},\lambda+1/2}(\tilde{s}(t))$, and then substitute $t = w + 1/\sqrt{2}$. The resulting expression is expanded as a power series in $w$ up to order $O(w^M)$, from which we construct the diagonal Padé approximant. This approximant is then re-expanded in powers of $t$. By comparing the terms with odd powers of $t$ in the expansions of $\mathcal{B}_{\lambda+1/2}$ and $\mathcal{B}_{\text{app},\lambda+1/2}$, we can extract the coefficients $c_n$.

---

[10]If the parameter $\lambda_-$, associated with the cut at $s = A_-$, differs from $\lambda_+$ by an integer, the transformation (G.10) also eliminates the square-root branch cut at $z = -1$.

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
