# Peer review of "Exploring superconformal Yang-Mills theories through matrix Bessel kernels"

_SciPost Physics, doi:SciPost Phys. 19, 004 (2025)_

## Round 1 · Referee Report · Niklas Beisert (Referee 1) · 2025-5-8

Strengths

1- thorough and detailed derivations 2- combination of many elaborate methods and results 3- efficient non-perturbative description of particular class of observables 4- careful high-precision numerical verifications 5- useful non-trivial but tractable application of Borel resummations and resurgence towards obtaining concrete transseries result for a concrete observable in theoretical model

Report

The paper discusses the strong-coupling expansion for a particular class of observables in a 4D superconformal Yang-Mills model which can be expressed as a Fredholm determinant of a semi-infinite matrix. Such observables appear as characteristic quantities in the context of scattering amplitudes, Wilson loops and correlation functions, and their determinant expression was derived by means of AdS/CFT integrability at arbitrary strengh of the 't Hooft coupling constant λ. The matrix coefficients have an integral representation in terms of Bessel functions with exact dependence on λ, and parts of the matrix are known as integrable Bessel kernels connected to the Tracy-Widom distribution in random matrix theory.

It is known that the large λ expansion has zero radius of convergence caused by non-perturbative terms. These terms can be treated by Borel resummation in the framework of resurgence leading to a transseries representation of the result. In order to perform the Borel resummation, the coefficients of the original pertubative series need to be known explicitly and efficiently. The paper approaches the problem by transforming the Fredholm determinant to a set of integro-differential equations.

The authors follow procedures established in previous works to derive the perturbative expansion of the desired function at large λ. The derivation is carried out thoroughly and described in all detail. Some exact expressions such as the Widom-Dyson constant are guessed by symbolic extrapolation based on delimiting cases and subsequently tested carefully to high numerical precision.

After the coefficients have been determined with good efficiency, the perturbative series is analysed and Borel resummed. The resulting transseries is tested with high-precision numerics, and it is confirmed that the resurgence relations are satisfied.

Even though the paper is lengthy, it was interesting to read as the derivations have many facets, and the relevant information is spelled out to a good extent. Further details are to be found in the appendices which substantiate the calculations alongside an attached mathematica notebook containing some of the longer encountered expressions.

The journals acceptance criteria are met. The paper: 0a- Opens a new pathway in an existing research direction, with potential for follow-up work 0b- Details an exciting and relevant theoretical computation 1- Is written in a clear and intelligible way, free of unnecessary jargon, ambiguities and misrepresentations 2- Provides many details (inside the bulk sections and in appendices) so that arguments and derivations can be reproduced by qualified experts 3- Provides citations to relevant literature in a way that is as representative and complete as possible 4- Contains a clear summary of the results (with objective statements on their reach and limitations) and offers perspectives for future work 5- Contains a detailed abstract and introduction explaining the context of the problem and objectively summarizing the achievements 6- Provides (directly in appendices) reproducibility-enabling resources: explicit details of processing methods and datasets

The paper contains several minor grammar mistakes, mostly missed words in the structure of phrases.

Requested changes

1- Suggestion: The introduction, specifically the beginning, focusses on strong 't Hooft coupling, but leaves the region of interest for the rank N of the gauge group somewhat aside. As far as I understand, the results of the paper apply only to the planar limit or at least to the expansion around large N. So it would make sense specify the region of applicability somewhere near the beginning rather than by using the word planar in passing later on.

Recommendation

Publish (easily meets expectations and criteria for this Journal; among top 50%)

---

## Round 1 · Referee Report · Anonymous (Referee 2) · 2025-5-24

Strengths

Develop a systematic method for analytically studying an important quantity that is applicable across diverse contexts.

Weaknesses

A bit technical which can be hard to follow for some readers.

Report

Fredholm determinants of integrable Bessel operators arise in a wide range of physical contexts, from random matrix theory to superconformal Yang-Mills theories. In this paper, the authors investigate a unifying quantity for such Fredholm determinants, expressed as a determinant involving a matrix generalization of the Bessel operator. They develop systematic techniques for performing weak and strong coupling expansions of this quantity and demonstrate that the strong coupling expansion yields an asymptotic series. They perform the resurgence analysis and found intriguing structures of the non-perturbative contributions.

While the subject matter is inherently technical, the authors have taken care to present their work in a clear and accessible manner. The paper is well-written and makes a valuable contribution to the field. I have no further criticisms and am pleased to recommend it for publication.

Recommendation

Publish (easily meets expectations and criteria for this Journal; among top 50%)

---

## Round 1 · Referee Report · Anonymous (Referee 3) · 2025-5-31

Strengths

  1. Develop a powerful system of integro-differential equations to study a family of Fredholm determinants at finite coupling in superconformal Yang–Mills theories. 2. Carry out a systematic analysis of the strong-coupling transseries, incorporating both perturbative and non-perturbative (exponentially suppressed) corrections. 3. Uncover a rich structure of simplifications and resurgence relations in the non-perturbative sectors.

Report

In this paper, the authors investigate an interesting family of Fredholm determinants associated with integrable Bessel kernels. These objects have been shown to govern a variety of observables in superconformal Yang–Mills (SYM) theories, including, most prominently, the cusp anomalous dimension (in planar N = 4 SYM) and its tilted versions. The present work focuses on computing their strong coupling expansion in a systematic manner, capturing both the perturbative series in inverse powers of the coupling and non-perturbative, exponentially suppressed contributions.

To this end, the authors develop a powerful framework based on integro-differential equations, building on earlier techniques used in related contexts. This approach leads to a detailed transseries representation for the Fredholm determinants, controlled by two distinct non-perturbative scales. The structure of the transseries reveals intricate patterns of cancellations and resurgence, which are analyzed in depth across the various non-perturbative sectors.

In addition, the paper proposes a natural candidate for the elusive Widom-Dyson constant - an additive term not accessible through the differential equation method - and identifies simplifications that arise in special regimes, such as the double-scaling limit.

Overall, this is a technically impressive and rich contribution. The methodology is sound, the results are novel, and the writing is clear and well-organized. The work opens promising avenues for further exploration in the strong coupling dynamics of gauge theories. I enthusiastically recommend the paper for publication in SciPost.

Recommendation

Publish (easily meets expectations and criteria for this Journal; among top 50%)

---

## Editorial Decision

published